# Population Growth – Land Use Land Cover Transformations – Water Quality Nexus in Upper Ganga River Basin

Anoop Kumar Shukla[1], Chandra Shekhar Prasad Ojha[1], Ana Mijic[2], Wouter Buytaert[2], Shray Pathak[1], Rahul Dev Garg[1] and Satyavati Shukla[3]

[1]Department of Civil Engineering, Indian Institute of Technology Roorkee, Uttarakhand, India

[2]Department of Civil and Environmental Engineering, Imperial College London, London, UK

[3]Centre of Studies in Resources Engineering (CSRE), Indian Institute of Technology Bombay, Mumbai, India

E-mail- anoopgeomatics@gmail.com, cspojha@gmail.com, ana.mijic@imperial.ac.uk,

w.buytaert@imperial.ac.uk, shraypathak@gmail.com, rdgarg@gmail.com, satyashukla@iitb.ac.in

## Abstract

Upper Ganga River basin is socio-economically the most important river basins in India, which is highly stressed in terms of water resources due to uncontrolled LULC activities. This study presents a comprehensive set of analyses to evaluate the population growth-land use land cover (LULC) transformations-water quality nexus for sustainable development in this river basin. The study was conducted at two spatial scales i.e. basin scale and district scale. First, population data was analyzed statistically to study demographic changes, followed by LULC change detection over the period of February/March 2001 to 2012 [Landsat 7 Enhanced Thematic Mapper Plus (ETM+) data] using remote sensing and Geographical Information System (GIS) techniques. Trends and spatio-temporal variations in monthly water quality parameters viz. Biological Oxygen Demand (BOD), Dissolve Oxygen (DO)%, Flouride (F), Hardness $CaCO_3$, pH, Total Coliform bacteria and Turbidity were studied using Mann-Kendall rank test and Overall Index of Pollution (OIP) developed specifically for this region, respectively. Relationship was deciphered between LULC classes and OIP using multivariate techniques viz. Pearson's correlation and multiple linear regression. From the results, it was observed that population has increased in the river basin. Therefore, significant and characteristic LULC changes are observed. River gets polluted in

both rural and urban areas. In rural areas, pollution is due to agricultural practices mainly fertilizers, whereas in urban areas it is mainly contributed from domestic and industrial wastes. Water quality degradation has occurred in the river basin, consequently the health status of the river has also changed from range of acceptable to slightly polluted in urban areas. Multiple linear regression models developed for Upper Ganga River basin could successfully predict status of the water quality i.e. OIP, using LULC classes.

**Keywords:** Demographic change, Land use land cover, Overall Index of Pollution**,** Remote sensing, Upper Ganga River basin.

## 1. Introduction

Water quality is defined in terms of chemical, physical and biological (bacteriological) characteristics of the water. These characteristics may vary for different regions based on their topography, land use land cover (LULC) and climatic factors. Demographic changes, anthropogenic activities and urbanization are potential drivers affecting the quantity and quality of available water resources on local, regional and global scale. They pose threat to the quantity and quality of water resources, directly by increased anthropogenic water demands and water pollution. Indirectly, the water resources are affected by LULC changes and associated changes in water use patterns (Yu et al. 2016). In a region, urbanization occurs due to natural population growth and migration of people from rural to urban areas due to economic hardship (Bjorklund et al. 2011; Shukla and Gedam 2018). It may change natural landscape characteristics, river morphometry and increase pollutant load in water bodies. Anthropogenic activities are directly correlated with decline in the water quality (Haldar et al. 2014). In order to increase crop yield, farmers introduce various chemicals in the form fertilizers, pesticides, herbicides, etc., causing addition of pollutants to the river (Rashid and

Romshoo 2013; Yang et al. 2013). In urban areas, pollutants are introduced from leachates of
landfill sites, stormwater runoff and direct dumping of waste (Tsihrintzis and Hamid 1997).
LULC and water quality indicator parameters are often used in water quality assessment
studies (Kocer and Sevgili 2014; Liu et al. 2016; Sanchez et al. 2007; Tu 2011).
LULC changes may alter the chemical, physical and biological properties of a river system
viz. Biological Oxygen Demand (BOD), temperature, pH, Chloride (Cl), Colour, Dissolved
Oxygen (DO), Hardness $CaCO_3$, Turbidity, Total Dissolved Solids (TDS), etc. (Ballestar et
al. 2003; Chalmers et al. 2007; Smith et al. 1999). Several studies have been carried out
across the world to understand this phenomenon. Hong et al. (2016) studied the effects of
LULC changes on water quality of a typical inland lake of an arid region in China. The study
concluded that water pollution is positively correlated to agricultural land and urban areas
whereas negatively correlated to water and grassland. Li et al. (2012) studied effects of
LULC changes on water quality of the Liao River basin, China. In this river basin water
quality of upstream was found better than downstream due to less influence from LULC
changes in the region. Similarly, impact of LULC changes was studied on Likangala
catchment, southern Malawi. Even though the water quality remained in acceptable class, the
downstream of the river was found polluted with increase in the number of *E.Coli* and
cations/anions (Pullanikkatil et al. 2015). The composition and distribution of benthic
macroinvertebrate assemblage were studied in the Upper Mthatha River, Eastern Cape, South
Africa (Niba and Mafereka 2015). Results revealed that the distribution of the benthic
macroinvertebrate assemblage is affected by season, substrate and habitat heterogeneity.
LULC changes induce changes into the river water which affects their species distribution.

Water quality changes of the Ganga river, at various locations in Allahabad were studied for post-monsoon season by Sharma et al. (2014) using Water Quality Index (WQI) and statistical methods. Considerable water quality deterioration was observed at various locations due to the vicinity of the river to a highly urbanized city of Allahabad. A combination of water quality indices viz. Canadian WQI by Canadian Council of Ministers of the Environment (CCME-WQI), Oregon Water Quality Index (OWQI) and National Sanitation Foundation Water Quality Index (NSF-WQI) were used to analyse the pollution of Sapanca Lake Basin (Turkey) and a good relationship was observed between the indices and parameters. Eutrophication was identified as a major threat to Sapanca Lake and stream system (Akkoyunlu and Akiner 2012). A river has capability to reduce its pollutant load, also known as self-purification (Hoseinzadeh et al. 2014). In extreme situations, degradation of river ecosystem caused by anthropogenic factors can be irreversible. Hence, it is crucial to understand the effects of demographic changes and LULC transformations on water quality for pollution control and sustainable water resources development in a river basin (Milovanovic 2007; Teodosiu et al. 2013).

Ganga River is extremely significant to its inhabitants as it supports various important services such as: (i) source of irrigation for farmers in agriculture and horticulture; (ii) provides water for domestic and industrial purposes in urban areas; (iii) source of hydro-power; (iv) serves as a drainage for waste and helps in pollution control; (v) acts as support system for terrestrial and aquatic ecosystems, (vi) provides religious and cultural services; (vii) helps in navigation; (viii) supports fisheries and other livelihood options, etc. (Amarasinghe et al. 2016; SoE report, 2012; Watershed Atlas of India, 2014). However, for the past few decades Upper Ganga River basin has experienced rapid growth in population, urbanization, industrialization, infrastructure development activities and agriculture. Due to

these changes, maintaining the acceptable water quality for various uses is being challenged.
Therefore, there is a need of comprehensive study to understand the causative connection
(nexus) between the changing patterns of population, LULC and water quality in this river
basin.

Remote sensing and GIS are efficient aids in preparing and analyzing spatial datasets such as
satellite data, Digital Elevation Model (DEM), etc. Remote sensing technology is used in
preparing LULC maps of a region whereas GIS helps in delineation of river basin boundaries,
extraction of study area, hydrological modeling, spatio-temporal data analysis, etc. (Kindu et
al. 2015; Kumar and Jhariya 2015; Wilson 2015). Selection of appropriate method for a study
is based on the objectives and availability of the data/tools required for the study. Ban et al.
(2014) observed that water quality monitoring programs monitor and produce large and
complex water quality datasets. Water quality trends vary both spatially and temporally,
causing difficulty in establishing relationship between water quality parameters and LULC
changes (Phung et al. 2015; Russell 2015). Assessment of surface water quality of a river
basin can be done using various water quality/pollution indices based on environmental
standards (Rai et al. 2011). These indices are simplest and fastest indicators to evaluate the
status of water quality in a river (Hoseinzadeh et al. 2014). Demographic growth, LULC
changes and their effects on water quality in a region are very site specific. Hence, different
regions/countries have developed their own water quality/pollution indices for different types
of water uses based on their respective water quality standards/permissible pollution limits
(Abbasi and Abbasi 2012; Rangeti et al. 2015).

There are various water quality indices available worldwide that can be used for water quality
assessment e.g. Composite Water Quality Identification Index (CWQII) (Ban et al. 2014);
River Pollution Index (RPI), Forestry Water Quality Index (FWQI) and NSF-WQI
(Hoseinzadeh et al. 2014); Canadian Water Quality Index (CWQI) (Farzadkia et al. 2015);
Comprehensive water pollution index of China (Li et al. 2015); Prati's implicit index of
pollution (Prati et al. 1971); Horton's index, Nemerow and Sumitomo Pollution Index,
Bhargava's index, Dinius second index, Smith's index, Aquatic toxicity index, Chesapeake
Bay water quality indices, Modified Oregon WQI, Li's regional water resource quality
assessment index, Stoner's index, Two-tier WQI, CCME-WQI, DELPHI water quality index,
Universal WQI, Overall index of pollution (OIP), Coastal WQI for Taiwan, etc. (Abbasi and
Abbasi 2012; Rai et al. 2011). Currently, not sufficient literature is available on comparisons
between all the above mentioned water quality indices based on clusters, differences, validity,
etc. However in a study, comparison was made between CCME and DELPHI water quality
indices based on multivariate statistical techniques viz. coefficient of determination ($R^2$), root
mean square error, and absolute average deviation. Results revealed that the DELPHI method
had higher predictive capability than the CCME method (Sinha and Das 2015). There is no
universally accepted method for development of water quality indices. Therefore, there is no
established method by which 100% objectivity or accuracy can be achieved without any
uncertainties. There is continuing interest across the world to develop accurate water quality
indices that suit best for a local or regional area. Each water quality index has its own merits
and demerits (Sutadian et al. 2016; Tyagi et al 2013).

Water quality management and planning in a river basin requires an understanding of the
cumulative pollution effect of all the water quality indicator parameters under consideration.
This helps in assessing the overall water quality/pollution status of the river in a given space
and time, in a specific region. In this study, a WQI called 'Overall Index of Pollution' (OIP)
developed specifically for Indian conditions by Sargoankar and Deshpande (2003) is used to
assess the health status of surface waters across Upper Ganga River basin. A number of
studies have successfully used OIP to assess the surface water quality of various Indian
rivers. The concentration ranges used in the class indices and Individual Parameter Indices
(IPIs) assisted in evaluating the changes in individual water quality parameters whereas OIP
assessed the overall water quality status of Indian rivers. This index helped to identify the
parameters that are affected due to pollution from various sources. It is immensely helpful in
studying the spatial and temporal variations in the surface water quality of both rural and
urban subbasins due to the influence of demographic and LULC changes. The self-cleaning
capacity of the river system investigated using OIP helped to comprehend the resilience
capacity of the river system against the changes occurring in water quality due to
anthropogenic activities. OIP has been used successfully to study the surface water quality
status of the two most important and highly polluted rivers of the tropical Indian region viz.
Ganga and Yamuna. It is also used for water quality assessment of comparatively smaller
river like Chambal River and Sukhna lake of Chandigarh (Chardhry et al. 2013; Katyal et al.
2012; Shukla et al. 2017; Sargaonkar and Deshpande 2003; Yadav et al. 2014). Therefore,
OIP is used in the present study as an effective tool to communicate the water quality
information. In the recent years, combinations of multivariate statistical techniques viz.
Pearson's correlation, regression analyses, etc. have been used successfully to study the links
between LULC changes and water quality (Attua et al. 2014; Gyamfi et al. 2016; Hellar-
Kihampa et al. 2013).

The main objective of this study is to understand the *causative connection (nexus)* between
the changing patterns of population growth-LULC transformations-water quality of water
stressed Upper Ganga River basin through a comprehensive set of analyses. The present
study is conducted at two different spatial scales i.e. (a) at complete river basin level (small
scale), and (b) at district level (large scale) to evaluate the changes at both regional and local
scales. The effect of different seasons viz. pre-monsoon, monsoon and post-monsoon on the
water quality is also examined. A relationship is developed between LULC and OIP using
Pearson's correlation and multiple linear regression. Findings from this research work may
help engineers, planners, policy makers and different stakeholders for sustainable
development in the Upper Ganga River basin.

**2. Study area**
The Upper Ganga River basin (UGRB) is experiencing rapid rate of change in LULC and
irrigation practices. A part of the Upper Ganga River basin is selected as the study area (Fig.
1). It is located partly in Uttarakhand, Uttar Pradesh, Bihar and Himanchal Pradesh states of
India and covers a total drainage area of 2,38,348 km$^2$. The geographical extent of the river
basin is between 24° 32' 16"−31° 57' 48" N to 76° 53' 33"−85° 18' 25" E. The altitude ranges
from 7500 m in the Himalayan region to 100 m in the lower Gangetic plains. Some mountain
peaks in the headwater reaches are permanently covered with snow. Annual average rainfall
in the UGRB is in the range of 550-2500 mm (Bharati and Jayakody 2010). Major rivers
contributing to this river basin are Bhagirathi, Alaknanda, Yamuna, Dhauliganga, Pindar,
Mandakini, Nandakini, Ramganga, Tamsa (Tons), etc. Tehri Dam constructed on Bhagirathi
River is an important multipurpose hydropower project along with several other smaller
hydropower projects of low capacity. This region comprises of major cities and towns such as
Allahabad, Kanpur, Varanasi, Dehradun, Rishikesh, Haridwar, Moradabad, Bareilly Bijnor,
Garhmukteshwar, Narora, Farrukhabad, Badaun, Chandausi, Amroha, Kannauj, Unnao,
Fatehpur, Mirzapur, etc. Most predominant soil groups found in this region are alluvial, sand,
loam, clay and their combinations. Due to favorable agricultural conditions majority of the
population practices agriculture and horticulture. However, a large portion of the total

population lives in cities located mainly along Ganga River. Most of them work in urban or

industrial areas.

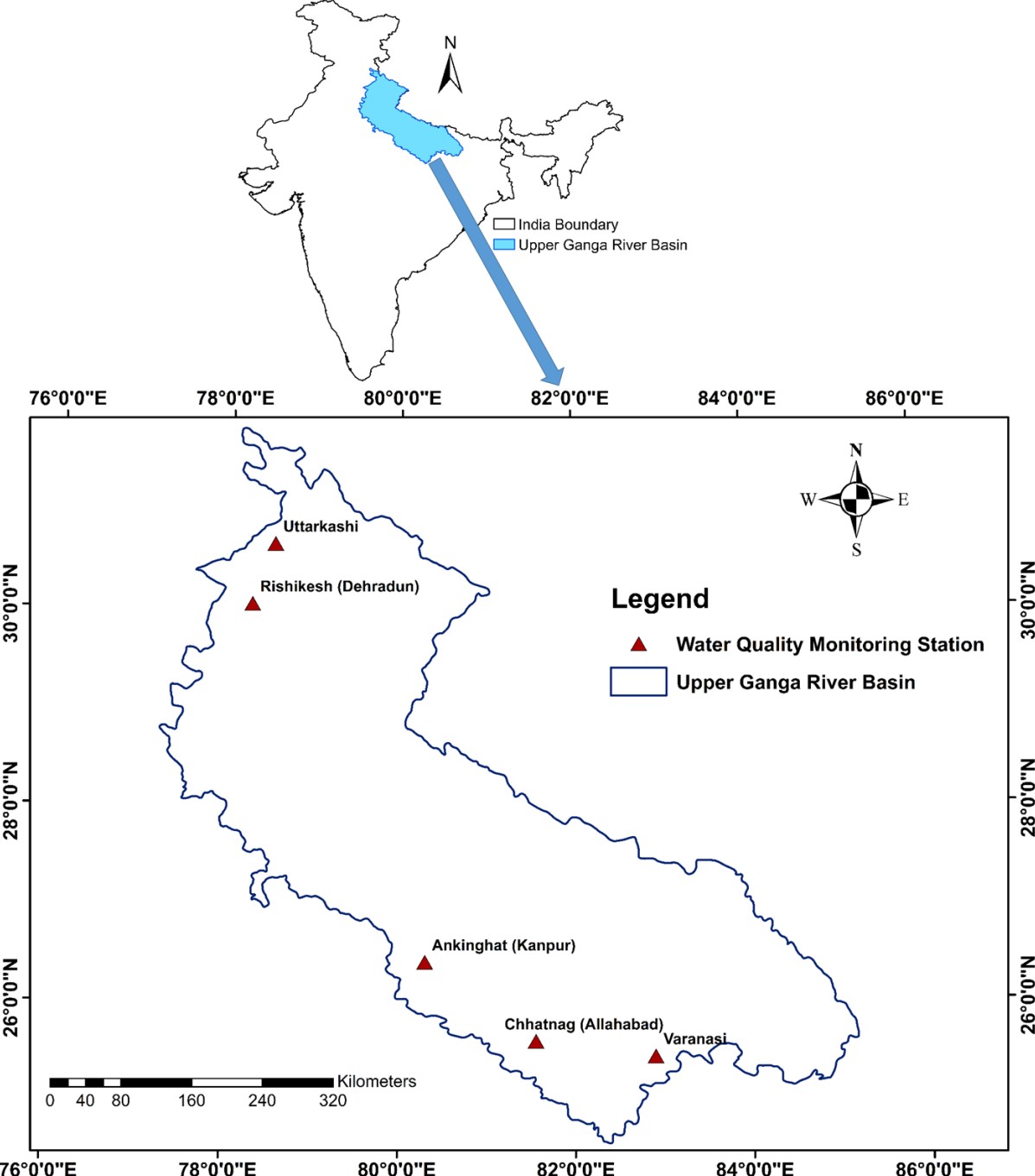

**Figure 1.** Location map of the study area in northern India and water quality monitoring stations across Upper Ganga River basin

**3. Data acquisition**

In this study, broadly two types of dataset were used which are listed below: (i) Spatial
dataset: (a) Shuttle Radar Topography Mission (SRTM) 1 arc-second global Digital Elevation
Model (DEM) of 30 m spatial resolution; and (b) Landsat 7 Enhanced Thematic Mapper Plus
(ETM+) images, 23 in total, for the month of February/March in 2001 and 2012, having 30 m
spatial resolution. Both SRTM DEM and time series Landsat dataset were collected from
United States Geological Survey (USGS), (USGS 2016); (c) Survey of India toposheets of
1:50,000 scale from Survey of India (SoI), Government of India (GoI); (d) Published LULC,
water bodies, urban land use and wasteland maps from Bhuvan Portal, Indian Space Research
Organization (ISRO), GoI (Bhuvan 2016). SoI toposheets and published maps were used as
reference to improve the LULC classification results; and (e) For ground truthing of prepared
LULC maps, Ground Control Points (GCPs) were collected using Global Positioning System
(GPS) during the field visit and Google Earth.

(ii) Non-spatial dataset were acquired from various departments of GoI: (a) Census records
and related reports of the years 2001 and 2011 from Census of India (Census of India 2011);
(b) Reports on LULC statistics from Bhuvan Portal, ISRO, GoI; (c) Monthly water quality
dataset (BOD, DO%, Flouride (F), Hardness $CaCO_3$, pH, Total Coliform Bacteria and
Turbidity) of the year 2001-2012 from Central Water Commission (CWC); and (d) Water
quality reports from Central Pollution Control Board (CPCB), Uttar Pradesh Pollution
Control Board (UPPCB), CWC and National Remote Sensing Centre (NRSC), ISRO, GoI.

**4. Data preparation and methodology**
**4.1 Delineation of the river basin**
This section discusses the data preparation and step-by-step methodology carried out in this
study. Flowchart of the methodology is illustrated in Fig. 2. First, a field reconnaissance
survey was conducted in the Upper Ganga River basin, India to understand the study area.
The global SRTM DEM (30 m spatial resolution) was pre-processed by filling sinks in the
dataset using ArcGIS 10.1 Geo-processing tools. Further, Upper Ganga River basin boundary
was delineated following a series of steps using ArcHydro tools. The following base layers
were manually digitized for the study area viz. stream network, railway lines, road network,
major reservoirs, canals and settlements using SoI topographic maps and updated further with
recent available Landsat ETM+ dataset of the year 2012.

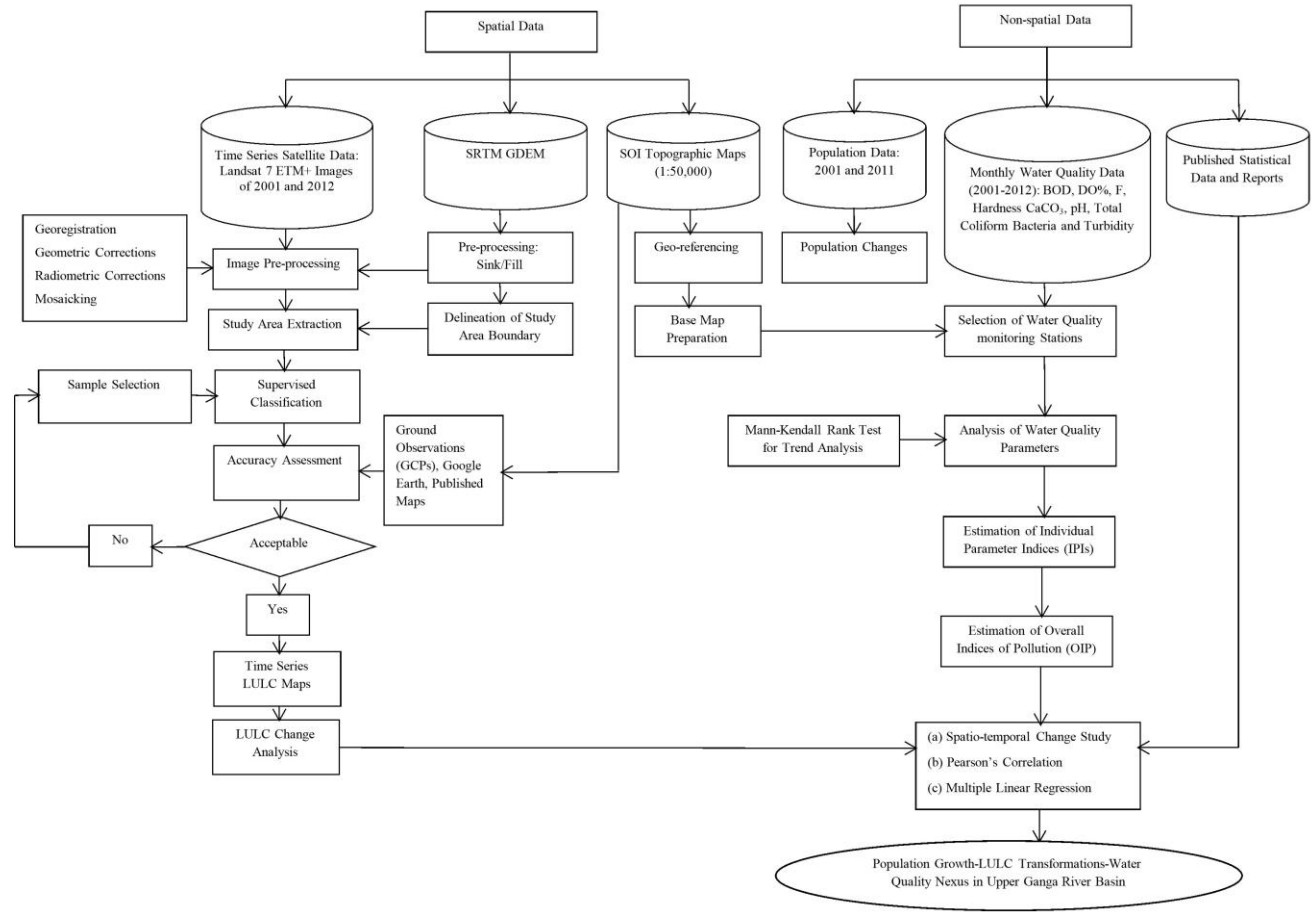


**Figure 2.** Flowchart illustrating methodology and steps followed in the study

**4.2 Population analysis**
Census of India, GoI provided village wise population data for rural areas and ward/city wise
population data for urban areas for the years 2001 and 2011. Village and ward wise
population data of 77 districts, falling into Upper Ganga River basin were identified and
organized into rural and urban population. Total population and population growth rate
(PGR) were statistically estimated for 77 individual districts and for the complete study area
over the years 2001 and 2011. Population growth rates were also estimated for rural and
urban populations. In addition, the total population and population growth rates were
estimated for upper and lower reaches of the study area. These comprehensive analyses were
done to understand the demographic changes occurring in the study region.

**4.3 LULC mapping and change detection**
For LULC mapping and change analysis, preprocessing of the time series satellite dataset is
required (Lu and Weng 2007). Landsat 7 ETM+ dataset of the years 2001 and 2012 were
downloaded from USGS website. Each year consisted of 23 images of February/March
months. Images of same months were used to reduce errors in LULC change detection due to
different seasons. Due to failure in Scan Line Corrector (SLC) of the Landsat 7 satellite, the
images of year 2012 had scan line errors, which resulted in 22% of data gap in each scene.
However, with only 78% of data availability per scene, it is some of the most radiometrically
and geometrically accurate satellite dataset in the world and therefore it is still very useful for
various studies (USGS 2018). For heterogeneous regions, Neighbourhood Similar Pixel
Interpolator (NSPI) is the simple and most effective method to interpolate the pixel values
within the gaps with high accuracy (Chen et al. 2011; Gao et al. 2016; Liu and Ding 2017;
Zhu et al. 2012; Zhu and Liu 2014). Therefore to correct scan line errors, IDL code for NSPI
algorithm developed by Chen et al. (2011) was run on ENVI version 5.1. This algorithm
filled the data gaps in the satellite images with high accuracy i.e. Root Mean Square Error
(RMSE) of 0.0367.

Further, satellite images were georeferenced to a common coordinate system i.e. Universal
Transverse Mercator Zone 43 N with World Geodetic System (WGS) 1984 datum for proper
alignment of features in the study area. Total 75 control points were chosen from Survey of
India (SoI) toposheets of scale 1:50,000, which were used as base map for georectification.
To make the two satellite images comparable, a good radiometric consistency and proper
geometric alignment is required. But it is difficult to achieve due differences in atmospheric
conditions, satellite sensor characteristics, phonological characteristics, solar angle, and
sensor observation angle on different images (Shukla et al. 2017). A relative geometric
correction (image to image coregistration) method was employed to maintain geometric
consistency of both the satellite images using Polynomial Geometric Model and Nearest
Neighbour resampling method. The recent Landsat ETM+ image of 2012 was used as
reference image for co-registration and the image of 2001 was georectified with respect to it.
Root Mean Square Error (RMSE) of less than 0.5 was used as criteria for geometric
corrections of the images to ensure good accuracy (Gill et al 2010; Samal and Gedam 2015).

To reduce the radiometric errors and get the actual reflectance values, the Topographic and
Atmospheric Correction for Airborne Imagery (ATCOR-2) algorithm available in ERDAS
Imagine 2016 was used. SRTM DEM was used to derive the characteristics viz. slope, aspect,
shadow and skyview. This algorithm provided a very good accuracy in removing haze, and in
topographic and atmospheric corrections of the images (Gebremicael et al. 2017; Muriithi
2016). Finally, image regression method was applied on the images to normalize the
variations in the pixel brightness value due to multiple scenes taken on different dates.

The images were mosaicked and study area was extracted. Total 2014 Ground Control Points
(GCPs) were collected from GPS (dual frequency receiver: SOKKIA: Model No. S-10)
survey during the field visit and from Google Earth, with horizontal accuracy in the range of
2-5 m. 1365 GCPs were used to train the Maximum Likelihood Classifier (MLC) and the
remaining 649 points (collected from GPS) were later used for accuracy assessment. Out of
1365 GCPs, 830 GCPs were collected using GPS survey and remaining 535 were collected
from Google Earth images. In the present study, to account for spatial autocorrelation among
different LULC features, before image classification an exploratory spectral analysis was
carried out using histograms of each band to understand the spectral characteristics of the
LULC features. The spatial autocorrelation was analysed using semivariogram function
which is measured by setting variance against variable distances (Brivio et al. 1993). The
estimated semivariogram was plotted to assess the spatial autocorrelation in respective bands
in the satellite image. The range and shape (piecewise slope) of the semivariograms were
examined visually to determine the appropriate sizes for training data, window size and
sampling interval for spatial feature extraction (Chen 2004; Xiaodong et al. 2009).

A window size of $7 \times 7$ was chosen for sampling the training data, which gives the better
classification results on Landsat ETM+ images (Wijaya et al. 2007). While developing the
spectral signatures for different LULC classes, information acquired from band histograms
and Euclidean distances were used for class separability. SoI topographic maps, Google Earth
images, published LULC, water bodies, urban land use and wasteland maps of Bhuvan Portal
were used as reference to improve the LULC classification results. Due to higher confusion
between barren land and urban areas at few places, urban areas were classified independently
by masking these on the image. Uncertainties in misclassification between forest and
agricultural land were reduced by adding more training samples. This significantly improved
the classification accuracy (Gebremicael et al. 2017). Hence, Maximum Likelihood Classifier
(MLC) of supervised classification approach was used to classify the time series images into
six LULC classes, viz. snow/glaciers, forests, built-up lands, agricultural lands, water bodies
and wasteland. LULC distribution was estimated for the years 2001 and 2012. Due to lack of
ground truth data of the year 2001, the accuracy assessment was done for the LULC of the
year 2012. Both time series satellite dataset are of Landsat ETM+ with same spatial
resolution of 30 m and a large number of GCPs are available for the year 2012.  Hence,
LULC map of year 2012 would represent the overall accuracy of both the maps. A simple
random sampling of 649 test pixels belonging to corresponding image objects were selected
and verified against reference data.

In this sampling method, selection of sample units was done in such a way that every possible
distinct sample got the equal chance of selection. This sampling method provided
comparatively better results on the large image size following the rule of thumb
recommended by Congalton i.e. minimum 75-100 samples should be selected per LULC
category for large Images (Congalton 1991; Foody 2002; Goncalves et al. 2007; Hashemian
et al. 2004; Kiptala et al. 2013; Samal and Gedam 2015). Following the Congalton's thumb
rule for better accuracy in simple random sampling, GCPs were selected in the range of 94-
137 for each LULC class in proportion to their areal extent on the image. Therefore,
sufficient spatial distribution of the sampling points was achieved for each LULC class.
Accuracy assessment results were presented in confusion matrix showing characteristic
coefficients viz. User's accuracy, Producer's accuracy, Overall accuracy and Kappa
coefficients. The confusion matrix gave the ratio of number of correctly classified samples to
the total number of samples in the reference data. The User's accuracy (errors of commission)
and Producer's accuracy (errors of omission) expressed the accuracy of each LULC types
whereas the overall accuracy estimated the overall mean of user accuracy and producer
accuracy (Campbell 2007; Congalton 1991; Jensen 2005). The Kappa coefficient denoted the
agreement between two datasets corrected for the expected agreement (Gebremicael et al.
2017). Further, post classification change detection method was employed for comparing
LULC maps of 2001 and 2012. This method provided comparatively accurate results than
image difference method (Samal and Gedam 2015). LULC distribution and change statistics
between the years 2001 and 2012 were estimated for individual districts and for complete
UGRB.

**4.4 Water quality analysis**
**4.4.1 Selection of water quality monitoring stations**
To understand the impact of LULC transformations on water quality of the UGRB, two water
quality monitoring stations viz. Uttarkashi and Rishikesh were chosen in the upper reaches of
the river basin. This part of the river basin comprises of highly undulating terrain with
moderately less anthropogenic influences. Moreover, three water quality monitoring stations
viz. Ankinghat (Kanpur), Chhatnag (Allahabad) and Varanasi were selected in the lower
reaches of the river basin. This part of the river basin falls under Gangetic plains with
extreme anthropogenic activities. Spatio-temporal changes in the water quality of these
monitoring stations were examined over a period of the year 2001-2012 and LULC-OIP
relationship was studied using various statistical analyses viz. Mann Kendall rank test, OIP,
Pearson's correlation and multiple linear regression.

**4.4.2 Mann-Kendall test on monthly water quality data**

A non-parametric Mann-Kendall rank test (Mann 1945; Kendall 1975) was performed on the
seven monthly water quality parameters viz. BOD, DO%, F, Hardness $CaCO_3$, pH, Total
Coliform Bacteria and Turbidity, observed at the five water quality monitoring stations to
understand the existing trends in the water quality parameters of the years 2001-2012. In this
test, the null hypothesis $H_o$ assumed that there is no trend (data is independent and randomly
ordered) and it was tested against the alternative hypothesis $H_1$, which assumes that there is a
trend. The standard normal deviate (Z-statistic) was computed following a series of steps as
given by Helsel and Hirsch 1992; and Shukla and Gedam 2018. The positive value of Z test
showed a rising trend and a negative value of it indicates a falling trend in the water quality
data series. The significance of Z test was observed on confidence level of 90%, 95% and
99%. The test was performed on monthly water quality data of January to December of the
years 2001-2012. Standard Deviation (SD) was estimated separately for each month.

**4.4.3 Estimation of OIP**
For selecting water quality index, the following criteria is followed (Abbasi and Abbasi,
2012; Horton 1965): (i) limited number of variables should be handled by the used index to
avoid making the index unwieldy; (ii) the variables used in the index should be significant in
most areas, (iii) only reliable data variables for which the data are available should be
included. Hence, seven most relevant water quality parameters in Indian context i.e. BOD,
DO%, Total Coliform (TC), F, Turbidity, pH and Hardness $CaCO_3$ that are affected due to
changes in LULC are chosen. BOD, DO%, and Total Coliform (TC) are the parameters
mainly affected by urban pollution. F, Turbidity and pH are general water quality parameters
affected by both natural and anthropogenic factors. However, Hardness $CaCO_3$ is a parameter
affected mainly by agricultural activities and urban pollution.

In the present study, Overall Index of Pollution (OIP) developed by Sargaonkar and
Deshpande (2003) is used which is a general water quality classification scheme developed
specifically for tropical Indian conditions where, in the proposed classes (C1:Excellent;
C2:Acceptable; C3:Slightly Polluted; C4:Polluted; and C5:Heavily Polluted water), the
concentration levels/ranges of the significant water quality indicator parameters are defined
with due consideration to the Indian water quality standards (Indian Standard Specification
for Drinking Water, IS-10500, 1983; Central Pollution Control Board, Government of India,
classification of inland surface water, CPCB- ADSORBS/3/78-79). Wherever, the water
quality criteria were not defined, international water quality standards [Water quality
standards of European Community (EC); World Health Organization (WHO) guidelines;
standards by WQIHSR; and Tehran Water Quality Criteria by McKee and Wolf] were used.
It was observed that different agencies use different, indicator parameters,
terminologies/definitions for classification scheme and criteria such as Action Level,
Acceptable Level, Guide Level, and Maximum Allowable Concentration, etc. for different
uses of water. Hence, a common classification scheme was required to be defined to
understand the water quality status in terms of pollution effects of the water quality
parameters being considered. Table 1 illustrates the OIP classification scheme and the ranges
of concentrations of the parameters under consideration. The basis on which the
concentration levels for each of the parameters in the given classes are selected, are described
below (Sargaonkar and Deshpande 2003):

*Turbidity***:** According to the Indian Standards for Drinking Water (IS 10500, 1983) and
European Community (EC) water quality standards, 10 NTU is maximum desirable level/
maximum admissible level for turbidity. Therefore, in the OIP classification scheme this
value is considered for class C2 (Acceptable) water quality. As per WQIHSR standards and
WHO Guidelines, 5 NTU is considered as maximum acceptable level, hence it is considered
in class C1 (Excellent). 10-250 NTU is considered as Good water quality, and >250 NTU as
poor water quality by the Wolf and McKee water quality criteria. Therefore, accordingly the
Turbidity is split into the following ranges: 10-100 for class C3 (Slightly Polluted), 100-250
for class C4 (polluted) and >250 as class C5 (heavily polluted) water quality.

**BOD:** For BOD, the classification given by Prati et al. (1971) is used which conforms with
the CPCB water quality standards i.e. for class "A" water (drinking water) , BOD values
should be 2 mg/L and for class "B" water (outdoor bathing), BOD values should be 3 mg/L.
According to EC water quality standards, for freshwater fish water quality or recreational use
the guide level and maximum admissible level should be 3 and 6 mg/L respectively. And
according to McKee and Wolf water quality scheme, the BOD of >2.5 indicates poor water
quality. Hence, in OIP classification scheme, for classes C3 (Slightly Polluted), C4 (Polluted)
and C5 (Heavily Polluted) water quality, the higher concentration values are assigned in
geometric progression.

**DO%:** The maximum DO at a given space and time is the
function of water temperature. It is highly variable and specific to a location. The average
tropical temperature of India is 27°C and 8 mg/L is the corresponding average DO saturation
concentration reported from studies, which represents 100% DO concentration and applies to
class C1. During day time, in eutrophic water bodies with high organic loading very high DO
concentration is observed which is undesirable situation. Therefore, in the OIP classification
scheme for DO% in a particular class, the concentration ranges on both lower and higher
sides of the average DO% level are considered. The ranges of %DO concentration defined
are illustrated in Table 1.

**F:** As Fluoride is a toxic element, the classification criteria for it is more stringent. According to Indian standards for drinking water (IS 10500, 1983), the desirable limit for Fluoride is 0.6-1.2 mg/L which is considered under class C1 in OIP classification scheme. According to EC standards for surface water (potable abstraction) and action level in WHO Guidelines, the mandatory limit for F is 1.5 mg/L which is considered the maximum level in class C2. 1.5-3.0 mg/L of F is considered as good water quality but the concentration >3.0 mg/L indicates poor water quality according to McKee and Wolf water quality standards. Hence, for class C3 (slightly polluted) water quality, the concentration value of 2.5 mg/L is used. The F concentration >1.5 mg/L is bad for human health as it can result in tooth decay and further higher levels can cause bone damage through Fluorosis. Therefore, concentration values of 6.0 and >6.0 mg/L is used for classes C4 and C5 respectively.

455

**Hardness CaCO₃:** As per Indian standards for drinking water, the desirable limit (maximum) for hardness is 300 mg/L whereas the concentration value of 500 mg/L is indicated as action level according to WHO Guidelines. Hence, accordingly the ranges of Hardness were taken as: class C1 as 0-75 mg/L, class C2 as 75-150 mg/L, class C3 as 150-300 mg/L, class C4 as 300-500 mg/L and >500 mg/L in class C5.

461

**pH:** According to CPCB, ADSORBS/3/78-79, pH range of 6.5 to 8.5 is considered for classes A (drinking water), B (outdoor bathing) and D (Propagation wild life, fisheries, recreation and aesthetic). EC standards guide limit for surface waters (potable abstractions) is 5.5-9.0. Hence, based on these the concentration level of pH in the OIP classification scheme is defined for classes C1-C5, as given in Table 1.


***Total Coliform*:** In the given OIP scheme, for class C1, C2 and C3 the Coliform bacteria
count of 50, 500 and 5000 MPN/100 mL respectively as specified in CPCB classification of
inland surface water is considered. Coliform count range of 50-100, 100-5000 and >5000 is
considered as excellent, good and poor water quality respectively by McKee and Wolf water
quality criteria. EC bathing water standards consider count of 10000 MPN/100 mL as the
maximum admissible level, therefore, the concentration range 5000-10000 is assigned to
class C4 which indicates polluted water quality and makes the criteria more stringent. The
count of >10000 indicates heavily polluted water and therefore, it was assigned to class C5.

After the concentration level/ranges were assigned to each parameter in the given classes, the
information on water quality data was transformed in discrete terms. Different water quality
parameters are measured in different units. Therefore, in order to bring the different water
quality parameters into a commensurate unit so that the integrated index can be obtained to
be used for decision making, an integer value 1, 2, 4, 8 and 16 (also known as Class Index
Score as given in Table 1) was assigned to each class i.e. C1, C2, C3, C4 and C5 respectively
in geometric progression. The number termed as class index indicated the pollution level of
water in numeric terms and it formed the basis for comparing water quality from Excellent to
Heavily Polluted (Table 1). For each of the parameter concentration levels, the mathematical
expressions were fitted to obtain this numerical value called an index ($P_i$) or (IPI) which
indicated the level of pollution for that particular parameter. Table 2 illustrates these
mathematical equations. The value function curves, wherein, on the Y-axis the concentration
of the parameter is taken and on the X-axis index value is plotted for each parameter. The
figures of value function curves for important water quality parameters used in OIP scheme
can be referred from Sargaonkar and Deshpande (2003). The value function curves provide
the pollution index ($P_i$) or (IPI) for individual pollutants. For any particular given
concentration, the corresponding index can be read directly from these curves or can be
estimated using mathematical equations given for the value function curves as illustrated in
Table 2. Hence, IPIs were calculated for each parameter at a given time interval. Finally, the
Overall Index of Pollution (OIP) is calculated as the mean of ($P_i$) or IPIs of all the seven
water quality parameters considered in the study and mathematically it is given by expression

498    (1):

$$Overall\ Index\ of\ Pollution\ (OIP) = \frac{\Sigma_i P_i}{n}$$

499                                                                                                            (1)

Where, $P_i$ is the pollution index for the $i^{th}$ parameter, i=1, 2,…., n and n denotes the number
of parameters. Finally, OIP was estimated for each water quality monitoring station across
the UGRB over a period of 2001 to 2012. It gave the cumulative pollution effect of all the
water quality parameters on the water quality status of a particular monitoring station in a
given time. For each water quality monitoring station of UGRB, the OIP was estimated for
three primary seasons i.e. pre-monsoon, monsoon and post-monsoon seasons. The
interpretation of IPI values for individual parameter index or OIP values to determine the
overall pollution status is done as follows: The index value of 0-1 (class C1) indicates
Excellent water quality, 1-2 (class C2) indicates Acceptable, 2-4 (class C3) indicates Slightly
Polluted, 4-8 (class C4) indicates Polluted and 8-16 (class C5) indicates Heavily Polluted
water. The upper limit of the range is to be included in that particular class. In case some
additional relevant water quality parameters are required to be considered, an updated OIP
can be developed using methodology given by Sargaonkar and Deshpande (2003). The
mathematical value function curves can be plotted for the new parameters to get the
mathematical equations which will help to calculate IPIs. As OIP uses an additive
aggregation method, the average of IPIs of all the parameters will estimate updated OIP.

**Table 1.** Classification scheme of water quality used in OIP (Source: Sargoankar and Deshpande 2003)

| Classification | Class | Class Index (Score) | Concentration Limit / Ranges of Water Quality Parameters | | | | | | |
| --- | --- | --- | --- | --- | --- | --- | --- | --- | --- |
| | | | BOD (mg/L) | DO (%) | F (mg/L) | Hardness CaCO$_3$ (mg/L) | pH (pH unit) | Total Coliform (MPN/100 mL) | Turbidity (NTU) |
| Excellent | C$_1$ | 1 | 1.5 | 88-112 | 1.2 | 75 | 6.5-7.5 | 50 | 5 |
| Acceptable | C$_2$ | 2 | 3 | 75-125 | 1.5 | 150 | 6.0-6.5 and 7.5-8.0 | 500 | 10 |
| Slightly Polluted | C$_3$ | 4 | 6 | 50-150 | 2.5 | 300 | 5.0-6.0 and 8.0-9.0 | 5000 | 100 |
| Polluted | C$_4$ | 8 | 12 | 20-200 | 6.0 | 500 | 4.5-5 and 9-9.5 | 10000 | 250 |
| Heavily Polluted | C$_5$ | 16 | 24 | <20 and >200 | <6.0 | >500 | <4.5 and >9.5 | 15000 | >250 |


 **Table 2.** Mathematical expressions for value function curves (Source: Sargoankar and

Deshpande 2003)

| S. No. | Parameter | Concentration Range | Mathematical Expressions |
|--------|-----------|---------------------|--------------------------|
| 1. | BOD | <2 | $x = 1$ |
| | | 2-30 | $x = y/1.5$ |
| 2. | DO% | ≤50 | $x = \exp(-(y - 98.33)/36.067)$ |
| | | 50-100 | $x = (y - 107.58)/14.667$ |
| | | ≥100 | $x = (y - 79.543)/19.054$ |
| 3. | F | 0-1.2 | $x = 1$ |
| | | 1.2-10 | $x = ((y/1.2) - 0.3819)/0.5083$ |
| 4. | Hardness CaCO$_3$ | ≤75 | $x = 1$ |
| | | 75-500 | $x = \exp(y + 42.5)/205.58$ |
| | | >500 | $x = (y + 500)/125$ |
| 5. | pH | 7 | $x = 1$ |
| | | >7 | $x = \exp((y - 7.0)/1.082)$ |
| | | <7 | $x = \exp((7 - y)/1.082)$ |
| 6. | Total Coliform | ≤50 | $x = 1$ |
| | | 50-5000 | $x = (y/50)**0.3010$ |
| | | 5000-15000 | $x = ((y/50) - 50)/16.071$ |
| | | >15000 | $x = (y/15000) + 16$ |
| 7. | Turbidity | ≤10 | $x = 1$ |
| | | 10-500 | $x = (y + 43.9)/34.5$ |


## 4.5 Statistical analysis

Due to religious, economic and historical importance of River Ganga, the most important
cities/districts of UGRB are present in the proximity to River Ganga. The water quality of
selected monitoring stations is highly influenced by type of activities undergoing in the
district where they are located. In a study, buffer zones of different thresholds were created
surrounding a water quality monitoring station to determine the dominant LULC class that
affects the water quality of that particular station (Kibena et al. 2014). However, in UGRB
the population data was available at district level not at buffer level. Districts selected in this
study consisted of both urban and rural areas. District wise LULC change was extremely
helpful in comprehending the water quality changes at the local scale and to identify source
of pollutants at a particular monitoring station. Whereas LULC changes at the basin level
provided a broad outlook on the status of water quality of the complete study area which is
also very useful for some applications. Though the spatial/mapped data could be more useful
and relevant when compared with remote sensing data, but the monitoring stations in the
UGRB were scarce. Therefore, over a relatively large study area, the interpolation maps
generated using OIP were not likely to provide very good comparison results with LULC
changes. Hence, districts were chosen as a unit and district wise population and LULC
distribution were related to water quality (OIP) of the monitoring stations to comprehend the
nexus between them.

Various methods/models are already developed to study effects of LULC changes on water
quality. However, these methods could not be applied directly to a region because of the
differences in the data availability, climatic, topographic and LULC variations that may
introduce errors. Necessary modifications were made in the present evaluation methodology
as required. Due to unavailability of the continuous data on population, satellite based LULC
and water quality at desired interval in UGRB, establishing the interrelationship between
these factors is not trivial. Therefore, to develop the relationship between LULC classes and
water quality (OIP), a 2-time slice analysis was done for the years 2001 and 2012 with
seasonal component. Multivariate statistical analyses viz. Pearson's Correlation and multiple
linear regression were employed between LULC classes (independent variable) and OIP
(dependent variable). Pearson's Correlation determined strength of association between the
variables whereas prediction regression model was developed using multiple linear
regression.

**5. Results and discussion**
Section 5.1 presents the results of population changes in the districts of UGRB and complete
study area. Section 5.2 presents the accuracy assessment results of LULC map, followed by
Section 5.3, where the LULC distribution across the study area is discussed both at basin
scale and at district scale. Section 5.4 presents the trend analysis results of monthly water
quality data. In Section 5.5 population growth-LULC transformation-water quality nexus has
been described for complete UGRB, whereas Section 5.6 presents it for the five districts
separately. Finally, Section 5.7 described the relationship between LULC and water quality
(OIP).

**5.1 Population dynamics**

Analysis of the population dataset of the years 2001 and 2011 acquired from Census of India,
GoI reveals that in the UGRB, out of the 77 districts that fall in four different states, viz.
Uttar Pradesh, Uttarakhand, Bihar and Himanchal Pradesh, total population and PGR has
increased in 74 districts. With majority of the districts showing population increase, the total
population of UGRB has increased consequently (Table 3). The population growth rate
(PGR) of 20.45% is observed in the total population of UGRB from 2001 to 2011. Table 3
illustrates that the PGR is ≥20% in the districts having bigger urban agglomerations or cities
e.g. Agra, Allahabad, Bahraich, Ghaziabad, Lucknow, Kanpur (Dehat+Nagar), Varanasi,
Patna, etc. However, Almora, Pauri Garhwal and Shravasti are showing decreasing PGR. It is
to be observed that these are either hilly or very small towns with poor employment

opportunities. People migrate from these locations to nearby cities, therefore, decreasing the

PGR. It was noticed from Census of India reports that the population density of Dehradun

(Rishikesh), Kanpur, Allahabad and Varanasi districts are much higher against the average

population density of Ganga River basin, i.e. 520 per square km. Varanasi is one of the most

populated districts in the country.

**Table 3.** Table showing total population and Population Growth Rate (PGR) % in the census

years 2001 and 2011

| S. No. | Districts | Total Population (2001) | Total Population (2011) | Population Growth Rate (PGR) % |
|---|---|---|---|---|
| 1 | Agra | 36,20,436 | 44,18,797 | 22.1 |
| 2 | Aligarh | 29,92,286 | 36,73,889 | 22.8 |
| 3 | Allahabad | 49,36,105 | 59,54,391 | 20.6 |
| 4 | Almora | 6,30,567 | 6,22,506 | -1.3 |
| 5 | Ambedkar Nagar | 20,26,876 | 23,97,888 | 18.3 |
| 6 | Azamgarh | 39,39,916 | 46,13,913 | 17.1 |
| 7 | Bageshwar | 2,49,462 | 2,59,898 | 4.2 |
| 8 | Baghpat | 11,63,991 | 13,03,048 | 11.9 |
| 9 | Bahraich | 23,81,072 | 34,87,731 | 46.5 |
| 10 | Ballia | 27,61,620 | 32,39,774 | 17.3 |
| 11 | Balrampur | 16,82,350 | 21,48,665 | 27.7 |
| 12 | Barabanki | 26,73,581 | 32,60,699 | 22.0 |
| 13 | Bareilly | 36,18,589 | 44,48,359 | 22.9 |
| 14 | Basti | 20,84,814 | 24,61,056 | 18.0 |
| 15 | Bhojpur | 22,43,144 | 27,28,407 | 21.6 |
| 16 | Bijnor | 31,31,619 | 36,82,713 | 17.6 |
| 17 | Budaun | 30,69,426 | 36,81,896 | 20.0 |
| 18 | Bulandshahar | 29,13,122 | 34,99,171 | 20.1 |
| 19 | Buxar | 14,02,396 | 17,06,352 | 21.7 |
| 20 | Chamoli | 3,70,359 | 3,91,605 | 5.7 |
| 21 | Champawat | 2,24,542 | 2,59,648 | 15.6 |
| 22 | Dehradun | 12,82,143 | 16,96,694 | 32.3 |
| 23 | Deoria | 27,12,650 | 31,00,946 | 14.3 |
| 24 | Etah | 15,61,705 | 17,74,480 | 13.6 |
| 25 | Faizabad | 20,88,928 | 24,70,996 | 18.3 |
| 26 | Farrukhabad | 15,70,408 | 18,85,204 | 20.0 |
| 27 | Fatehpur | 23,08,384 | 26,32,733 | 14.1 |
| 28 | Firozabad | 20,52,958 | 24,98,156 | 21.7 |
| 29 | Gautam Buddha Nagar | 12,02,030 | 16,48,115 | 37.1 |
| 30 | Ghaziabad | 32,90,586 | 46,81,645 | 42.3 |
| 31 | Ghazipur | 30,37,582 | 36,20,268 | 19.2 |
| 32 | Gonda | 27,65,586 | 34,33,919 | 24.2 |

| 33 | Gopalganj | 21,52,638 | 25,62,012 | 19.0 |
|---|---|---|---|---|
| 34 | Gorakhpur | 37,69,456 | 44,40,895 | 17.8 |
| 35 | Hardoi | 33,98,306 | 40,92,845 | 20.4 |
| 36 | Haridwar | 14,47,187 | 18,90,422 | 30.6 |
| 37 | Hathras | 13,36,031 | 15,64,708 | 17.1 |
| 38 | Jaunpur | 39,11,679 | 44,94,204 | 14.9 |
| 39 | Jyotiba Phule Nagar | 14,99,068 | 18,40,221 | 22.8 |
| 40 | Kannauj | 13,88,923 | 16,56,616 | 19.3 |
| 41 | Kanpur Dehat | 15,63,336 | 17,96,184 | 14.9 |
| 42 | Kanpur Nagar | 41,67,999 | 45,81,268 | 9.9 |
| 43 | Kaushambi | 12,93,154 | 15,99,596 | 23.7 |
| 44 | Kheri | 32,07,232 | 40,21,243 | 25.4 |
| 45 | Kinnaur | 78,334 | 84,121 | 7.4 |
| 46 | Kushinagar | 28,93,196 | 35,64,544 | 23.2 |
| 47 | Lucknow | 36,47,834 | 45,89,838 | 25.8 |
| 48 | Maharajganj | 21,73,878 | 26,84,703 | 23.5 |
| 49 | Mainpuri | 15,96,718 | 18,68,529 | 17.0 |
| 50 | Mau | 18,53,997 | 22,05,968 | 19.0 |
| 51 | Meerut | 29,97,361 | 34,43,689 | 14.9 |
| 52 | Mirzapur | 21,16,042 | 24,96,970 | 18.0 |
| 53 | Moradabad | 38,10,983 | 47,72,006 | 25.2 |
| 54 | Muzaffarnagar | 35,43,362 | 41,43,512 | 16.9 |
| 55 | Nainital | 7,62,909 | 9,54,605 | 25.1 |
| 56 | Patna | 47,18,592 | 58,38,465 | 23.7 |
| 57 | Pauri Garhwal | 6,97,078 | 6,87,271 | -1.4 |
| 58 | Pilibhit | 16,45,183 | 20,31,007 | 23.5 |
| 59 | Pithoragarh | 4,62,289 | 4,83,439 | 4.6 |
| 60 | Pratapgarh | 27,31,174 | 32,09,141 | 17.5 |
| 61 | Rae Bareli | 28,72,335 | 34,05,559 | 18.6 |
| 62 | Rampur | 19,23,739 | 23,35,819 | 21.4 |
| 63 | Rudraprayag | 2,27,439 | 2,42,285 | 6.5 |
| 64 | Sant Kabir Nagar | 14,20,226 | 17,15,183 | 20.8 |
| 65 | Sant Ravidas Nagar | 13,53,705 | 15,78,213 | 16.6 |
| 66 | Saran | 32,48,701 | 39,51,862 | 21.6 |
| 67 | Shahjahanpur | 25,47,855 | 30,06,538 | 18.0 |
| 68 | Shravasti | 11,76,391 | 11,17,361 | -5.0 |
| 69 | Siddharthnagar | 20,40,085 | 25,59,297 | 25.5 |
| 70 | Sitapur | 36,19,661 | 44,83,992 | 23.9 |
| 71 | Siwan | 27,14,349 | 33,30,464 | 22.7 |
| 72 | Sultanpur | 32,14,832 | 37,97,117 | 18.1 |
| 73 | Tehri Garhwal | 6,04,747 | 6,18,931 | 2.3 |
| 74 | Udhamsingh Nagar | 12,35,614 | 1,648,902 | 33.4 |
| 75 | Unnao | 27,00,324 | 31,08,367 | 15.1 |
| 76 | Uttarkashi | 2,95,013 | 3,30,086 | 11.9 |
| 77 | Varanasi | 31,38,671 | 36,76,841 | 17.1 |
| Total | Upper Ganga River basin | 17,11,86,859 | 20,61,88,401 | 20.45 |


Ganga River basin is the most sacred as well as populated river basins in India that is
endowed with varying topography, climate and mineral rich alluvial soils in the Gangetic
Plains area. Due to high soil fertility in the region, 60% of the population practice agricultural
activities especially in the Gangetic Plains or lower reaches of the UGRB. This accounts for
the high rural population in the region. Due to hilly terrain in the upper reaches of the basin,
the population is less compared to the lower reaches of the basin. Due to its religious and
economic significance, a large number of densely populated cities and towns are located on
the banks of the river mainly in the Gangetic Plain region. These cities have large growing
populations and an expanding industrial sector (NRSC 2014).

Growth rates for urban and rural areas of upper and lower reaches of UGRB were calculated
from official statistics (Fig. 3). It brings forth the clear picture of comparatively high rise in
the rural population of lower reaches. Urban population has also increased along with rural
population in the lower reaches (Fig. 3a). Both rural and urban population have increased in
upper reaches but the growth is relatively less than lower reaches. However, PGR is higher in
urban areas of both reaches between 2001 -2011, which indicates urbanization of the region
(Fig. 3b). After Dehradun city was declared capital of the Uttarakhand state in the year 2000
and due to subsequent industrialization in the region, the PGR of the upper reaches has
increased. Hence, population rise in UGRB is due to natural population growth and migration
of the people from remote/rural areas to urban areas.








**(a)**

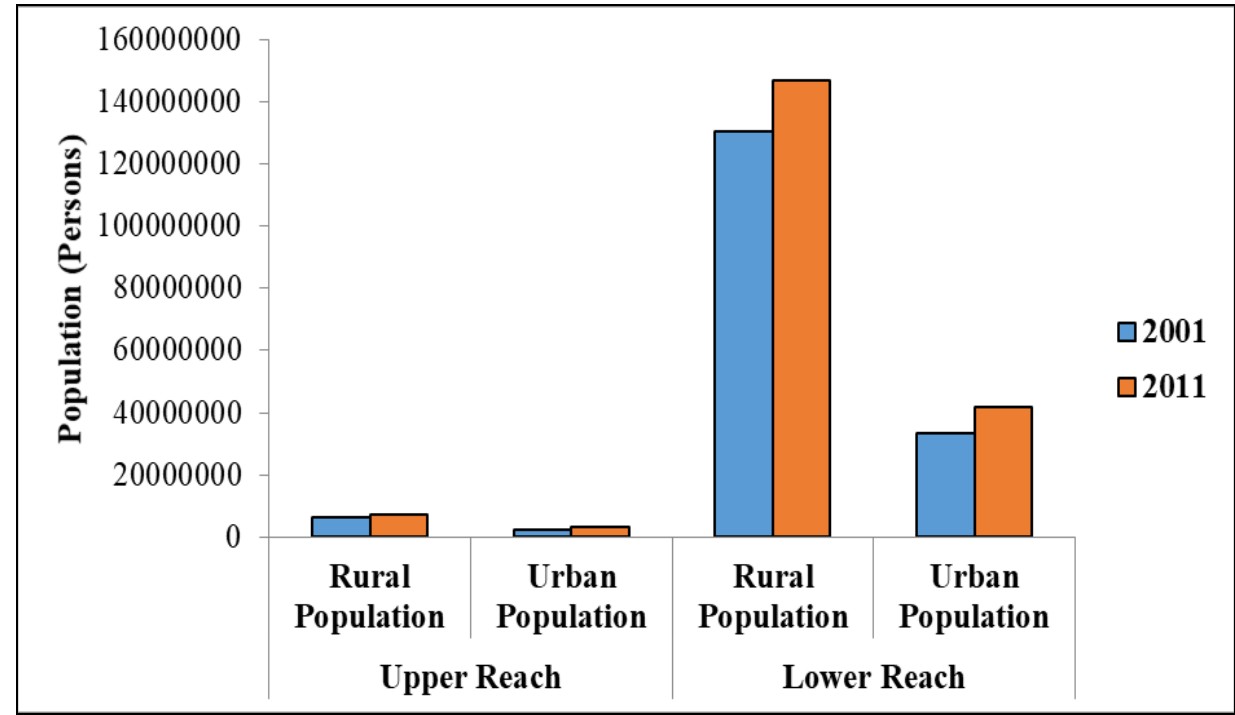


**(b)**

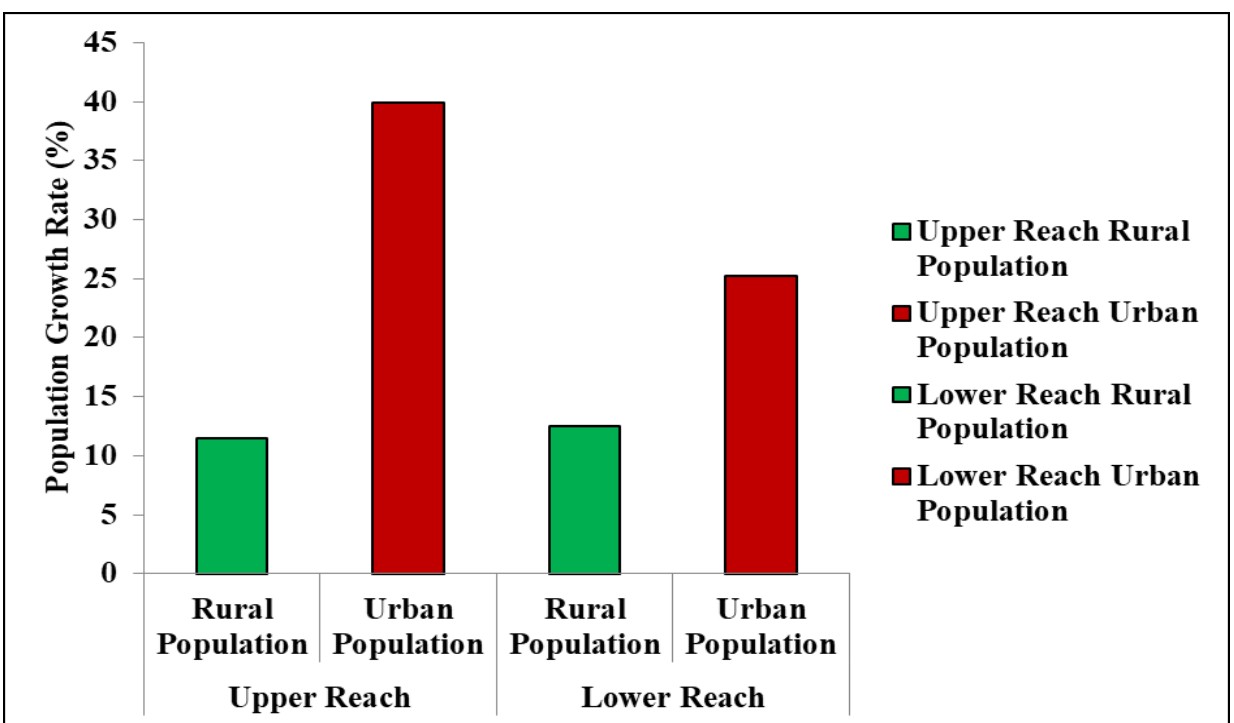


**Figure 3:** Growth in the rural and urban population of upper and lower reaches of UGRB
between 2001-2011 **(a)** Total population, and **(b)** Population Growth Rate (PGR)

 **5.2 Accuracy assessment of LULC map**

Post accuracy assessment, the cross-tabulation (confusion matrix) of the mapped LULC
classes against that observed on the ground (or reference data) for a sample of cases at
specified locations are presented in Table 4. From the results it is observed that spectral
confusion is common between few classes. For e.g. frozen snow/glaciers are sometimes
misclassified as built-up or wasteland whereas melted ones are misinterpreted as water
bodies. Similarly, forest areas are wrongly depicted as agricultural lands at few occasions.
Sometimes barren rocky wastelands are misclassified as built-up and wastelands having
shrubs/grasses are misjudged as agricultural lands. Therefore, in terms of producer's accuracy
all classes are over 90%, except for three classes i.e. forest, wasteland and snow/glacier,
while in terms of user's accuracy, all the classes are very close to or more than 90% (Table
4). Both producer's and user's accuracy are found to be consistent for all LULC classes. For
the past LULC map, a similar level of accuracy can be expected with a very little deviation.
An overall classification accuracy of 90.14% was achieved with Kappa statistics of 0.88,
showing good agreement between LULC classes and reference GCPs. From the accuracy
assessment results, it is evident that the present classification approach has been effective in
producing LULC maps with good accuracy.

**Table 4.** Accuracy assessment of the 2012 LULC map produced from Landsat ETM+ data,
representing both the confusion matrix and the Kappa statistics

| Classified Data | Reference Data | | | | | | Row Total | User's Accuracy (%) | Overall Kappa Statistics |
|---|---|---|---|---|---|---|---|---|---|
| | AG | BU | F | SG | WL | WB | | | |
| AG | 128 | 0 | 6 | 0 | 3 | 0 | 137 | 93.43 | |
| BU | 2 | 96 | 2 | 5 | 1 | 0 | 106 | 90.57 | |
| F | 11 | 0 | 88 | 3 | 0 | 3 | 105 | 83.81 | |
| SG | 0 | 4 | 1 | 103 | 2 | 1 | 111 | 92.79 | |
| WL | 1 | 2 | 0 | 7 | 82 | 2 | 94 | 87.23 | 0.88 |
| WB | 0 | 0 | 1 | 1 | 6 | 88 | 96 | 91.67 | |

| Column Total | 142 | 102 | 98 | 119 | 94 | 94 | **649** | | |
|---|---|---|---|---|---|---|---|---|---|
| Producer's Accuracy (%) | 90.14 | 94.12 | 89.80 | 86.55 | 87.23 | 93.62 | | | |
| Overall Classification Accuracy (%) | 90.14 | | | | | | | | |


* AG = Agricultural Land, BU = Built-up, F = Forest, SG = Snow/Glacier, WL = Wasteland
and WB = Water Bodies

## 5.3 Distribution of LULC

The LULC maps of the UGRB for February/March 2001 and 2012 are shown in Fig. 4.
District boundaries of the five districts i.e. Uttarkashi, Dehradun, Kanpur, Allahabad and
Varanasi, chosen for district wise LULC analysis are highlighted in this figure. The gross
percentage area in each LULC class and their changes from 2001 to 2012 in UGRB are
illustrated in Fig. 5. From the results it is observed that the agricultural lands, built-up, forest,
and snow /glaciers have increased whereas the water bodies and wasteland have decreased.
The highest % change is observed in built-up class that has increased by 43.4%. In 2001,
17.1% of wastelands were present in the study area which have reduced to 11.4%. Therefore,
the wastelands are the second most dynamic category with the significant decrease of 33.6%.
Agriculture land, forest and snow/glaciers have also increased by 2.9%, 14.5% and 1.1%
respectively. Conversely, water bodies have decreased from 2.0% in 2001 to 1.8% in 2012
(Fig. 5).

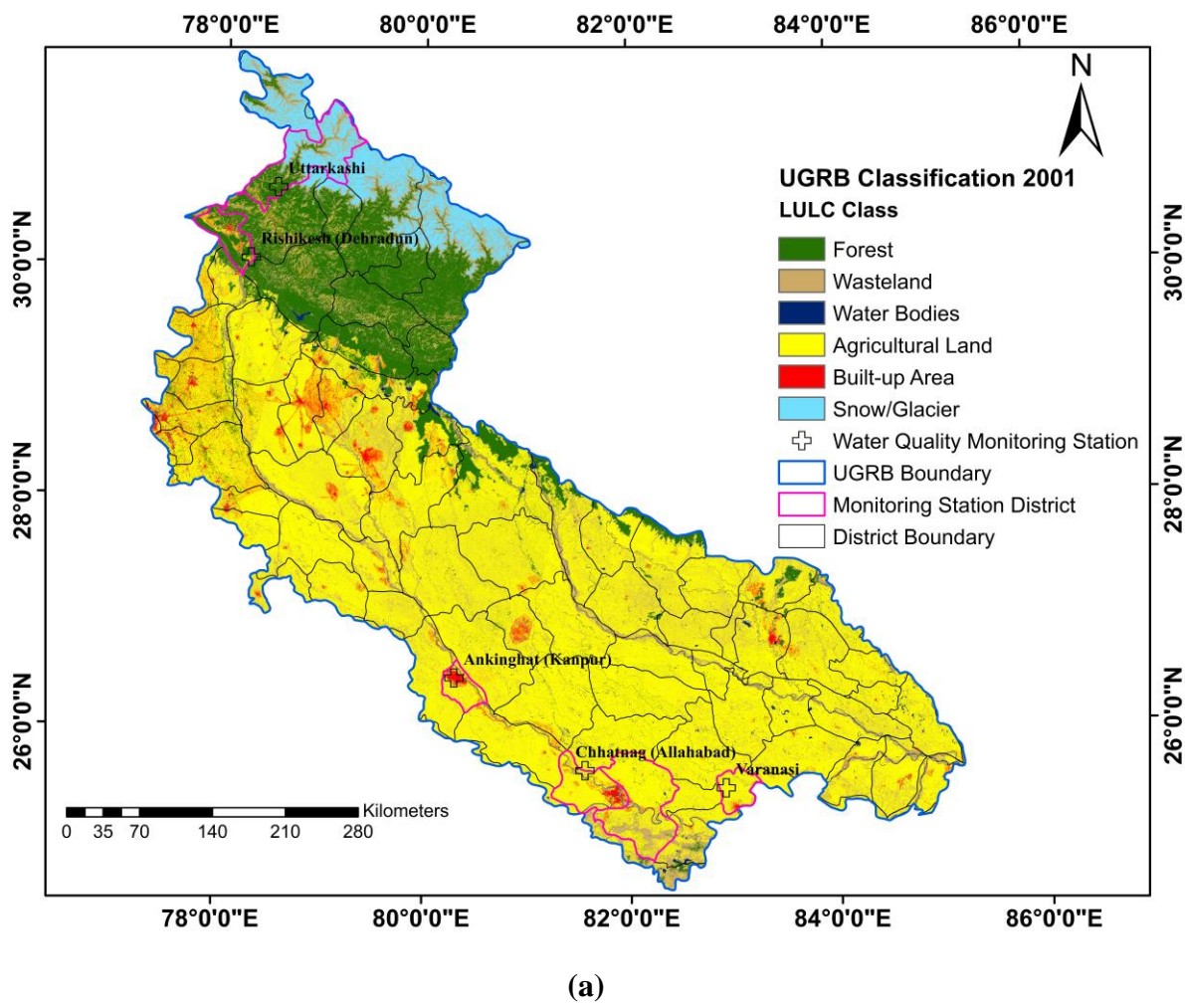


**(a)**



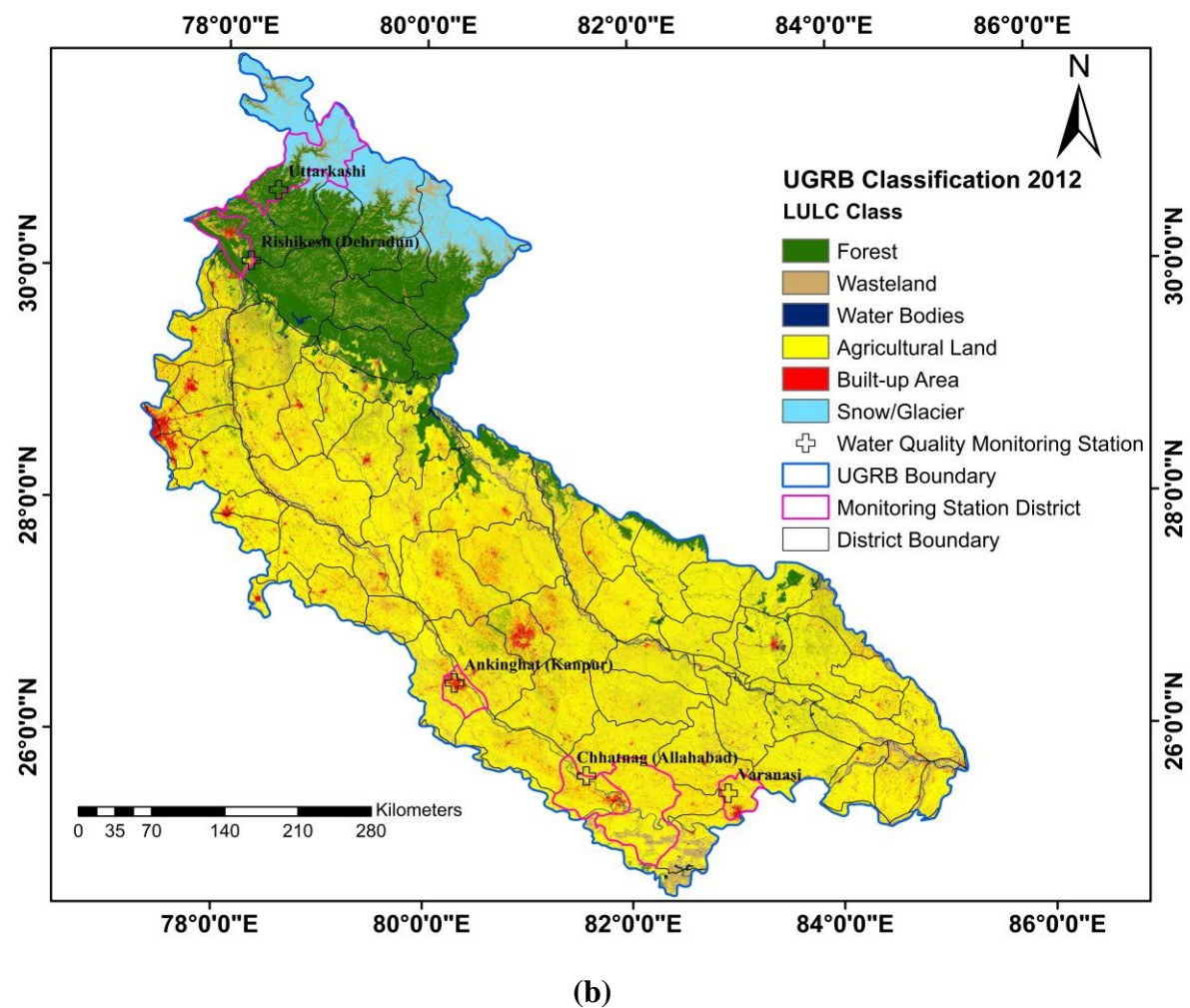


**(b)**
**Figure 4.** LULC maps of Upper Ganga River basin **(a)** LULC map of February/March 2001,
and **(b)** LULC map of February/March 2012


**(a)**

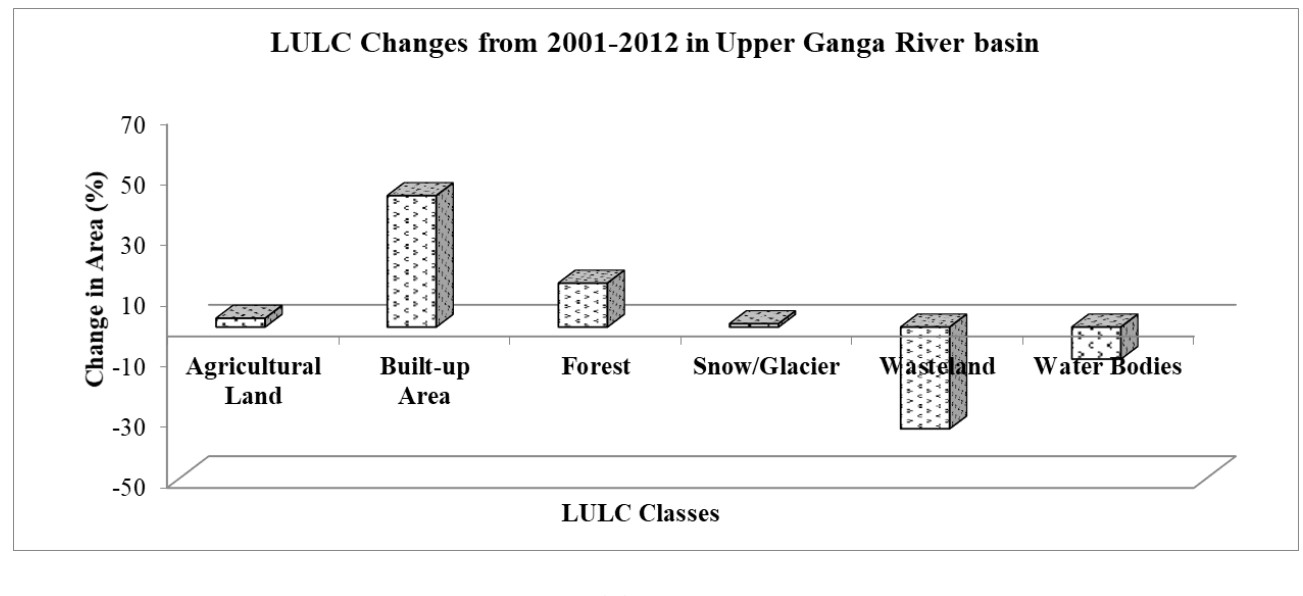

**670**

**671**                                                                 **(b)**

**672**    **Figure 5.** Graph showing LULC distribution of the years 2001-2012 **(a)** LULC area in

**673**    percentage (%) and **(b)** LULC changes from 2001-2012 in Upper Ganga River basin

**674**

**675**    Table 5 presents the change matrix, showing the conversion of one LULC class to another

**676**    between the years 2001 to 2012. Results reveal that 1.7%, 1.7%, 2.2% and 0.1% of the

**677**    wastelands in the basin area have converted to forest, agricultural land, built-up and

**678**    snow/glaciers respectively. Therefore, significant increases in these LULC classes are

**679**    observed in UGRB on the expense of wastelands, resulting in high water demand. With

**680**    increase in agricultural lands and built-up, water requirements have increased in the river

**681**    basin to meet irrigation, domestic and industrial water demands of rural and urban regions.

**682**    About 0.2% of the water bodies in the region are converted to forest during summer season

**683**    due to natural vegetation growth. Forest areas have also increased in the region due to

**684**    implementation of various Government policies for forest protection and reforestation.

**685**    Hence, slight reduction and increase in the water bodies and forest classes are observed

**686**    respectively.

**687**

**Table 5.** Change matrix showing LULC interconversion between the year 2001 and 2012 in
Upper Ganga River basin

| LULC Class | F | WL | WB | AG | BU | SG | LULC 2001 |
|---|---|---|---|---|---|---|---|
| **F** | 13.3 | 0.0 | 0.0 | 0.0 | 0.0 | 0.0 | **13.3** |
| **WL** | 1.7 | 11.4 | 0.0 | 1.7 | 2.2 | 0.1 | **17.1** |
| **WB** | 0.2 | 0.0 | 1.8 | 0.0 | 0.0 | 0.0 | **2.0** |
| **AG** | 0.0 | 0.0 | 0.0 | 58.3 | 0.0 | 0.0 | **58.3** |
| **BU** | 0.0 | 0.0 | 0.0 | 0.0 | 5.3 | 0.0 | **5.3** |
| **SG** | 0.0 | 0.0 | 0.0 | 0.0 | 0.0 | 4.0 | **4.0** |
| **LULC 2012** | **15.2** | **11.4** | **1.8** | **60.0** | **7.5** | **4.1** | **100.0** |


* Figures indicate the percentage (%) of basin area

District wise LULC change study is useful in comprehending link between LULC-water
quality at the local scale; and to identify source of pollutants at a particular monitoring
station. Table 6 presents the LULC statistics of the five districts from 2001 to 2012, where
water quality monitoring stations are located. It shows increase in built-up and agricultural
lands in all the districts whereas wastelands have decreased. Forest areas have slightly
increased in Uttarkashi and Varanasi, however they have remained unchanged in the
remaining districts. Snow/glacier class is only present in Uttarkashi district and it has slightly
increased from 2001 to 2012. Water bodies have slightly increased in all the districts except
Dehradun where it has slightly reduced. Hence, significant LULC changes are observed in
UGRB both at basin and district scales.

**Table 6.** District wise changes in LULC **(a)** Uttarkashi, **(b)** Dehradun, **(c)** Kanpur, **(d)**
Allahabad and **(e)** Varanasi
**(a)**

| **Uttarkashi** (LULC Class) | 2001% | 2012% | % Change (2001-2012) |
|---|---|---|---|
| Forest | 39.3 | 39.7 | 1.1 |

| | | | |
|---|---|---|---|
| Wasteland | 10.3 | 8.3 | -19.3 |
| Water Bodies | 1.4 | 1.5 | 4.6 |
| Agricultural Land | 0.6 | 1.4 | 122.8 |
| Built-up Area | 0.2 | 0.6 | 186.3 |
| Snow and Glacier | 48.2 | 48.6 | 0.8 |
| Total Area % | 100.0 | 100.0 | |


(**b**)

| Dehradun (LULC Class) | 2001% | 2012% | % Change (2001-2012) |
|---|---|---|---|
| Forest | 59.8 | 59.8 | 0.1 |
| Wasteland | 18.8 | 3.4 | -82.1 |
| Water Bodies | 4.8 | 4.3 | -9.8 |
| Agricultural Land | 13.5 | 20.3 | 50.6 |
| Built-up Area | 3.2 | 12.2 | 283.9 |
| Total Area % | 100.0 | 100.0 | |


(**c**)

| Kanpur (LULC Class) | 2001% | 2012% | % Change (2001-2012) |
|---|---|---|---|
| Forest | 0.3 | 0.3 | 8.7 |
| Wasteland | 23.4 | 4.7 | -79.8 |
| Water Bodies | 2.5 | 2.6 | 3.8 |
| Agricultural Land | 63.7 | 67.0 | 5.2 |
| Built-up Area | 10.1 | 25.3 | 152.1 |
| Total Area % | 100.0 | 100.0 | |


(**d**)

| Allahabad (LULC Class) | 2001% | 2012% | % Change (2001-2012) |
|---|---|---|---|
| Forest | 1.5 | 1.5 | -1.2 |
| Wasteland | 22.1 | 16.0 | -27.8 |
| Water Bodies | 3.0 | 3.1 | 1.3 |
| Agricultural Land | 70.5 | 73.4 | 4.2 |
| Built-up Area | 2.8 | 6.0 | 111.7 |

| Total Area % | 100.0 | 100.0 | |
|---|---|---|---|


**(e)**

| **Varanasi** (LULC Class) | 2001% | 2012% | % Change (2001-2012) |
|---|---|---|---|
| Forest | 0.6 | 0.7 | 24.4 |
| Wasteland | 16.8 | 6.0 | -64.5 |
| Water Bodies | 3.1 | 3.3 | 7.1 |
| Agricultural Land | 76.8 | 79.4 | 3.4 |
| Built-up Area | 2.7 | 10.5 | 291.8 |
| Total Area % | 100.0 | 100.0 | |

**5.4 Trend analysis on monthly water quality data**

From the results of trend analysis (Mann Kendall rank test) it is observed that each water quality parameter varies with time and location, hence the changes in the water quality parameters are observed in all the months (Table 7). No regular trends are observed in the water quality data, therefore, they are very site-specific. Results from statistical analyses reflect that comparatively high SD and significant changes are observed in water quality of the monsoon month (July), which is followed by pre-monsoon and post-monsoon months in decreasing order. Effect of different seasons on water quality is reported from various studies (Islam et al. 2017; Sharma and Kansal 2011; Singh and Chandna 2011). In this study, three significant seasons are identified and hence the water quality data is organized into three groups: pre-monsoon season (February-May), monsoon season (June-September) and post-monsoon season (October-January).

From each group, one representative month i.e. May, July and November month is chosen, which represents that particular season the best. It reduced the redundancy of the dataset and avoided the confusion to be created due to large insignificant dataset of varying trends that makes no sense. For e.g. SD in BOD of Kanpur station in May, July and November months are 2.01, 2.67 and 1.04 respectively. In other months, SD value of the BOD is close to the SD value of the representative months. In addition, from Table 7 it is evident that trends for BOD and Turbidity in July month are significant for almost all the stations against other water quality parameters. They are increasing over the years from 2001-2012. Pre-monsoon (May) data signifies the water quality pollution from point sources of pollution from various sewage drains and industrial effluents. In addition to the point sources of pollution, monsoon (July) data took into account the non-point source of pollution, e.g. discharge of surface runoff from urban areas

into the nearby streams during rainfall. Post-monsoon (November) data helps to understand the
water quality condition of the rivers after the rainfall is over. Therefore, further in this study,
water quality data analysis was done for the same three representative months.

**Table 7.** Trends in monthly water quality parameters from 2001 to 2012 across Upper Ganga
River basin (Z value, a Mann-Kendal statistic parameter is shown. (*), (**), (***) and +ve suffix
indicate different significance levels)

| Station | Parameter | Jan | Feb | Mar | Apr | May | Jun | Jul | Aug | Sep | Oct | Nov | Dec |
|---|---|---|---|---|---|---|---|---|---|---|---|---|---|
| Uttarkashi | BOD | -2.4 (*) | 1.3 | -2.2 (*) | 0.0 | 1.2 | -0.4 (**) | 2.8 | -1.9 (+) | -2.2 (*) | 0.0 | 1.9 (+) | 1.3 |
| | DO% | 1.2 | -1.5 | 0.5 | 0.0 | -3.3 (**) | -2.8 (**) | -2.2 (*) | -3.3 (**) | 1.4 | 0.0 | -2.6 (**) | -1.5 |
| | F | -1.9 (+) | 2.0 (*) | -3.2 (**) | 1.1 | -3.0 (**) | 0.8 | 2.0 (*) | 2.0 (*) | 1.1 | 1.9 (+) | 1.1 | -3.0 (**) |
| | Hardness | 1.3 | -2.5 (*) | 1.8 (+) | -1.1 | -1.9 (+) | -2.1 (*) | -2.5 (*) | -1.9 (+) | 1.2 | 1.8 (+) | -1.1 | -2.5 (*) |
| | pH | 2.7 (**) | -1.3 | 1.2 | -0.1 | -0.2 | 0.0 | -1.5 | -1.1 | -0.2 | -1.3 | -1.3 | -1.1 |
| | TC | - | - | - | - | - | - | - | - | - | - | - | - |
| | Turbidity | - | - | - | - | - | - | - | - | - | - | - | - |
| Rishikesh | BOD | -0.1 | 0.0 | 0.6 | 1.9 (+) | 0.4 | -2.5 (*) | 2.4 (*) | 2.0 (*) | 2.6 (*) | -1.3 | 1.3 | -0.5 |
| | DO% | -1.3 | 1.5 | 2.3 (*) | -2.3 (*) | 3.0 (**) | -2.3 (*) | 2.9 (**) | 0.6 | 0.5 | 3.4 (***) | 3.2 (**) | -3.6 (***) |
| | F | -1.0 | -0.5 | 2.2 (*) | -1.2 | 1.2 | -1.7 (+) | 1.7 (+) | 2.7 (**) | -0.8 | -0.6 | 0.0 | 2.5 (*) |
| | Hardness | 1.4 | -1.6 | 0.6 | 2.7 (**) | -2.3 (*) | 0.6 | -2.4 (*) | 1.3 | 0.0 | 3.2 (**) | -1.6 | -2.7 (**) |
| | pH | -1.6 | 0.0 | 0.0 | -0.7 | -0.9 | 0.2 | -0.2 | 1.1 | 1.9 (+) | 1.6 | -0.8 | 0.3 |
| | TC | - | - | - | - | - | - | - | - | - | - | - | - |
| | Turbidity | - | - | - | - | - | - | - | - | - | - | - | - |
| Kanpur | BOD | 2.0 (*) | 2.7 (**) | 2.6 (**) | 2.3 (*) | 3.0 (**) | 3.4 (***) | 3.4 (***) | 2.7 (**) | 1.7 (+) | 0.6 | 1.6 | 2.2 (*) |
| | DO% | -2.7 (**) | -2.0 (*) | -0.3 | -1.1 | -0.5 | -0.3 | -2.1 (*) | -0.5 | -0.1 | -0.8 | -1.0 | -1.8 (+) |

| | | | | | | | | | | | | | |
|---|---|---|---|---|---|---|---|---|---|---|---|---|---|
| | F | 1.5 | 2.0 (*) | 1.7 (+) | 1.6 | 1.2 | 2.1 (*) | 2.4 (*) | 2.2 (*) | 2.6 (**) | 2.4 (*) | 1.7 (+) | 2.0 (*) |
| | Hardness | 0.4 | 0.2 | 0.1 | 0.1 | 0.0 | 1.2 | 1.7 (+) | 0.0 | 0.0 | -0.2 | -1.0 | -1.0 |
| | pH | 0.3 | -0.2 | 0.7 | 1.9 (+) | 1.7 (+) | 0.2 | 1.2 | -0.9 | -0.3 | -1.0 | -0.4 | -1.2 |
| | TC | - | - | - | - | - | - | - | - | - | - | - | - |
| | Turbidity | 3.5 (***) | 1.7 (+) | 1.7 (+) | -0.4 | -0.2 | 0.8 | 0.8 | 1.7 (+) | -1.6 | 0.0 | 1.9 (+) | 0.3 |
| Allahabad | BOD | 0.8 | 0.2 | -1.3 | 0.3 | -0.1 | 0.2 | -1.0 | -0.1 | -0.5 | -0.1 | -0.4 | 0.0 |
| | DO% | 0.6 | -0.5 | 0.6 | 0.0 | -0.2 | 0.4 | 1.0 | 1.7 (+) | 0.7 | 1.0 | -0.3 | -0.2 |
| | F | 1.6 | 1.2 | 2.0 (*) | 2.6 (**) | 1.6 | 1.4 | 2.2 (*) | 2.2 (*) | 2.7 (*) | 1.7 (+) | 1.6 | 1.0 |
| | Hardness | -0.8 | 0.0 | -1.3 | -0.3 | 0.2 | 0.1 | -0.1 | 0.3 | -0.1 | 0.4 | 0.5 | 1.5 |
| | pH | -1.0 | -1.3 | 0.1 | -0.3 | 0.2 | 0.1 | 1.0 | 0.1 | -1.1 | -0.4 | 0.4 | 0.0 |
| | TC | -1.1 | -1.0 | -1.4 | -1.0 | -1.1 | 0.6 | -0.5 | -2.0 (*) | -1.7 (+) | -1.4 | -1.1 | -0.3 |
| | Turbidity | -0.9 | 0.2 | -0.6 | -0.2 | -1.4 | 0.9 | 0.4 | 0.6 | 0.4 | -0.3 | 0.0 | -1.4 |
| Varanasi | BOD | 2.4 (*) | 1.5 | 1.1 | 1.4 | 2.2 (*) | 2.8 (**) | 2.7 (**) | 1.9 (+) | 2.4 (*) | 2.9 (**) | 2.6 (**) | 3.0 (**) |
| | DO% | 1.2 | 1.4 | 2.2 (*) | 2.3 (*) | 1.7 (+) | 0.8 | 1.5 | 2.5 (*) | 3.2 (**) | 3.3 (***) | 2.5 (*) | 2.5 (*) |
| | F | 2.5 (*) | 2.1 (*) | 2.4 (*) | 2.4 (*) | 1.6 | 1.8 (+) | 2.1 (*) | 2.1 (*) | 3.0 (**) | 2.2 (*) | 1.2 | 2.2 (*) |
| | Hardness | -0.3 | -0.3 | 0.0 | 0.1 | -0.5 | -0.7 | -0.5 | 0.1 | 0.3 | 0.8 | 0.3 | 1.9 (+) |
| | pH | 0.0 | 0.0 | 1.9 (+) | 1.5 | 0.4 | 0.2 | 0.4 | 0.2 | 1.8 (+) | 0.4 | 0.6 | 0.2 |
| | TC | 0.8 | 0.6 | 0.8 | 0.6 | 0.3 | -0.1 | 0.5 | 0.9 | 1.0 | 1.4 | 1.4 | 1.4 |
| | Turbidity | -0.5 | 0.0 | 0.0 | -0.2 | -0.6 | -1.8 (+) | -0.9 | 0.9 | 0.0 | -1.4 | 0.2 | -0.2 |


*** trend at α = 0.001 level of significance; ** trend at α = 0.01 level of significance; * trend at
α = 0.05 level of significance; + trend at α = 0.1 level of significance; If there is no sign after
values in the table then, the significance level is greater than 0.1 (Amnell et al. 2002).

**5.5 State of the population growth-LULC transformations-water quality nexus in UGRB**
In this section, the association between the three components population growth-LULC
transformations-water quality are established. Seasonal water quality parameter values for
UGRB over the periods of 2001-2012 are presented in Table 8. Their respective IPI values and
OIP for each monitoring station are illustrated in Table 9. In UGRB the population increase in
both rural and urban areas have resulted into significant changes in LULC distribution. Increase
in PGR of 20.45% in the complete basin has resulted in 43.4% and 2.9% increase in urban and
rural areas respectively. Therefore, this river basin is urbanizing gradually with increase in
industrial operations. Urbanization, industrialization and intense agricultural activities have
caused water quality degradation between the periods of 2001-2012. Nearly all the parameters
are relatively higher in the July month, which is rainy season. Hence, their subsequent IPI values
and resulting OIP are also high in this month. Hardness $CaCO_3$ and pH values are higher in
monsoon month as bicarbonates, hydroxides and phosphates from rock weathering are
transported to the river water by surface runoff. Turbidity is also high due to addition of organic
matter from land surfaces to the nearby stream through surface runoff. F is introduced into the
river by surface runoff carrying F from industrial regions. High DO% values are attributed to
increased diffusion of Oxygen into the water during increased stream flow caused by storm
events. Increase in BOD and Total Coliform bacteria is a result of increased transportation of
municipal sewage containing organic matter and various strains of Coliform bacteria. Similar
results were reported from the studies done by various researchers (Attua et al. 2014; Chapman
1992; Hellar-Kihampa et al. 2013; Jain et al. 2006).

**Table 8.** Water quality parameters across Upper Ganga River basin for pre-monsoon, monsoon
and post-monsoon seasons over periods of 2001-2012
**(i)**

| Parameters | Water Quality Monitoring Stations | | | | | | | | | | | | | | |
|---|---|---|---|---|---|---|---|---|---|---|---|---|---|---|---|
| (Year 2001) | Uttarkashi | | | Rishikesh | | | Kanpur | | | Allahabad | | | Varanasi | | |
| | May | Jul | Nov | May | Jul | Nov | May | Jul | Nov | May | Jul | Nov | May | Jul | Nov |
| BOD | 1.1 | 1.1 | 1.1 | 1.1 | 1.0 | 1.1 | 2.8 | 1.7 | 2.4 | 4.0 | 4.2 | 3.7 | 2.5 | 2.2 | 1.8 |
| DO% | 88 | 104 | 89 | 71 | 60 | 64 | 89 | 96 | 93 | 92 | 84 | 95 | 90 | 92 | 85 |
| F | 0.19 | 0.04 | 0.22 | 0.23 | 0.16 | 0.26 | 0.61 | 0.21 | 0.34 | 0.09 | 0.50 | 0.51 | 0.3 | 0.05 | 0.51 |
| Hardness $CaCO_3$ | 65 | 60 | 68 | 76 | 67 | 74 | 99 | 78 | 86 | 95 | 194 | 159 | 99 | 176 | 142 |
| pH | 8.1 | 8.1 | 8.1 | 8.1 | 8.1 | 8.1 | 8.0 | 8.3 | 8.1 | 8.2 | 8.3 | 8.2 | 8.2 | 8.4 | 8.2 |
| Total Coliform | - | - | - | - | - | - | - | - | - | 3000 | 6200 | 6500 | 5100 | 5300 | 2400 |
| Turbidity | - | - | - | - | - | - | 2.0 | 3.1 | 2.3 | 0.1 | 0.2 | 0.1 | 0.1 | 0.1 | 0.1 |


**(ii)**

| Parameters | Water Quality Monitoring Stations | | | | | | | | | | | | | | |
|---|---|---|---|---|---|---|---|---|---|---|---|---|---|---|---|
| (Year 2012) | Uttarkashi | | | Rishikesh | | | Kanpur | | | Allahabad | | | Varanasi | | |
| | May | Jul | Nov | May | Jul | Nov | May | Jul | Nov | May | Jul | Nov | May | Jul | Nov |
| BOD | 1.1 | 1.2 | 1.0 | 1.0 | 1.2 | 1.2 | 7.0 | 10.0 | 4.0 | 2.9 | 3.2 | 2.4 | 3.0 | 3.9 | 2.9 |
| DO% | 73 | 64 | 73 | 81 | 75 | 77 | 86 | 75 | 90 | 85 | 108 | 98 | 101 | 98 | 98 |
| F | 0.45 | 0.26 | 0.44 | 0.09 | 0.19 | 0.06 | 0.70 | 0.80 | 0.51 | 0.51 | 0.67 | 0.56 | 0.57 | 0.54 | 0.52 |
| Hardness $CaCO_3$ | 45 | 24 | 34 | 33 | 23 | 56 | 110 | 102 | 90 | 97 | 85 | 92 | 89 | 75 | 81 |
| pH | 7.8 | 7.7 | 7.6 | 7.8 | 8.0 | 7.8 | 8.7 | 8.4 | 8.1 | 8.2 | 8.5 | 8.2 | 8.7 | 8.4 | 8.7 |
| Total Coliform | - | - | - | - | - | - | - | - | - | 5200 | 5800 | 4600 | 5600 | 7300 | 4700 |
| Turbidity | - | - | - | - | - | - | 4.0 | 6.0 | 5.4 | 0.1 | 0.5 | 0.1 | 0.1 | 0.2 | 0.1 |


*Units: BOD=mg/L; DO%=%; F= mg/L; Hardness $CaCO_3$= mg/L; pH=No unit; Total
Coliform=MPN; Turbidity=NTU





**Table 9.** Individual parameter indices (IPIs) and overall indices of pollution (OIPs) computed at various water quality monitoring stations of Upper Ganga River basin over periods of 2001 and 2012 for pre-monsoon, monsoon and post-monsoon seasons

**(i)**

| Parameters | Water Quality Monitoring Stations | | | | | | | | | | | | | | |
| | Uttarkashi | | | Rishikesh | | | Kanpur | | | Allahabad | | | Varanasi | | |
| | May | Jul | Nov | May | Jul | Nov | May | Jul | Nov | May | Jul | Nov | May | Jul | Nov |
| *BOD* | 1.00 | 1.00 | 1.00 | 1.00 | 1.00 | 1.00 | **2.87** | **2.40** | **2.60** | **2.67** | **2.80** | **2.47** | 1.67 | 1.47 | 1.20 |
| DO% | 1.33 | 1.28 | 1.27 | **2.49** | **3.24** | **2.97** | 1.27 | 0.79 | 0.99 | 1.06 | 1.61 | 0.86 | 1.20 | 1.06 | 1.54 |
| F | 1.00 | 1.00 | 1.00 | 1.00 | 1.00 | 1.00 | 1.00 | 1.00 | 1.00 | 1.00 | 1.00 | 1.00 | 1.00 | 1.00 | 1.00 |
| **Hardness CaCO$_3$** | 1.00 | 1.00 | 1.00 | 1.78 | 1.00 | 1.00 | 1.99 | 1.80 | 1.87 | 1.95 | **3.16** | **2.66** | 1.99 | **2.89** | **2.45** |
| pH | **2.76** | **2.76** | **2.76** | **2.76** | **2.76** | **2.76** | 2.52 | **3.33** | **2.76** | **3.03** | **3.33** | **3.03** | **3.03** | **3.65** | **3.03** |
| **Total Coliform** | - | - | - | - | - | - | - | - | - | **3.43** | **4.60** | **4.98** | **4.02** | **3.48** | **3.21** |
| Turbidity | - | - | - | - | - | - | 1.00 | 1.00 | 1.00 | 1.00 | 1.00 | 1.00 | 1.00 | 1.00 | 1.00 |
| **OIP (2001)** | **1.42** | **1.41** | **1.41** | **1.81** | **1.80** | **1.75** | *2.61* | *2.49* | *2.54* | *2.02* | *2.50* | *2.29* | *1.99* | *2.08* | *1.92* |

**(ii)**

| Parameters | Water Quality Monitoring Stations | | | | | | | | | | | | | | |
| | Uttarkashi | | | Rishikesh | | | Kanpur | | | Allahabad | | | Varanasi | | |
| | May | Jul | Nov | May | Jul | Nov | May | Jul | Nov | May | Jul | Nov | May | Jul | Nov |
| **BOD** | 1.00 | 1.00 | 1.00 | 1.00 | 1.00 | 1.00 | **4.67** | **6.67** | **2.67** | 1.93 | **2.13** | 1.60 | **2.00** | **2.60** | 1.93 |
| **DO%** | **2.36** | **2.97** | **2.36** | 1.81 | **2.22** | **2.08** | 1.47 | **2.22** | 1.20 | 1.54 | 1.49 | 0.65 | 1.13 | 0.65 | 0.65 |
| F | 1.00 | 1.00 | 1.00 | 1.00 | 1.00 | 1.00 | 1.00 | 1.00 | 1.00 | 1.00 | 1.00 | 1.00 | 1.00 | 1.00 | 1.00 |
| **Hardness CaCO$_3$** | 1.00 | 1.00 | 1.00 | 1.00 | 1.00 | 1.00 | **2.10** | **2.02** | **2.91** | 1.97 | 1.86 | 1.92 | 1.90 | 1.00 | 1.82 |
| pH | **2.09** | 1.91 | 1.74 | **2.09** | **2.52** | **2.09** | **4.81** | **3.65** | **2.76** | **3.03** | **4.00** | **3.03** | **4.81** | **3.65** | **4.81** |
| **Total Coliform** | - | - | - | - | - | - | - | - | - | **4.05** | **4.11** | **3.90** | **4.14** | **5.97** | **3.93** |
| Turbidity | - | - | - | - | - | - | 1.00 | 1.20 | 1.08 | 1.00 | 1.00 | 1.00 | 1.00 | 1.00 | 1.00 |
| **OIP (2012)** | **1.49** | **1.58** | **1.42** | **1.38** | **1.55** | **1.44** | *2.51* | *2.79* | *2.77* | *2.07* | *2.23* | *1.87* | *2.28* | *2.27* | *2.16* |

* Bold IPI and Italic OIP values are significant

**(a)**

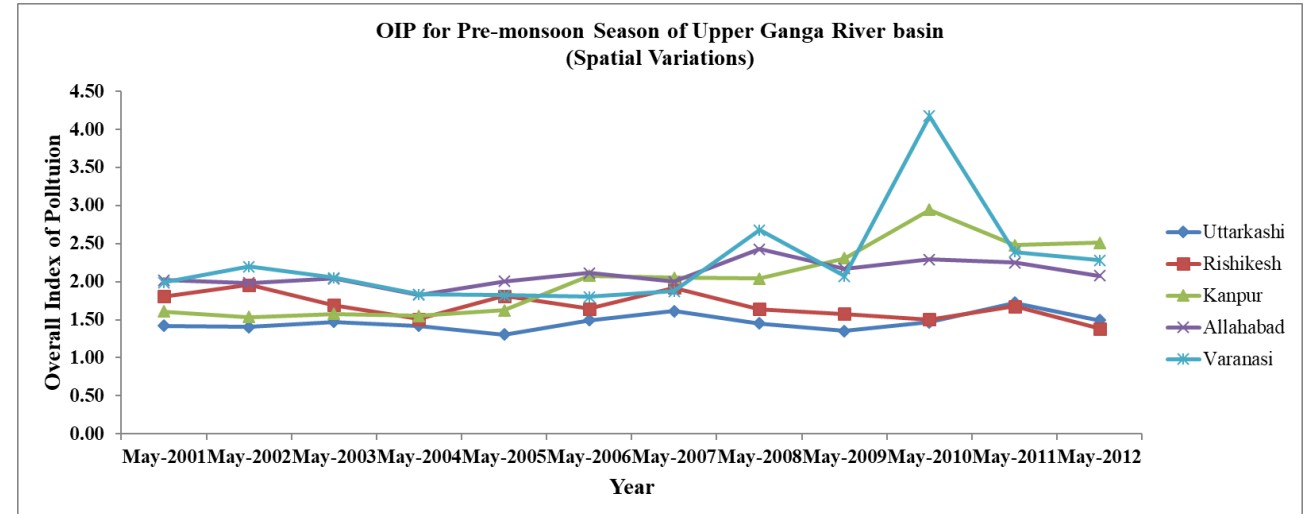



**(b)**

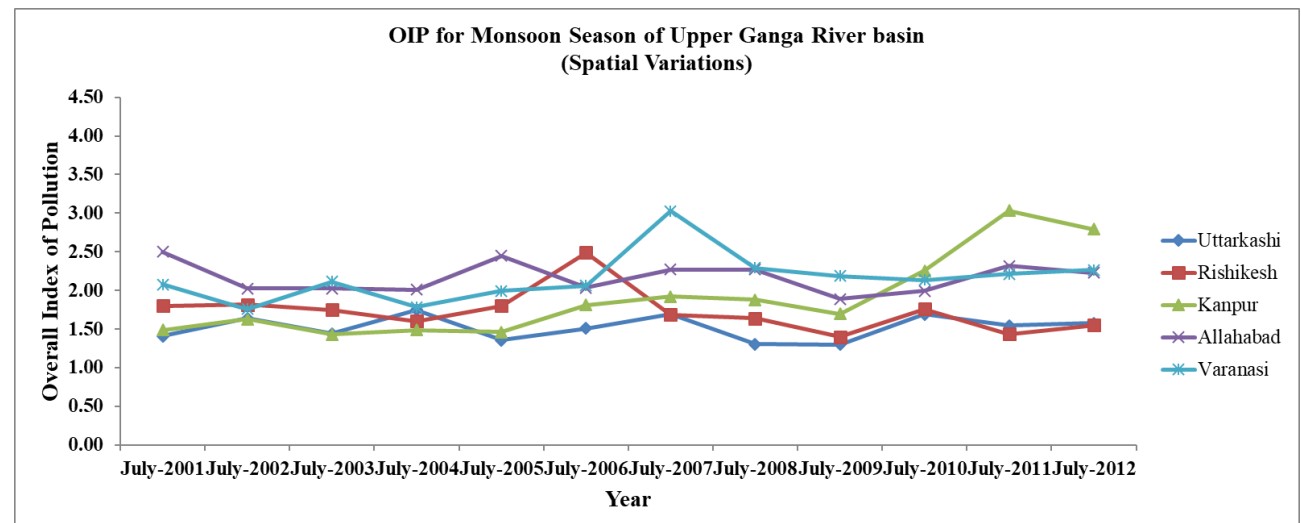







**(c)**

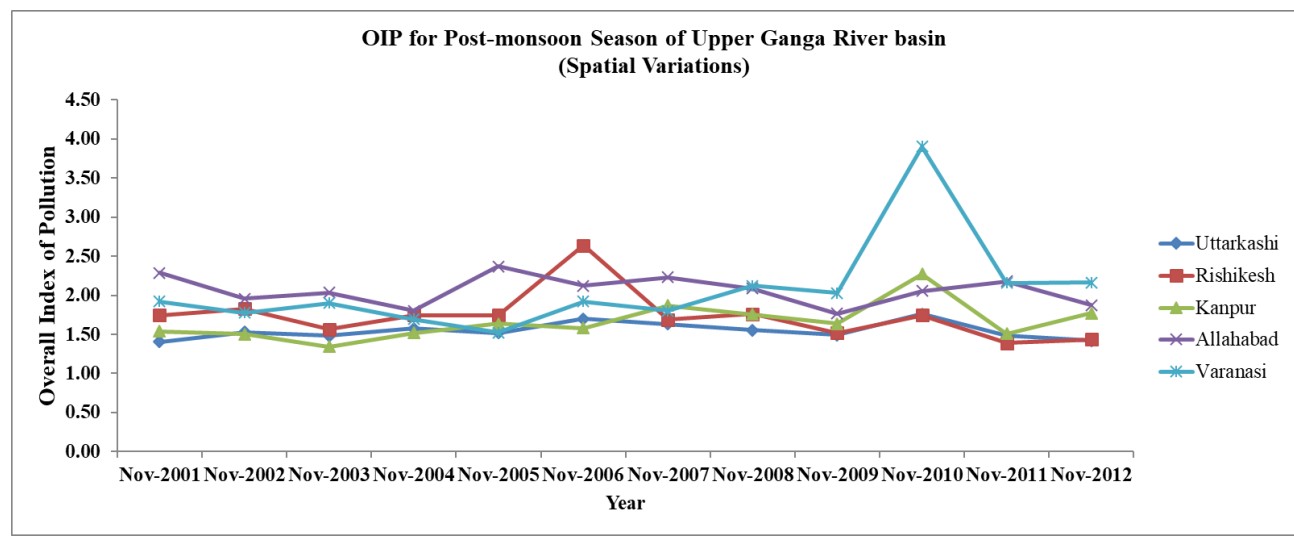


**Figure 6.** Spatial variations in the overall indices of pollution (OIP) of Upper Ganga River basin
from 2001-2012 for **(a)** Pre-monsoon period **(b)** Monsoon period, **(c)** Post-monsoon period

In UGRB, the population growth and LULC transformations are lower in the upper reaches
therefore the water quality of the monitoring stations located in this region (Uttarkashi and
Rishikesh) has remained in acceptable class range (OIP: 1.38-1.58) from 2001-2012. Conversely
in the lower reaches, the water quality has deteriorated from acceptable class to slightly polluted
class (OIP: 1.87-2.79) at the monitoring stations (Ankinghat, Chhatnag and Varanasi) due to
increasing pollutants in the river water from urban, agriculture and industrial sectors (Fig. 6 and
Table 9). Further, explanation on the connection between population growth-LULC
transformations-water quality in UGRB is given at the district or local scale in Section 5.6.

**5.6 State of the population growth-LULC transformations-water quality nexus in the**
**districts of UGRB**
Besides analysis at complete river basin level, the district level studies are also important. Each
district has different topography, climate, population and LULC distribution. Therefore, the
water management strategies in these districts should be based on the sources of pollutants and
the health status of the river. Spatio-temporal variations in the water quality of the UGRB are
studied using OIPs for three different seasons viz. pre-monsoon (May), monsoon (July) and post-
monsoon (November) from the year 2001-2012. Rainfall amount, duration and intensity are
important drivers affecting surface water quality parameters of a water body primarily during
monsoon and post-monsoon seasons. For e.g. OIP at Ankinghat (Kanpur) has slightly increased
from 2.51 in pre-monsoon season to 2.79 in monsoon season in the year 2012. In post-monsoon
season, it has further decreased to 2.77. Similarly, at Chhatnag (Allahabad) station higher OIP
(2.23) is noticed in monsoon season than other two stations in the year 2012 (Table 9). Other
factors such as type of LULC, type of soils, amount and type of waste generation, treatment
facilities, etc. also affect the water quality. At Varanasi station, OIP values are higher in pre-
monsoon season (2.28) than other two seasons in 2012. Reduced values in monsoon season are
probably due to relatively lower rainfall at this station. It indicates high influence of
anthropogenic activities on the river water than natural drivers such as rainfall. But at the same
station, in the year 2001 the OIP values were higher in monsoon season (2.08) than other
remaining seasons. Hence, high spatio-temporal variations are observed in the water quality
status of the river (Table 9). Water quality parameters viz. Hardness $CaCO_3$, F, pH and Turbidity
generally increase during post-monsoon season due to addition of various pollutants and
sediments in the river water during monsoon period.

Water quality monitoring stations of Uttarkashi (PGR=11.9%) and Rishikesh (Dehradun
PGR=32.3%) are located in the foothills of Himalaya with relatively low gross population in
small towns. These stations are least influenced by human intervention among all the stations.
They are mainly influenced from the generation of silts (due to steep hilly slopes) and climatic
factor such as rainfall. For example, IPI for pH in 2001 remained 2.76 in both the stations. In
2012 the pH ranged between 1.74 (post-monsoon season) to 2.09 (pre-monsoon season) at
Uttarkashi station. At Rishikesh station it ranged between 2.09 (pre and post-monsoon season) to
2.52 (monsoon season) which is slightly better than the IPI values in 2001. Therefore, all the
water quality parameters at these stations are in acceptable range with no significant variations in
the IPI values of the parameters over time. As the Ganga River descends down to Gangetic
Plains, a large number of tributaries join river Ganga. One of those, river Yamuna that passes
from metropolitan city of New Delhi and many other Class-I cities (population>1,00,000) joins
river Ganga at Allahabad. It carries a large amount of untreated pollutant load from both
municipal and industrial areas of these cities on its way and adds to the river Ganga. During
rainfall, toxic urban runoff is discharged to the river directly or through storm water drains.
Similarly, water pollution at Kanpur is caused by urban domestic wastes and industries, mainly
tanneries. At Varanasi river water again gets affected by municipal and industrial discharges into
the river. Varanasi being the last monitoring station collects pollutants from all the above cities,
hence it is identified as the most severely polluted station in UGRB, which keeps varying with
the time. In 2001, Allahabad is the most polluted station followed by Varanasi and Kanpur.
However, in 2012, Kanpur is the most polluted station followed by Varanasi and Allahabad
indicating LULC changes. The water quality remained in the acceptable to slightly polluted class
range.

Total population of all the three cities is very high and Kanpur has the highest population (6,377,452) amongst them. Varanasi has the highest population density in the region. Similarly, Allahabad has a PGR of 20.6% between 2001-2011. These cities are the biggest centres of commercial activities in UGRB. The main industry types in Allahabad district are glass, wire products, battery, etc. whereas Varanasi consists of textile, printing, electrical machinery related industries. In the lower reaches of the Ganga River, major industrialization has occurred in and around Kanpur. Tanneries are the major types of industries in Kanpur; majority of them are located in the Jajmau area which is close to River Ganga. The wastewater generated from various tanning operations, viz. soaking, liming, deliming and tanning, etc. result in increased levels of organic loading, salinity and specific pollutants such as Sulphide and Chromium. These are very toxic pollutants and affect the parameters, viz. BOD, Hardness $CaCO_3$, pH and Turbidity (Rajeswari 2015). Hence, due to wastewater from tanneries and municipal discharges, high IPI values of Hardness $CaCO_3$ (2.10) and pH (4.81) are observed for Kanpur station in 2012. IPI values of Hardness $CaCO_3$ (1.90) and pH (4.81) at Varanasi station is just lower to Kanpur and it is followed by water quality of Allahabad which showed close IPI values of 1.97 and 4.00, respectively. These cities do not have tanneries but their urban sewage and industrial effluents affect water quality of the river.

889

Other than tanneries, agro-based, textile, paper, mineral, metal and furniture based industries are also present. Unnao is other industrial town located close to Kanpur. Large amount of municipal sewage generated in the urban residential areas and industrial effluents are discharged into the water. In total, 6087 MLD of wastewater is discharged into the Ganga River. Out of the complete

river basin, six sub-regions namely Kanpur, Unnao, Rai-Bareeilly, Allahabad, Mirazapur and
Varanasi alone discharge 3019 MLD of wastewater directly/indirectly into the river. Particularly,
cities of Kanpur, Allahabad and Varanasi contribute about 598.19 MLD, 293.5 MLD and 410.79
MLD of wastewater into the river respectively (CPCB 2013; NRSC 2014). Municipal sewage
water is characterized by high BOD and Total Coliform bacteria count. Table 9 illustrates a very
high IPI value in the BOD of Kanpur (6.67), Allahabad (2.13) and Varanasi (2.60) in the year
2012. It has increased from 2001 to 2012. Similarly in the year 2012, IPI of Total Coliform
bacteria count is found in the range of minimum 3.90 (Allahabad) to 5.97 (Varanasi). It falls in
the class of slightly polluted to polluted. F, pH and Turbidity are the factors mainly affected by
natural drivers. IPI is within acceptable to slightly polluted range in all the three stations in 2012.
F and Turbidity have remained in excellent and acceptable classes over the years. Various other
studies have reported that the water quality of Ganga River near Kanpur, Allahabad and Varanasi
cities is highly polluted (Gowd et al. 2010; Rai et al. 2010; Sharma et al. 2014). Rapid
urbanization and industrialization has highly affected the water quality of River Ganga in these
districts.

**5.7 Relationship between LULC and water quality (OIP)**
Pearson's correlation analysis between OIP and different LULC classes in UGRB helped in
studying strength of association between these variables (Table 10). In all the three seasons of
the year 2001, wasteland, built-up and agricultural lands are positively correlated showing
significant relationship (moderate to strong association) with OIP. Water bodies have shown
very weak positive correlation whereas moderate to strong negative correlation is observed with
forest class. Due to change in the LULC distribution and water quality parameters between 2001-
2012, variations are observed in the strength of association in the year 2012. In this
year, OIP showed very strong negative and a very weak negative correlationship with forest and water
bodies classes respectively. A very strong positive association is observed with agricultural
lands. Moderate to strong positive correlationship is observed with built-up class. Association of
OIP with wasteland is in the broad range of very weak positive to very weak negative
correlation.

**Table 10.** Pearson's correlation coefficients relating LULC to water quality (OIP) in the Upper
Ganga River basin (Pre-monsoon, Monsoon and Post-monsoon seasons of 2001 and 2012)

| Stations | OIP Pre-monsoon (2001) | F% | WL% | WB% | AG% | BU% |
|---|---|---|---|---|---|---|
| Uttarkashi | 1.42 | 39.3 | 10.3 | 1.4 | 0.6 | 0.2 |
| Rishikesh | 1.81 | 59.8 | 18.8 | 4.8 | 13.5 | 3.2 |
| Kanpur | 2.61 | 0.3 | 23.4 | 2.5 | 63.7 | 10.1 |
| Allahabad | 2.02 | 1.5 | 22.1 | 3.0 | 70.5 | 2.8 |
| Varanasi | 1.99 | 0.6 | 16.8 | 3.1 | 76.8 | 2.7 |
| Pearson's correlation coefficients | | -0.65 | 0.87 | 0.12 | 0.71 | 0.95 |


| Stations | OIP Monsoon (2001) | F% | WL% | WB% | AG% | BU% |
|---|---|---|---|---|---|---|
| Uttarkashi | 1.41 | 39.3 | 10.3 | 1.4 | 0.6 | 0.2 |
| Rishikesh | 1.80 | 59.8 | 18.8 | 4.8 | 13.5 | 3.2 |
| Kanpur | 2.49 | 0.3 | 23.4 | 2.5 | 63.7 | 10.1 |
| Allahabad | 2.50 | 1.5 | 22.1 | 3.0 | 70.5 | 2.8 |
| Varanasi | 2.08 | 0.6 | 16.8 | 3.1 | 76.8 | 2.7 |
| Pearson's correlation coefficients | | -0.77 | 0.93 | 0.15 | 0.87 | 0.69 |


| Stations | OIP Post-monsoon (2001) | F% | WL% | WB% | AG% | BU% |
|---|---|---|---|---|---|---|

| Uttarkashi | 1.41 | 39.3 | 10.3 | 1.4 | 0.6 | 0.2 |
| Rishikesh | 1.75 | 59.8 | 18.8 | 4.8 | 13.5 | 3.2 |
| Kanpur | 2.54 | 0.3 | 23.4 | 2.5 | 63.7 | 10.1 |
| Allahabad | 2.29 | 1.5 | 22.1 | 3.0 | 70.5 | 2.8 |
| Varanasi | 1.92 | 0.6 | 16.8 | 3.1 | 76.8 | 2.7 |
| Pearson's correlation coefficients | | -0.73 | 0.93 | 0.09 | 0.78 | 0.83 |


| Stations | OIP Pre-monsoon (2012) | F% | WL% | WB% | AG% | BU% |
|---|---|---|---|---|---|---|
| Uttarkashi | 1.49 | 39.7 | 8.3 | 1.5 | 1.4 | 0.6 |
| Rishikesh | 1.38 | 59.8 | 3.4 | 4.3 | 20.3 | 12.2 |
| Kanpur | 2.51 | 0.3 | 4.7 | 2.6 | 67.0 | 25.3 |
| Allahabad | 2.07 | 1.5 | 16.0 | 3.1 | 73.4 | 6.0 |
| Varanasi | 2.28 | 0.7 | 6.0 | 3.3 | 79.4 | 10.5 |
| Pearson's correlation coefficients | | -0.94 | 0.10 | -0.09 | 0.88 | 0.63 |


| Stations | OIP Monsoon (2012) | F% | WL% | WB% | AG% | BU% |
|---|---|---|---|---|---|---|
| Uttarkashi | 1.58 | 39.7 | 8.3 | 1.5 | 1.4 | 0.6 |
| Rishikesh | 1.55 | 59.8 | 3.4 | 4.3 | 20.3 | 12.2 |
| Kanpur | 2.79 | 0.3 | 4.7 | 2.6 | 67.0 | 25.3 |
| Allahabad | 2.23 | 1.5 | 16.0 | 3.1 | 73.4 | 6.0 |
| Varanasi | 2.27 | 0.7 | 6.0 | 3.3 | 79.4 | 10.5 |
| Pearson's correlation coefficients | | -0.89 | 0.08 | -0.09 | 0.83 | 0.72 |


| Stations | OIP Post-monsoon (2012) | F% | WL% | WB% | AG% | BU% |
|---|---|---|---|---|---|---|
| Uttarkashi | 1.42 | 39.7 | 8.3 | 1.5 | 1.4 | 0.6 |
| Rishikesh | 1.44 | 59.8 | 3.4 | 4.3 | 20.3 | 12.2 |
| Kanpur | 2.77 | 0.3 | 4.7 | 2.6 | 67.0 | 25.3 |
| Allahabad | 1.87 | 1.5 | 16.0 | 3.1 | 73.4 | 6.0 |
| Varanasi | 2.16 | 0.7 | 6.0 | 3.3 | 79.4 | 10.5 |
| Pearson's correlation coefficients | | -0.79 | -0.14 | -0.07 | 0.75 | 0.82 |


This study found that increase in forest cover can decrease OIP due to increased aeration of
flowing river water. High sediment load, generally from surface runoff causes the increase in
turbidity. Forest areas control turbidity, Hardness $CaCO_3$ and pH parameters by acting as a buffer
against these parameters. Similarly, increase in the water bodies decrease OIP by diluting the
pollutants with excess water, thus improving the water quality. In UGRB, increase in OIP i.e.
deterioration of water quality is observed with increase in the agricultural lands and built-up due
to introduction of pollutants from various agro-chemicals, municipal sewage, industrial effluents
and other types of organic matter. These lower the DO% level and increase BOD parameter.
Correlation between wasteland and OIP are not much significant. Another study done by Attua et
al. (2014) reported similar results for the study conducted on African rivers. Multiple linear
regression analysis can efficiently predict the OIP using one or combination of LULC classes
(Table 11). OIP of 2001 could be predicted by the combined coverage area of forest, wasteland,
agricultural land and built-up area (adjusted $R^2$=0.94) whereas OIP of 2012 by forest,
agricultural land and built-up area (adjusted $R^2$=0.95). High $R^2$ and adjusted $R^2$ values in both
the years showed strong relationship between OIP and LULC classes of the respective models.
However, these relationships may vary for different regions or time periods.

**Table 11.** Multiple linear regression models for OIP and LULC classes in the Upper Ganga
River basin

| Year | Independent variable | Regression model equation | $R^2$ | Adjusted $R^2$ |
|---|---|---|---|---|
| OIP (2001) | Forest, Wasteland, Agricultural Land and Built-up area | OIP= 1.1354 - 0.6331 F + 5.08 WL - 0.0828 AG + 2.7425 BU | 0.94 | 0.94 |

| OIP (2012) | Forest, Agricultural Land and Built-up area | OIP = 2.1266 - 1.6296 F - 0.2756 AG + 2.9894 BU | 0.96 | 0.95 |
|---|---|---|---|---|


## 6. Summary and conclusions


Upper Ganga River basin is suffering from chronic water shortages since past few decades.
Population growth is the primary driver behind gradual urbanization and industrialization in this
region. In addition, infrastructure development activities and agriculture have also intensified.
Hence, the natural resources of UGRB are over-exploited. Sustainable water resources planning
and management by policy makers and planners need understanding of nexus between
components of population growth-LULC transformations-water quality at both regional and local
scale. 20.45% increase in PGR leads to 43.4% increase in built-up. It was identified as most
dynamic LULC class in the region followed by wasteland. Mann-Kendall rank test revealed that
water quality parameters are highly variable in time and space with no significant trends. Even
though gross rural population is much higher in the lower reaches of the river basin, but the PGR
is higher in the urban population of upper reaches. The water quality of majority of the stations
was most degradable in monsoon season. Water quality of upper reaches (Uttarkashi and
Rishikesh) remained in excellent to acceptable (1.38-1.81) class from 2001-2012 whereas it
changed from acceptable to slightly polluted class (1.87-2.79) in lower reaches (Kanpur,
Allahabad and Varanasi). In UGRB, BOD, DO% and Total Coliform are the parameters most
influenced by anthropogenic activities. Conversely, the remaining parameters viz. pH, F,
Hardness $CaCO_3$ and Turbidity are mainly influenced by climatic factors. The highest increase in
built-up of 291.8% observed in the Varanasi district is directly related to the highest deterioration
of water quality in UGRB. But Allahabad and Kanpur are identified as most polluted stations in
2001 and 2012 respectively. Sewage, industrial effluents and runoff from urban/rural areas

introduce pollutants at these stations. Future population growth and LULC changes in UGRB
may further jeopardize their nexus with water. Forests and water bodies are negatively correlated
with OIP. However, built-up and agricultural lands are positively correlated. Wasteland is not
significantly correlated to OIP. Multiple linear regression models developed for UGRB could
successfully predict OIP (water quality) using LULC classes. The future scope of this study
comprises the understanding of hydro-ecological response of the water quality changes across
the river basin. The following recommendations are made for judicious regulation and control of
water quality pollution in UGRB: (a) control of deforestation and encouraging afforestation; (b)
efficient town planning for better LULC distribution in the river basin; (c) reduction in the use of
agro-chemicals in the fields (use of organic alternatives); (d) proper waste disposal and
management system; (e) strategies to control runoff from fields (construction of bunds/canals);
and (f) spreading water pollution awareness and strict policies on pollution control.

**Competing interests**: The authors declare no conflict of interest.

**Acknowledgements:** The authors thankfully acknowledge all the support provided by
Department of Civil Engineering, Indian Institute of Technology Roorkee, Uttarakhand, India.
We would like to express our gratitude to the Census Department (Government of India) and
Central Water Commission (CWC), New Delhi for providing census and water quality datasets
respectively. We are also grateful to anonymous reviewers and editors for their valuable
suggestions that helped to improve the manuscript further.

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
