# Peer review of "Population Growth – Land Use Land Cover Transformations – Water"

_Hydrology and Earth System Sciences, 2017_

## Referee Comment (RC1) · Anonymous Referee #1 · 21 Nov 2017

The paper sets out to investigate the relationships between land use/cover, population growth and water quality for a large river basin. Remotely sensed data is used in combination with population census and water quality measurements to analyse correlations between the data available. The authors have attempted to identify causal links between the patterns of change seen between 2001 and 2012. The datasets used are appropriate but there are some technical issues to be addressed as highlighted below. General observations are made with regard to the potential sources of pollutants which given the scale of investigation is probably appropriate although it would have been better if some clear cut examples could be presented that show how a specific change in land use/cover has changed the level of pollutants measured. The OIP classification is helpful in the categorisation of water quality at this scale of analysis and matches the scales of the available population and land cover/use data.

Specific comments: 1. Better clarity is required in the description of the remote sensing methodology. Section 3.2 starts by describing the validation points used for the accuracy assessment. The sampling design used to identify these points is not described here, this information appears much later in section 5.3 and is only described as a simple random survey. The sample design must be described in section 3.2 and the selection of the sampling methodology applied justified. In particular, the use of a simple random sample must be clearly justified as this approach potentially raises a number of issues not least the potential to have a poor spatial distribution of sample points. 2. How many of the validation points were ground truthed? What was the accuracy of the validation point interpretations? 3. How many GPS survey data points were used to train the MLC? How did you account for potential autocorrelation in the training data? 4. A comment is needed that assesses the impact of the scan line corrector failure on Landsat 7 imagery from 2003 onwards with regard to the 2012 image classifications. 5. What radiometric correction was applied to ensure consistency of reflectance values across the large number of images used in your classifications? 6. You refer to 'relative geometric correction'. What is this? What algorithm was used? 7. The districts you have selected for analysis should be included on the maps of LULC (Figure 4) to give the reader of the paper the spatial context for them 8. I would recommend the addition of district specific land use change maps to help support your discussion. At present it is impossible to visually relate the pattern of land use change to the water quality and population statistics because the scale of the mapping in figure 4 is too small. 9. In place of table 4, I would present a cross tabulation table of the 2001 and 2012 LULC classes. This will clearly show the reader what has changed to what and then the gross and net changes can be shown in figure 5.

Technical Comments: 1. Repetition is a problem in several places through the text. The worst example of this is presenting results back again in the conclusions. Go through

the paper carefully and remove the repetition. 2. The English needs to be corrected throughout the paper. Please find a native English speaker to go through the paper to correct for missing words, improve the phraseology used and correct the grammar. 3. Avoid the use of superlatives e.g. 'tremendous' and 'colossal changes'. These terms cannot be substantiated and so should not be used. 4. The long list of water quality indicators is excessive. Highlight only those that could be relevant to the data available for this study and those commonly used. 5. Avoid excessive precision e.g. 238,347.74 km2. At the scale you are working expressing to the nearest km2 is appropriate. 6. Figure 1 - The inset map should be inside the map frame, the water quality monitoring station location labels conflict with the basin boundary line - change the position of the labels so that this doesn't occur, remove the underscore characters from the legend text (this also applies to figure 4). 7. The population statisitcs on page 18 should be presented as a table with the PGR statistics given for each district. Figure 3 on page 19 is repetition of the data that will be presented in the tbale so remove figure 3. 8. The information on pH on page 30 can be regarded as a known fact and so doesn't need to be explained. 9. The discussion that follows the pH description needs to be written with reference to just the set of figures showing the OIP values plotted against the stations. The other figure is effectively repetition so remove the other figure.

---

## Referee Comment (RC2) · Anonymous Referee #2 · 12 Feb 2018

General comments: The authors have a clear understanding of his topic area and have applied remote sensing and data collation techniques to answer investigate the impacts of demographic changes on water quality in the upper Ganga River Basin. However I believe considerable work is needed to bring the paper to publishable standard.

My major concerns are threefold: 1) What is it about this paper that is academically novel? Is it the application of existing methods to a new area? Or the evaluation of new methodology? The authors need to make this much clearer as the current introduction suggests that the paper aims to identify drivers; however the method takes drivers as given and the conclusions focus on the utility of the method. . . which aspect of the

research are important to academia as a whole? 2) There is an imbalance in the level of detail applied throughout. Some areas provide too much detail (I don't think the equation for growth rate needs five lines of explanation and an equation) whereas some key aspects of the methodology (e.g. radiometric correction) aren't described at all and there is little critical interegation of the results (why are they the way they are – what factors contribute and what doesn't). There is a need to work on structuring the data to make it make better sense 3) The paper needs a thorough restructure as it is repetitive, provides considerable amounts of extraneous material and doesn't clearly signpost what is relevant to read. The language would benefit from a thorough proof read by a native English speaker also.

Technical/Specific comments: 1. There is too great a level (e.g. population figures to the person on page 18. Round to the nearest 1000?) 2. Check English 3. Identify a clear research question. 4. Explain why it is novel – this is very important and does not come across well in the current draft. 5. Find a clear argument that flows throughout the paper and only select figures and data that make it easier for the reader to understand this argument. E.g. Remove superfluous data such as the city populations on page 18. 6. Section 5.1 could be summarised in a paragraph of text. I over-simplify but much of it can be covered by the following sentence: "Growth rates for urban and rural areas were calculated from official statistics (Figure 3)". Is the individual city data relevant? How does it fit to the overall argument? Would spatial/mapped data be more useful or relevant when compared with RS data? Figure 3 simply repeats statistics shown in the text. 7. The remote sensing work seems well carried out. However more detail is needed on the interpretation of the confusion matrices etc. (what is confused with what?; why?; what does this mean for the interpretation of the results?) 8. There is too much detail in some areas (e.g. full description of equation for population growth rate; detail of full mann-kendall method etc.) Only add detail like this if it is needed to help the reader understand the method, or if there is new method development else use references. Much of the Mann-Kendal work in 5.4.1 should be in methods not results.

---

## Referee Comment (RC3) · Anonymous Referee #3 · 13 Feb 2018

The study aims at analysing population and land use changes in the Upper Ganga River basin and identifying the drivers associated to the spatio-temporal variations of water quality using a water quality index. To do so, a methodology that links remotely sensed land cover data to estimate changes in time and a water quality index derived from measured data in several points of the river. The scientific approach is valid but there is no reference to previous similar studies analysing land use-water quality index relationships (e.g. https://doi.org/10.2989/16085914.2015.1077777, doi:10.1016/j.proenv.2012.01.140, https://www.ncbi.nlm.nih.gov/pubmed/27498508) nor any justification of the scientific relevance and novelty of the study. In relation to the writing, some parts of the text should be moved to different sections specially from results to methods. There is too much repetition of information along the paper. English is understandable but should be revised (e.g. word confusion, articles, etc.). A few specific comments are detailed below:

Keywords - Population or demographic change should be included

Introduction - Paragraphs could be used to better organise the ideas in the text - Lines 76-86: many water quality indices are cited, but no comment about their validity, similarities (clusters), differences, etc. is made. - Why the OIP index is good compared to other? From the methods section, I see that it is only the average of individual indices of different pollutants. Should all pollutants have the same weight according to their impact on health, removal costs…? Does the OIP propose more pollutants apart from those considered in the present study? If not, what would be the approach if there are other relevant pollutants in the studied region? - Objectives should be better organised and explained: o Not clear that it is not a continuous time analysis, but a 2 time slice analysis (2001 and 2012) with seasonal component. o Should the test of OIP as a valid index be principal in the study before it is used to extract conclusions?

Case study - Justify why the 7 selected pollutants are important and not others - Not sure if the data sources should be detailed after the methods section and be, therefore, better related to each methodological stage.

Methods - There is repetition of information. Try to avoid it by re-organising the text (section 4.1, remove details from the introduction and explain them only in this section) - Move table 1 to data section or to results, but it is not part of methods - The classification for OIP that relates the obtained values with an overall water quality status should be included in the methods section, like the IPI classification. The OIP classification is currently described in the results section (5.4.2). - Explanation about the link between LULC and water quality at the diverse stations is missing. I assume that the sub-catchments draining to the locations of the water quality stations are defined and this is used to relate the impact of spatial LULC change with water quality. I think it is worth including that as part of the methods section and also in Figure 2.

Results - No need to explain how to calculate a % change (lines 27-31). If you decide to include it anyway, it should be placed under the methods section - Figure 3 would be more useful if showing the results for the upper and lower reaches separately, instead of aggregated for the whole basin. It would support the statement in lines 20-21 which is not proved based on results - Table 4 and Figure 5 present the same results. Only one of them should be included in the paper to avoid repetition of information - Lines 95-99 should be moved to the methods section, including a description and reference about the Kappa statistics - The meaning of user's and producer's accuracy is not clear - Lines 124-158 could be moved to methods section as they describe the Mann-Kendal test - Best scenario to select representative months based on what? - In figures 6 and 7 it would be useful to depict the OPI thresholds as horizontal lines instead of as a legend - Some discussion about the conclusions and comparison with the current study should be included

Please also note the supplement to this comment:
https://www.hydrol-earth-syst-sci-discuss.net/hess-2017-384/hess-2017-384-RC3-supplement.pdf

**Supplement:**

[revised manuscript text omitted]

---

## Author Comment (AC1) · 13 Mar 2018

**Authors' Response to the Reviewer#1 Comments**

**Manuscript Ref.: hess-2017-384**

**Title: Population Growth – Land Use/Land Cover Transformations-Water Quality Nexus in Upper Ganga River Basin**

**Authors:** Anoop Kumar Shukla, Chandra Shekhar Prasad Ojha, Ana Mijic, Wouter Buytaert, Shray Pathak, Rahul Dev Garg and Satyavati Shukla

We sincerely thank the reviewers for offering their critical comments and valuable suggestions that has helped to improve the manuscript. We hereby provide our responses to the reviewer's comments and highlight the changes made in the revised manuscript based on the comments provided. These have been incorporated in the revised manuscript as follows. The point wise replies of the comments of the Reviewer#1 are given below:

**General Comments:**

The paper sets out to investigate the relationships between land use/cover, population growth and water quality for a large river basin. Remotely sensed data is used in combination with population census and water quality measurements to analyse correlations between the data available. The authors have attempted to identify causal links between the patterns of change seen between 2001 and 2012. The datasets used are appropriate but there are some technical issues to be addressed as highlighted below. General observations are made with regard to the potential sources of pollutants which given the scale of investigation is probably appropriate although it would have been better if some clear cut examples could be presented that show how a specific change in land use/cover has changed the level of pollutants measured. The OIP classification is helpful in the categorisation of water quality at this scale of analysis and matches the scales of the available population and land cover/use data.

We sincerely thank the anonymous reviewer 1 and appreciate his/her efforts in providing very useful comments for the improvements of our contribution. Present study attempts to analyse the causative connection (nexus) between the changing patterns of population, Land Use/Land Cover (LULC) and water quality of water stressed Upper Ganga River basin. We agree with the reviewer that in the previous manuscript due to the given scale of investigation and data constraints the general observations were made with regard to the potential sources of pollutants. But some clear cut examples showing how a specific change in LULC has changed the level of pollutants measured can further improve the work. We thank reviewer 1 for pointing it out as it is extremely important to bring out in this study. Therefore, as suggested further in the specific comments (Comment 8), authors have further related the changes in the LULC classes with the level of pollutants at a finer scale i.e. at district level. It is explained in Response#8 of this draft. In this study a comprehensive set of analyses are presented to assess and comprehend the current status of the population-LULC-water quality nexus in the study region, with respect to their changing patterns from 2001 to 2011. The present study is conducted at two different spatial scales i.e. (a) at river basin level (small scale), and (b) at district level (large scale) for three different seasons viz. pre-monsoon, monsoon and monsoon seasons. Such study is not done before for Upper Ganga River basin. Various methodologies are developed to study effects of LULC changes on water quality. But these methods cannot be applied directly to a region because of the differences in the data availability, climatic, topographic and LULC variations which may introduce errors. Hence,

necessary modifications are made in the existing evaluation methodology as per the requirement. And a relationship is developed between LULC and Overall Index of Pollution (OIP) using multi linear regression. This work helped to improve our contribution. OIP used in this study is a water quality index developed specifically for Indian conditions to categorize the water quality which helped to relate it with available population and LULC data. All the responses given are suitably added and necessary modifications are made throughout the revised manuscript.

**Specific comments:**

**Comment 1:** Better clarity is required in the description of the remote sensing methodology. Section 3.2 starts by describing the validation points used for the accuracy assessment. The sampling design used to identify these points is not described here; this information appears much later in section 5.3 and is only described as a simple random survey. The sample design must be described in section 3.2 and the selection of the sampling methodology applied justified. In particular, the use of a simple random sample must be clearly justified as this approach potentially raises a number of issues not least the potential to have a poor spatial distribution of sample points.

**Response 1:** We agree with the reviewer that description of the remote sensing methodology needs better clarity. Therefore, significant changes are made in Section 4 "Methodology" of the revised manuscript and it is further improved by addressing the responses for comments 1 to 6. Description of sampling design as just simple random survey is removed from section 5.3. As suggested by reviewer, description and justification of sampling design used to identify the validation points for accuracy assessment is explained and updated in section 3.2 which is given below:

To produce refined LULC maps in addition to expert judgement, ground truth (reference) data is required. In this study, a total 2014 Ground Control Points (GCPs) were collected from Global Positioning System (GPS) during the field visit and Google Earth. Out of which, 1365 were used for supervised classification of the satellite images and the remaining 649 points were used for accuracy assessment. Selection of sampling design differs in their suitability to achieve different objectives. During sampling of GCPs, selection of appropriate sampling method, sample size and measures of accuracy are very significant. Accuracy assessment using any of the probability sampling designs viz. simple random sampling, stratified random sampling, systematic sampling, and cluster sampling provides acceptable level of accuracy with not much statistical difference (Stehman and Czaplewski 1998). However, the selection of the appropriate method should be done with care considering the following elements: size as well as type of the study region; unbiased estimation of the uncertainty or variance; and characteristics of the objects being studied (Gu et al. 2012; Hashemian et al. 2004; Stephen 2009). Sample size must be enough to provide meaningful and representative basis for sampling. A good sample would sufficiently draw the properties of the objects from which it is selected. In stratified random sampling the population is divided into non-overlapping strata's or sub-populations. But this sampling method gives better results on smaller image size (Hashemian et al. 2004).

In systematic sampling, selection of samples is started at a random starting point and at fixed periodic interval but this method is not suitable for heterogeneous regions. In such regions systematic samples may not represent the properties of each class appropriately. In the cluster

sampling design, pixels are clustered in groups and then random sampling is done. Again it is not suitable for heterogeneous regions as sample size is comparatively large. In simple random sampling, selection of sample units is done so that every possible distinct sample gets the equal chance of selection. This sampling method provides comparatively better results on large image size if the rule of thumb recommended by Congalton is followed i.e. minimum 50 samples should be selected for smaller images and 75-100 samples for large Images (Congalton 1991; Hashemian et al. 2004). The present study is conducted for a large river basin (238348 km$^2$), therefore simple random sampling design was used for collecting GCPs across the study region for accuracy assessment. Simple random sampling is appropriate and can produce good results if sufficient samples are selected to ensure that all the classes are represented adequately (Congalton 1991; Foody 2002; Goncalves et al. 2007; Kiptala et al. 2013; Samal and Gedam 2015). Following the Congalton's thumb rule for better accuracy in simple random sampling, GCPs were selected in the range of 94-137 for each LULC class in proportion to their areal extent on the image (Table 7). Therefore, sufficient spatial distribution of the sampling points was achieved for each LULC class. Previously published thematic and topographic maps of Government of India (GoI) were useful to decide it.

**Comment 2:** How many of the validation points were ground truthed? What was the accuracy of the validation point interpretations?

**Response 2:** For better reliability of the results in accuracy assessment, all the 649 points used were collected using dual frequency receiver GPS (SOKKIA: Model No. S-10) which provided the horizontal accuracy in the range of 2-5 m. Further accuracy assessment was performed using these GCPs and overall accuracy of 90.14% was achieved with Kappa statistics of 0.88. Table 7 shows the accuracy assessment of the 2012 LULC map produced from Landsat Enhanced Thematic Mapper Plus (ETM+) data representing both the confusion matrix and the Kappa statistics.

**Table 7.** Accuracy assessment of the 2012 LULC map produced from Landsat Enhanced Thematic Mapper Plus (ETM+) data, representing both the confusion matrix and the Kappa statistics

| Classified Data | Reference Data | | | | | | Row Total | User's Accuracy (%) | Overall Kappa Statistics |
|---|---|---|---|---|---|---|---|---|---|
| | Agricultural Land | Built Up | Forest | Snow & Glacier | Wastelands | Water Bodies | | | |
| Agricultural Land | **128** | 0 | 6 | 0 | 3 | 0 | 137 | 93.43 | |
| Built Up | 2 | **96** | 2 | 5 | 1 | 0 | 106 | 90.57 | |
| Forest | 11 | 0 | **88** | 3 | 0 | 3 | 105 | 83.81 | |
| Snow & Glacier | 0 | 4 | 1 | **103** | 2 | 1 | 111 | 92.79 | |
| Wastelands | 1 | 2 | 0 | 7 | **82** | 2 | 94 | 87.23 | 0.88 |
| Water Bodies | 0 | 0 | 1 | 1 | 6 | **88** | 96 | 91.67 | |
| Column Total | 142 | 102 | 98 | 119 | 94 | 94 | **649** | | |
| Producer's Accuracy (%) | 90.14 | 94.12 | 89.80 | 86.55 | 87.23 | 93.62 | | | |
| Overall Classification Accuracy (%) | 90.14 | | | | | | | | |

**Comment 3:** How many GPS survey data points were used to train the MLC? How did you account for potential autocorrelation in the training data?

**Response 3:** Total 1365 GPS survey data points i.e. GCPs were used to train the Maximum Likelihood Classifier (MLC) in this study. Out of this, 830 GCPs were collected using GPS survey and remaining 535 were collected from Google Earth images. The accuracy of the GCPs collected from both GPS survey and Google Earth images were ensured before using them for MLC and the horizontal accuracy of 2-5 m was achieved. In a satellite image, when the presence, absence, or degree of pixel characteristic affects the presence, absence, or degree of the same characteristic in neighbouring pixels, it is referred to as spatial autocorrelation (Congalton 1991). Existing spatial autocorrelation in each LULC class affects the classification results, depending on properties viz. sensor resolution and landscape fragmentation. Study of spatial autocorrelation helps to select the training size and training methods for different LULC classes; selects the appropriate image scene models and spatial resolution during classification; identifies the parameters for effective classification; and helps understanding the classification errors during accuracy assessment (Chen 2004; Foody 2002). In the present study before image classification, an exploratory spectral analysis was done using histograms of each band to understand the spectral characteristics of the LULC classes. The spatial autocorrelation was analysed using semivariogram function which is measured by setting variance against variable distances. This method is efficient in measuring autocorrelation among different LULC features (Brivio et al. 1993). The estimated semivariogram was plotted to assess the spatial autocorrelation in respective bands in the satellite image. The range and shape (piecewise slope) of the semivariograms were examined visually to determine the appropriate sizes for training data, window size and sampling

interval for spatial feature extraction (Chen 2004; Xiaodong et al. 2009). A window size of 7 × 7 was chosen for sampling the training data, which gives the better classification results on Landsat ETM+ images (Wijaya et al. 2007). While developing the spectral signatures for different LULC classes, information acquired from band histograms and Euclidean distances were used for class separability. Due to higher confusion between barren land and urban areas at few places, urban areas were classified independently by masking it on the image. Uncertainties in misclassification between forest and agricultural land were reduced by adding more training samples. This significantly improved the classification accuracy (Gebremicael et al. 2017).

**Comment 4:** A comment is needed that assesses the impact of the scan line corrector failure on Landsat 7 imagery from 2003 onwards with regard to the 2012 image classifications.

**Response 4:** Due to failure in Scan Line Corrector (SLC) of the Landsat 7 satellite, all the images collected after May 31, 2003 are referred to as Landsat 7 ETM+ SLC off data. It resulted in 22% of data gap in each scene which has limited its scientific applications. However, with only 78% of data availability per scene it is some of the most radiometrically and geometrically accurate satellite dataset in the world and therefore it is still very useful for various studies (USGS 2018). A number of methods are developed to fill the data gaps in SLC off ETM+ datasets viz. Neighbourhood Similar Pixel Interpolator (NSPI), localized linear histogram match (LLHM), global linear histogram match (GLHM), Geostatistical Neighbourhood Similar Pixel Interpolator (GNSPI), and adaptive window linear histogram match (AWLHM) (Liu and Ding 2017). For heterogeneous regions, Neighbourhood Similar Pixel Interpolator (NSPI) is the simple and most effective method to interpolate the pixel values within the gaps with high accuracy (Chen et al. 2011; Gao et al. 2016; Zhu et al. 2012; Zhu and Liu 2014). The details on the NSPI algorithm are given in the research paper by Chen et al. (2011). In the present study, the Landsat ETM+ images of February/March 2012 had data gaps due to SLC off. Therefore, they were corrected using the IDL code for NSPI algorithm, which was run on software ENVI version 5.1. It is an open source algorithm developed by Chen et al. (2011) available freely on https://xiaolinzhu.weebly.com/open-source-code.html. As the study area is highly heterogeneous, this algorithm filled the data gaps in the satellite image with high accuracy i.e. Root Mean Square Error (RMSE) of 0.0367. As multiple scenes were involved in one image, necessary atmospheric, geometric and radiometric corrections were employed on the images to reduce the errors in classification. These corrections are explained in the Comment 5 and 6 of this draft. The accuracy assessment was done on the LULC map produced by 2012 image and it gave a very good accuracy i.e. overall accuracy of 90.14% and Kappa statistics of 0.88.

**Comment 5:** What radiometric correction was applied to ensure consistency of reflectance values across the large number of images used in your classifications?

**Response 5:** First the satellite images were georeferenced to a common coordinate system i.e. World Geodetic System (WGS) 1984 Universal Transverse Mercator Zone 43 N. Total 75 control points were chosen from Survey of India (SoI) toposheets of scale 1:50,000 which were used as base map for georectification. It was done for proper alignment of features in the study area. To make the two satellite images comparable a good radiometric consistency and proper geometric alignment is required. But it is difficult to achieve due differences in atmospheric conditions, satellite sensor characteristics, phonological characteristics, solar angle, and sensor observation angle on different images (Shukla et al. 2017). Image pre-processing involves atmospheric, radiometric and geometric corrections; and temporal as

well as topographic normalizations of the each satellite image to be used for LULC classification. For multi-temporal datasets image pre-processing is mandatory (Lu and Weng 2007). This study area is heterogeneous with rugged terrain and very undulating topography, therefore it is subjected to variations in the reflectance values due to temporal changes, hill shade effects, differences in viewing geometry, and solar illuminations. Hence to reduce the errors and get the actual reflectance values the Topographic and Atmospheric Correction for Airborne Imagery (ATCOR-2) algorithm available in ERDAS Imagine 2016 was used. The algorithm used Shuttle Radar Topographic Mission (SRTM) Digital Elevation Model (DEM) of 30 m spatial resolution to derive the characteristics viz. slope, aspect, shadow and skyview. This algorithm is well established and provides very good accuracy in removing haze, and in topographic and atmospheric corrections of the images (Gebremicael et al. 2017; Muriithi 2016). Finally image regression method was applied on the images to normalize the variations in the pixel brightness value due to multiple scenes taken on different dates.

**Comment 6:** You refer to 'relative geometric correction'. What is this? What algorithm was used?

**Response 6:** Raw satellite images may contain geometric distortions due to the differences in platform, atmosphere, sensor, earth and the total field of view (Shukla et al. 2017). To compare and study the changes between multiple satellite images, similar and precise alignment of the features is essential. In the absence of geometric corrections, errors may introduce in the classification and change detection results. In this study, a relative geometric correction method was used to maintain geometric consistency of both the satellite images. In the relative geometric correction method, one image is used a reference image and other is corrected with respect to it (image to image coregistration). The recent Landsat ETM+ image of 2012 was used as reference image for coregistration and the image of 2001 was georectified with respect to it. It was conducted on ERDAS Imagine 2016 image processing software. The following steps were involved in geometric correction (Gill et al 2010): Polynomial geometric model was determined and the ground control "tie" points were established across the images. It was followed by computation of geometric transformation parameters which provides the error analysis and describes the accuracy of the correction. Then Nearest Neighbour resampling method was used to populate new output grid of georectified image. The geometrically rectified images must have Root Mean Square Error (RMSE) less than 0.5. This is the criteria often used for geometric corrections of the satellite images which ensure good accuracy (Samal and Gedam 2015).

**Comment 7:** The districts you have selected for analysis should be included on the maps of LULC (Figure 4) to give the reader of the paper the spatial context for them.

**Response 7:** Authors are sincerely thankful to the reviewer for pointing out this. As per reviewer suggestion, the authors have been included the district boundaries on both the LULC maps. There are total 77 districts in the complete river basin but the water quality of the monitoring stations is mainly affected by the districts in which they are located. It is to be noted that due to religious, economic and historical importance of River Ganga, the most important cities of the districts selected for analysis are located on the banks or in the proximity to River Ganga. Hence, the district boundaries are overlaid and highlighted in magenta colour on the LULC maps (Figure 4) to give the reader of the paper the spatial context for them. The modified Figure 4 is as below:

[Figure]

**Figure 4.** LULC maps of Upper Ganga River basin **(a)** LULC map of February/March 2001, and **(b)** LULC map of February/March 2012

**Comment 8:** I would recommend the addition of district specific land use change maps to help support your discussion. At present, it is impossible to visually relate the pattern of land use change to the water quality and population statistics because the scale of the mapping in figure 4 is too small.

**Response 8:** Authors are sincerely grateful to the reviewer for suggesting the use of district wise changes in LULC classes to relate changes in the water quality of the monitoring station. Earlier in the manuscript LULC changes of complete river basin were used to relate the Overall Index of Pollution (OIP) of the river basin which could only provide the broad overview and causal links between LULC changes and water quality status of the study area at a regional scale. But study of district wise LULC changes is extremely helpful in comprehending the water quality changes at the local scale and to identify source of pollutants at a particular monitoring station. The statistics presented in Table 6 given below is important to explain the changes in LULC of the districts in Section 5.2 of the revised manuscript. It is further used in developing the relationship between the LULC and OIP of the river basin which is explained in detail in Subsection 5.4.2. All the reviewers have suggested removing redundant information from the revised manuscript especially if tables and figures are showing same information.

Reviewer#1 (Comment 7): *The population statistics on page 18 should be presented as a table with the PGR statistics given for each district. Figure 3 on page 19 is repetition of the data that will be presented in the table so remove figure 3.*

Reviewer#2 (Comment 5): *Find a clear argument that flows throughout the paper and only select figures and data that make it easier for the reader to understand this argument. E.g. remove superfluous data such as the city populations on page 18.*

Reviewer#3 (Comment 5): *Table 4 and Figure 5 present the same results. Only one of them should be included in the paper to avoid repetition of information.*

In addition to the revised manuscript, LULC maps in very high resolution will be provided as a supplementary file (Figure 4). District boundaries are now highlighted on the LULC of both the years therefore; very high-resolution image file can be used to visually depict the changes in the LULC between 2001 and 2011. Keeping all the above reasons in mind, instead of figures Table 6 is presented illustrating district wise changes in LULC of Upper Ganga River basin.

**Table 6.** District wise changes in LULC **(a)** Uttarkashi, **(b)** Dehradun, **(c)** Kanpur, **(d)** Allahabad, and **(e)** Varanasi

**(a)**

| Uttarkashi (LULC Class) | 2001 % | 2012% | % Change (2001-2012) |
|---|---|---|---|
| Forest | 39.3 | 39.7 | 1.1 |
| Wastelands | 10.3 | 8.3 | -19.3 |
| Water Bodies | 1.4 | 1.5 | 4.6 |
| Agricultural Land | 0.6 | 1.4 | 122.8 |
| Builtup Area | 0.2 | 0.6 | 186.3 |
| Snow and Glacier | 48.2 | 48.6 | 0.8 |
| **Total Area %** | 100.0 | 100.0 | |

**(b)**

| Dehradun (LULC Class) | 2001 % | 2012% | % Change (2001-2012) |
|---|---|---|---|
| Forest | 59.8 | 59.8 | 0.1 |
| Wastelands | 18.8 | 3.4 | -82.1 |
| Water Bodies | 4.8 | 4.3 | -9.8 |
| Agricultural Land | 13.5 | 20.3 | 50.6 |
| Builtup Area | 3.2 | 12.2 | 283.9 |
| **Total Area %** | 100.0 | 100.0 | |

**(c)**

| Kanpur (LULC Class) | 2001 % | 2012% | % Change (2001-2012) |
|---|---|---|---|
| Forest | 0.3 | 0.3 | 8.7 |
| Wastelands | 23.4 | 4.7 | -79.8 |
| Water Bodies | 2.5 | 2.6 | 3.8 |
| Agricultural Land | 63.7 | 67.0 | 5.2 |
| Builtup Area | 10.1 | 25.3 | 152.1 |
| **Total Area %** | 100.0 | 100.0 | |

**(d)**

| Allahabad (LULC Class) | 2001 % | 2012% | % Change (2001-2012) |
|---|---|---|---|
| Forest | 1.5 | 1.5 | -1.2 |
| Wastelands | 22.1 | 16.0 | -27.8 |
| Water Bodies | 3.0 | 3.1 | 1.3 |
| Agricultural Land | 70.5 | 73.4 | 4.2 |
| Builtup Area | 2.8 | 6.0 | 111.7 |
| **Total Area %** | 100.0 | 100.0 | |

**(e)**

| Varanasi (LULC Class) | 2001 % | 2012% | % Change (2001-2012) |
|---|---|---|---|
| Forest | 0.6 | 0.7 | 24.4 |
| Wastelands | 16.8 | 6.0 | -64.5 |
| Water Bodies | 3.1 | 3.3 | 7.1 |
| Agricultural Land | 76.8 | 79.4 | 3.4 |
| Builtup Area | 2.7 | 10.5 | 291.8 |
| **Total Area %** | 100.0 | 100.0 | |

**Comment 9:** In place of table 4, I would present a cross tabulation table of the 2001 and 2012 LULC classes. This will clearly show the reader what has changed to what and then the gross and net changes can be shown in figure 5.

**Response 9:** We acknowledge the suggestions reviewer has given. Table 5 given below presents the cross tabulation table of the 2001 and 2012 which shows what LULC class has changed to what. Gross and net changes are shown in Figure 5 as suggested. Further a

paragraph is added in Section 5.2 of the revised manuscript describing the change matrix and it is explained why it has happened. As district wise LULC change tables are also added to this section, editing is done accordingly.

**Table 5.** Change matrix showing LULC interconversion between the year 2001 and 2012 in Upper Ganga River basin

| LULC Class | F | WL | WB | AG | BU | SG | LULC 2001 |
|---|---|---|---|---|---|---|---|
| **F** | 13.3 | 0.0 | 0.0 | 0.0 | 0.0 | 0.0 | **13.3** |
| **WL** | 1.7 | 11.4 | 0.0 | 1.7 | 2.2 | 0.1 | **17.1** |
| **WB** | 0.2 | 0.0 | 1.8 | 0.0 | 0.0 | 0.0 | **2.0** |
| **AG** | 0.0 | 0.0 | 0.0 | 58.3 | 0.0 | 0.0 | **58.3** |
| **BU** | 0.0 | 0.0 | 0.0 | 0.0 | 5.3 | 0.0 | **5.3** |
| **SG** | 0.0 | 0.0 | 0.0 | 0.0 | 0.0 | 4.0 | **4.0** |
| **LULC 2012** | **15.2** | **11.4** | **1.8** | **60.0** | **7.5** | **4.1** | **100.0** |

\* Figures indicate the percentage of basin area

[Figure]

[Figure]

(a)                                         (b)

**Figure 5.** Graph showing LULC of the years 2001-2012 (a) LULC area in percentage (%) and (b) LULC changes from 2001-2012 in Upper Ganga River basin

**Technical Comments:**

**Comment 1:** Repetition is a problem in several places through the text. The worst example of this is presenting results back again in the conclusions. Go through the paper carefully and remove the repetition.

**Response 1:** Reviewer suggestions regarding the repetition of the words in the manuscript have been changed in the revised manuscript. Section 6 (Conclusions) is edited as required. Our endeavour will be that the revised paper is much better than the current version.

**Comment 2:** The English needs to be corrected throughout the paper. Please find a native English speaker to go through the paper to correct for missing words, improve the phraseology used and correct the grammar.

**Response 2:** The whole manuscript has been checked and modified suitably. Authors have corrected the missing words and grammar, improved the phraseology and checked the English in whole manuscript.

**Comment 3:** Avoid the use of superlatives e.g. 'tremendous' and 'colossal changes'. These terms cannot be substantiated and so should not be used.

**Response 3:** Authors remove the use of superlatives words in whole manuscript and used appropriate words in revised manuscript whenever required.

**Comment 4:** The long list of water quality indicators is excessive. Highlight only those that could be relevant to the data available for this study and those commonly used.

**Response 4:** Authors acknowledge the points the reviewer is making. For selecting water quality index the following criteria is followed (Abbasi and Abbasi, 2012; Horton 1965): (i) limited number of variables should be handled by the used index to avoid making the index unwieldy; (ii) the variables used in the index should be significant in most areas, (iii) only reliable data variables for which the data are available should be included. Therefore, the water quality parameters are chosen for this study with care. If each and every possible parameter is included in the index then it will become unwieldy and will not represent the water quality status of the particular region. Hence, only those water quality parameters, which together reflect the overall water quality at a location or for a given end use should be considered. The acceptability criteria of water quality indices vary from region to region due to differences in the water quality standards by the Government organizations of the region. The water quality standards vary for different countries due to the differences in the climatic and LULC characteristics of the region; and physico-chemical properties of the water body under study. They affect the water quality parameters therefore, the water quality standards vary for different countries.

Overall Index of Pollution (OIP) developed by Sargaonkar and Deshpande (2003) is a general water quality classification scheme developed specifically for tropical Indian conditions where in the proposed classes (Excellent, Acceptable, Slightly Polluted, Polluted and Heavily Polluted water), the concentration levels/ranges of the significant water quality indicator parameters viz. Hardness $CaCO_3$, TDS, BOD, Cl, Coliform Total, Colour, DO%, pH, and Turbidity are defined based on the Indian water quality standards (Indian Standard Specification for Drinking Water, IS-10500, 1983 and Central Pollution Control Board, Government of India, classification of inland surface water, CPCB- ADSORBS/3/78-79). This classification scheme takes into consideration various international water quality assessment schemes viz. European Community (EC) standards, World Health Organization (WHO) guidelines, standards by WQIHSR and Tehran Water Quality Criteria by McKee and Wolf. The concentration ranges used in the classes and the classification scheme helps to evaluate the surface water quality status with respect to particular individual parameter whereas the OIP helps to assess the overall water quality status specifically in the Indian context. This index uses only those water quality parameters that are important to Indian context. Therefore, of all the water quality parameters available, only 7 most important ones i.e. BOD, DO%, Total Coliform (TC), F, Turbidity, pH and Hardness $CaCO_3$ that are affected due to changes in LULC are chosen after extensive literature review. For example BOD, DO%, and Total Coliform (TC) are affected by urban pollution. F, Turbidity and pH are general water quality parameters affected by both natural and anthropogenic factors. However, Hardness $CaCO_3$ is a parameter affected mainly by agricultural activities and urban

pollution. It was discussed in the Section 1 "introduction" of the earlier manuscript. This section is slightly edited and updated in the revised manuscript. While discussing the results in Section 5.4.2, Individual Parameter Indices (IPIs) of only those water quality parameters are highlighted whose values are in "*polluted*" category and those are discussed in detail where significant change is observed over a period of 2001 to 2011. The IPIs and OIPs in polluted category are highlighted in the Table 9 given below.

**Table 9.** Individual parameter indices (IPIs) and overall indices of pollution (OIPs) computed at various water quality monitoring stations of Upper Ganga River basin over periods of 2001 and 2012 for pre-monsoon, monsoon and post-monsoon seasons

**(i)**

| Parameters | Water Quality Monitoring Stations | | | | | | | | | | | | | | |
| --- | --- | --- | --- | --- | --- | --- | --- | --- | --- | --- | --- | --- | --- | --- | --- |
| | Uttarkashi | | | Rishikesh | | | Kanpur | | | Allahabad | | | Varanasi | | |
| | May | Jul | Nov | May | Jul | Nov | May | Jul | Nov | May | Jul | Nov | May | Jul | Nov |
| BOD | 1.00 | 1.00 | 1.00 | 1.00 | 1.00 | 1.00 | 2.87 | 2.40 | 2.60 | 2.67 | 2.80 | 2.47 | 1.67 | 1.47 | 1.20 |
| DO% | 1.33 | 1.28 | 1.27 | 2.49 | 3.24 | 2.97 | 1.27 | 0.79 | 0.99 | 1.06 | 1.61 | 0.86 | 1.20 | 1.06 | 1.54 |
| F | 1.00 | 1.00 | 1.00 | 1.00 | 1.00 | 1.00 | 1.00 | 1.00 | 1.00 | 1.00 | 1.00 | 1.00 | 1.00 | 1.00 | 1.00 |
| Hardness CaCO$_3$ | 1.00 | 1.00 | 1.00 | 1.78 | 1.00 | 1.00 | 1.99 | 1.80 | 1.87 | 1.95 | 3.16 | 2.66 | 1.99 | 2.89 | 2.45 |
| pH | 2.76 | 2.76 | 2.76 | 2.76 | 2.76 | 2.76 | 2.52 | 3.33 | 2.76 | 3.03 | 3.33 | 3.03 | 3.03 | 3.65 | 3.03 |
| Total Coliform | - | - | - | - | - | - | - | - | - | 3.43 | 4.60 | 4.98 | 4.02 | 3.48 | 3.21 |
| Turbidity | - | - | - | - | - | - | 1.00 | 1.00 | 1.00 | 1.00 | 1.00 | 1.00 | 1.00 | 1.00 | 1.00 |
| **OIP (2001)** | **1.42** | **1.41** | **1.41** | **1.81** | **1.80** | **1.75** | **2.61** | **2.49** | **2.54** | **2.02** | **2.50** | **2.29** | **1.99** | **2.08** | **1.92** |

**(ii)**

| Parameters | Water Quality Monitoring Stations | | | | | | | | | | | | | | |
| --- | --- | --- | --- | --- | --- | --- | --- | --- | --- | --- | --- | --- | --- | --- | --- |
| | Uttarkashi | | | Rishikesh | | | Kanpur | | | Allahabad | | | Varanasi | | |
| | May | Jul | Nov | May | Jul | Nov | May | Jul | Nov | May | Jul | Nov | May | Jul | Nov |
| BOD | 1.00 | 1.00 | 1.00 | 1.00 | 1.00 | 1.00 | 4.67 | 6.67 | 2.67 | 1.93 | 2.13 | 1.60 | 2.00 | 2.60 | 1.93 |
| DO% | 2.36 | 2.97 | 2.36 | 1.81 | 2.22 | 2.08 | 1.47 | 2.22 | 1.20 | 1.54 | 1.49 | 0.65 | 1.13 | 0.65 | 0.65 |
| F | 1.00 | 1.00 | 1.00 | 1.00 | 1.00 | 1.00 | 1.00 | 1.00 | 1.00 | 1.00 | 1.00 | 1.00 | 1.00 | 1.00 | 1.00 |
| Hardness CaCO$_3$ | 1.00 | 1.00 | 1.00 | 1.00 | 1.00 | 1.00 | 2.10 | 2.02 | 2.91 | 1.97 | 1.86 | 1.92 | 1.90 | 1.00 | 1.82 |
| pH | 2.09 | 1.91 | 1.74 | 2.09 | 2.52 | 2.09 | 4.81 | 3.65 | 2.76 | 3.03 | 4.00 | 3.03 | 4.81 | 3.65 | 4.81 |
| Total Coliform | - | - | - | - | - | - | - | - | - | 4.05 | 4.11 | 3.90 | 4.14 | 5.97 | 3.93 |
| Turbidity | - | - | - | - | - | - | 1.00 | 1.20 | 1.08 | 1.00 | 1.00 | 1.00 | 1.00 | 1.00 | 1.00 |
| **OIP (2012)** | **1.49** | **1.58** | **1.42** | **1.38** | **1.55** | **1.44** | **2.51** | **2.79** | **2.77** | **2.07** | **2.23** | **1.87** | **2.28** | **2.27** | **2.16** |

**Comment 5:** Avoid excessive precision e.g. 238,347.74 km$^2$. At the scale you are working expressing to the nearest km$^2$ is appropriate.

**Response 5:** Authors have rounded off and changed the value to 23,8348 km$^2$ approximately (total drainage area) in the revised manuscript in place of 238,347.74 km$^2$.

**Comment 6:** Figure 1 - The inset map should be inside the map frame, the water quality monitoring station location labels conflict with the basin boundary line - change the position of the labels so that this doesn't occur, remove the underscore characters from the legend text (this also applies to figure 4).

**Response 6:** Authors have modified Figure 1 according to the reviewer suggestions. The modified figure has no conflict between the basin boundary and labels of water quality monitoring stations. The updated Figure 1 is given below and the same has been updated in the revised manuscript. Authors have removed underscore characters from the legend text of Figure 4 and it is updated in the revised manuscript. The updated Figure 4 is already given above in Response 7 of this draft.

[Figure]

**Figure 1.** Location map of the study area in northern India and water quality monitoring stations across Upper Ganga River basin

**Comment 7:** The population statistics on page 18 should be presented as a table with the PGR statistics given for each district. Figure 3 on page 19 is repetition of the data that will be presented in the table so remove figure 3.

**Response 7:** As per reviewer suggestion, the population statistics have been presented in Table 4 (revised manuscript) showing the Population Growth Rate (PGR) and total population in the census years 2001 and 2011. A discussion on this table is made in the Section 5.1 of the revised manuscript. Figure 3 have been removed in revised manuscript.

**Table 4.** Table showing Population Growth Rate (PGR) % and total population in the census years 2001 and 2011

| S. No. | Districts | Total Population (2001) | Total Population (2011) | Population Growth Rate (PGR) % |
|---|---|---|---|---|
| 1 | Agra | 3620436 | 4418797 | 22.1 |
| 2 | Aligarh | 2992286 | 3673889 | 22.8 |
| 3 | Allahabad | 4936105 | 5954391 | 20.6 |
| 4 | Almora | 630567 | 622506 | -1.3 |
| 5 | Ambedkar Nagar | 2026876 | 2397888 | 18.3 |
| 6 | Azamgarh | 3939916 | 4613913 | 17.1 |
| 7 | Bageshwar | 249462 | 259898 | 4.2 |
| 8 | Baghpat | 1163991 | 1303048 | 11.9 |
| 9 | Bahraich | 2381072 | 34,87,731 | 46.5 |
| 10 | Ballia | 2761620 | 32,39,774 | 17.3 |
| 11 | Balrampur | 1682350 | 2148665 | 27.7 |
| 12 | Barabanki | 2673581 | 3260699 | 22.0 |
| 13 | Bareilly | 3618589 | 4448359 | 22.9 |
| 14 | Basti | 2084814 | 24,61,056 | 18.0 |
| 15 | Bhojpur | 2243144 | 2728407 | 21.6 |
| 16 | Bijnor | 3131619 | 36,82,713 | 17.6 |
| 17 | Budaun | 3069426 | 3681896 | 20.0 |
| 18 | Bulandshahar | 2913122 | 3499171 | 20.1 |
| 19 | Buxar | 1402396 | 1706352 | 21.7 |
| 20 | Chamoli | 370359 | 391605 | 5.7 |
| 21 | Champawat | 224542 | 259648 | 15.6 |
| 22 | Dehradun | 1282143 | 1696694 | 32.3 |
| 23 | Deoria | 2712650 | 3100946 | 14.3 |
| 24 | Etah | 15,61,705 | 1774480 | 13.6 |
| 25 | Faizabad | 2088928 | 2470996 | 18.3 |
| 26 | Farrukhabad | 1570408 | 1885204 | 20.0 |
| 27 | Fatehpur | 2308384 | 26,32,733 | 14.1 |
| 28 | Firozabad | 2052958 | 2498156 | 21.7 |
| 29 | Gautam Buddha Nagar | 1202030 | 1648115 | 37.1 |
| 30 | Ghaziabad | 3290586 | 4681645 | 42.3 |
| 31 | Ghazipur | 3037582 | 3620268 | 19.2 |
| 32 | Gonda | 2765586 | 3433919 | 24.2 |
| 33 | Gopalganj | 2152638 | 2562012 | 19.0 |
| 34 | Gorakhpur | 3769456 | 4440895 | 17.8 |
| 35 | Hardoi | 3398306 | 4092845 | 20.4 |
| 36 | Haridwar | 1447187 | 1890422 | 30.6 |
| 37 | Hathras | 1336031 | 1564708 | 17.1 |
| 38 | Jaunpur | 3911679 | 4494204 | 14.9 |
| 39 | Jyotiba Phule Nagar | 1499068 | 1840221 | 22.8 |

| | | | | |
|---|---|---|---|---|
| 40 | Kannauj | 1388923 | 1656616 | 19.3 |
| 41 | Kanpur Dehat | 1563336 | 1796184 | 14.9 |
| 42 | Kanpur Nagar | 4167999 | 4581268 | 9.9 |
| 43 | Kaushambi | 1293154 | 1599596 | 23.7 |
| 44 | Kheri | 3207232 | 4021243 | 25.4 |
| 45 | Kinnaur | 78334 | 84121 | 7.4 |
| 46 | Kushinagar | 2893196 | 3564544 | 23.2 |
| 47 | Lucknow | 3647834 | 4589838 | 25.8 |
| 48 | Maharajganj | 2173878 | 2684703 | 23.5 |
| 49 | Mainpuri | 1596718 | 1868529 | 17.0 |
| 50 | Mau | 1853997 | 2205968 | 19.0 |
| 51 | Meerut | 2997361 | 3443689 | 14.9 |
| 52 | Mirzapur | 2116042 | 2496970 | 18.0 |
| 53 | Moradabad | 3810983 | 4772006 | 25.2 |
| 54 | Muzaffarnagar | 3543362 | 4143512 | 16.9 |
| 55 | Nainital | 762909 | 954605 | 25.1 |
| 56 | Patna | 4718592 | 5838465 | 23.7 |
| 57 | Pauri Garhwal | 697078 | 687271 | -1.4 |
| 58 | Pilibhit | 1645183 | 2031007 | 23.5 |
| 59 | Pithoragarh | 462289 | 483439 | 4.6 |
| 60 | Pratapgarh | 2731174 | 3209141 | 17.5 |
| 61 | Rae Bareli | 2872335 | 3405559 | 18.6 |
| 62 | Rampur | 1923739 | 2335819 | 21.4 |
| 63 | Rudraprayag | 227439 | 242285 | 6.5 |
| 64 | Sant Kabir Nagar | 1420226 | 1715183 | 20.8 |
| 65 | Sant Ravidas Nagar | 1353705 | 1578213 | 16.6 |
| 66 | Saran | 3248701 | 3951862 | 21.6 |
| 67 | Shahjahanpur | 2547855 | 3006538 | 18.0 |
| 68 | Shravasti | 1176391 | 1117361 | -5.0 |
| 69 | Siddharthnagar | 2040085 | 2559297 | 25.5 |
| 70 | Sitapur | 3619661 | 4483992 | 23.9 |
| 71 | Siwan | 2714349 | 3330464 | 22.7 |
| 72 | Sultanpur | 3214832 | 3797117 | 18.1 |
| 73 | Tehri Garhwal | 604747 | 618931 | 2.3 |
| 74 | Udhamsingh Nagar | 1235614 | 1648902 | 33.4 |
| 75 | Unnao | 2700324 | 3108367 | 15.1 |
| 76 | Uttarkashi | 295013 | 330086 | 11.9 |
| 77 | Varanasi | 3138671 | 3676841 | 17.1 |
| **Total** | **Upper Ganga River basin** | 171186859 | 206188401 | 20.45 |

**Comment 8:** The information on pH on page 30 can be regarded as a known fact and so does not need to be explained.

**Response 8:** As per reviewer suggestion, pH information on page 30 has been removed and suitably modified in manuscript.

**Comment 9:** The discussion that follows the pH description needs to be written with reference to just the set of figures showing the OIP values plotted against the stations. The other figure is effectively repetition so remove the other figure.

**Response 9:** As per reviewer suggestion, Figure 6 (a), 6 (b) and 6 (c) has been removed. The discussion that follows the pH description is rewritten with reference to the OIP values plotted against the stations. It is updated in the revised manuscript in Section 5.4.2.

**References**

Abbasi, T., & Abbasi, S. A. (2012). Water quality indices. Elsevier.

Brivio, P. A., Doria, I., & Zilioli, E. (1993). Aspects of spatial autocorrelation of Landsat TM data for the inventory of waste-disposal sites in rural environments. Photogrammetric engineering and remote sensing.

Chen, D. (2004). A Multi-Resolution Analysis and Classification framework for improving Land use/cover mapping from Earth Observation Data. The International Archives of the Photogrammetry, Remote Sensing and Spatial Information Sciences, 34, 1187-1191.

Chen, J., Zhu, X., Vogelmann, J. E., Gao, F., & Jin, S. (2011). A simple and effective method for filling gaps in Landsat ETM+ SLC-off images. Remote sensing of environment, 115(4), 1053-1064.

Congalton, R. G. (1991). A review of assessing the accuracy of classifications of remotely sensed data. Remote sensing of environment, 37(1), 35-46.

Foody, G. M. (2002). Status of land cover classification accuracy assessment. Remote sensing of environment, 80(1), 185-201.

Gao, G., Liu, T., & Gu, Y. (2016, July). Improved neighborhood similar pixel interpolator for filling unsacn multi-temporal Landsat ETM+ data without reference. In Geoscience and Remote Sensing Symposium (IGARSS), 2016 IEEE International (pp. 2336-2339). IEEE.

Gebremicael, T. G., Mohamed, Y. A., van der Zaag, P., & Hagos, E. Y. (2017). Quantifying longitudinal land use change from land degradation to rehabilitation in the headwaters of Tekeze-Atbara Basin, Ethiopia. Science of the Total Environment.

Gill, T., Collett, L., Armston, J., Eustace, A., Danaher, T., Scarth, P., ... & Phinn, S. (2010). Geometric correction and accuracy assessment of Landsat-7 ETM+ and Landsat-5 TM imagery used for vegetation cover monitoring in Queensland, Australia from 1988 to 2007. Journal of Spatial Science, 55(2), 273-287.

Gonçalves, R. P., Assis, L. C., & Vieria, C. A. O. (2007, October). Comparison of sampling methods to classification of remotely sensed images. In IV International Symposium in Precision in Agriculture (pp. 23-25).

Gu, J., Sun, G., Pan, Y., & Fan, D. (2012, August). Accuracy assessment based on the distribution of the classified errors on classified map: Simulation analysis. In Agro-

Geoinformatics (Agro-Geoinformatics), 2012 First International Conference on (pp. 1-4). IEEE.

Hashemian, M. S., Abkar, A. A., & Fatemi, S. B. (2004, July). Study of sampling methods for accuracy assessment of classified remotely sensed data. In International congress for photogrammetry and remote sensing (pp. 1682-1750).

Horton, R. K. (1965). An index number system for rating water quality. Journal of Water Pollution Control Federation, 37(3), 300-306.

Kiptala, J. K., Mohamed, Y., Mul, M. L., Cheema, M. J. M., & Van der Zaag, P. (2013). Land use and land cover classification using phenological variability from MODIS vegetation in the Upper Pangani River Basin, Eastern Africa. Physics and Chemistry of the Earth, Parts A/B/C, 66, 112-122.

Liu, X., & Ding, Y. (2017). Auxiliary pixel data selection for recovering Landsat ETM+ SLC-off images. The Egyptian Journal of Remote Sensing and Space Science.

Lu, D., & Weng, Q. (2007). A survey of image classification methods and techniques for improving classification performance. International journal of Remote sensing, 28(5), 823-870.

Muriithi, F. K. (2016). Land use and land cover (LULC) changes in semi-arid sub-watersheds of Laikipia and Athi River basins, Kenya, as influenced by expanding intensive commercial horticulture. Remote Sensing Applications: Society and Environment, 3, 73-88.

Na, X., Zhang, S., Zhang, H., Li, X., Yu, H., & Liu, C. (2009). Integrating TM and ancillary geographical data with classification trees for land cover classification of marsh area. Chinese Geographical Science, 19(2), 177-185.

Remote sensing & spatial analysis Lab (2018); https://xiaolinzhu.weebly.com/open-source-code.html. Accessed on 12 March, 2018.

Samal, D. R., & Gedam, S. S. (2015). Monitoring land use changes associated with urbanization: An object based image analysis approach. European Journal of Remote Sensing, 48(1), 85-99.

Shukla, A. K., Shukla, S., & Ojha, R. (2017). Geospatial Technologies for Rainfall and Atmospheric Water Vapor Measurement over Arid Regions of India. In Sustainable Water Resources Management (pp. 263-292).

Stehman, S. V. (2009). Sampling designs for accuracy assessment of land cover. International Journal of Remote Sensing, 30(20), 5243-5272.

Stehman, S. V., & Czaplewski, R. L. (1998). Design and analysis for thematic map accuracy assessment: fundamental principles. Remote sensing of environment, 64(3), 331-344.

USGS 2018: https://landsat.usgs.gov/slc-products-background accessed on 12 March, 2018.

Wijaya, A., Marpu, P. R., & Gloaguen, R. (2007). Geostatistical Texture Classification of Tropical Rainforest in Indonesia (in CD ROM). In ISPRS International Symposium on Spatial Data Quality, ITC Enschede, The Netherlands.

Zhu, X., & Liu, D. (2014). MAP-MRF approach to Landsat ETM+ SLC-Off image classification. IEEE Transactions on Geoscience and Remote Sensing, 52(2), 1131-1141.

Zhu, X., Gao, F., Liu, D., & Chen, J. (2012). A modified neighborhood similar pixel interpolator approach for removing thick clouds in Landsat images. IEEE Geoscience and Remote Sensing Letters, 9(3), 521-525.

---

## Author Comment (AC2) · 13 Mar 2018

**Authors' Response to the Reviewer#2 Comments**

**Manuscript Ref.: hess-2017-384**

**Title: Population Growth – Land Use/Land Cover Transformations-Water Quality Nexus in Upper Ganga River Basin**

**Authors:** Anoop Kumar Shukla, Chandra Shekhar Prasad Ojha, Ana Mijic, Wouter Buytaert, Shray Pathak, Rahul Dev Garg and Satyavati Shukla

We sincerely thank the reviewers for offering their critical comments and valuable suggestions that has helped to improve the manuscript. We hereby provide our responses to the reviewer's comments and highlight the changes made in the revised manuscript based on the comments provided. These have been incorporated in the revised manuscript as follows. The point wise replies of the comments of the Reviewer#2 are given below:

**General comments:**

The authors have a clear understanding of his topic area and have applied remote sensing and data collation techniques to answer investigate the impacts of demographic changes on water quality in the upper Ganga River Basin. However, I believe considerable work is needed to bring the paper to publishable standard.

We are grateful to the anonymous reviewer 2 for his/her efforts in providing extremely useful comments that has helped in improving our research work. The comments provided by all the three reviewers are duly complied in the revised manuscript which we believe have significantly improved the manuscript.

**Major concerns are threefold:**

**Comment 1:** What is it about this paper that is academically novel? Is it the application of existing methods to a new area? Or the evaluation of new methodology? The authors need to make this much clearer as the current introduction suggests that the paper aims to identify drivers; however the method takes drivers as given and the conclusions focus on the utility of the method…which aspect of the research are important to academia as a whole?

**Response 1:** Authors sincerely thank the reviewer for pointing out this very important question. We agree with reviewer that the previous manuscript required extensive restructuring and editing which is done in the revised manuscript. The main aim of the study is to analyse the *causative connection* (nexus) between the changing patterns of population, Land Use/Land Cover (LULC) and water quality of water stressed Upper Ganga River basin. In this study a comprehensive set of analyses are presented to assess and comprehend the current status of the population-LULC-water quality nexus in the study region, with respect to their changing patterns from 2001 to 2011. The present study is conducted at two different spatial scales i.e. (a) at river basin level (small scale), and (b) at district level (large scale) for three different seasons viz. pre-monsoon, monsoon and monsoon seasons. Such study is not done before for Upper Ganga River basin. Various methodologies are developed to study effects of LULC changes on water quality. But these methods cannot be applied directly to a region because of the differences in the data availability, climatic, topographic and LULC variations which may introduce errors. Hence, necessary modifications are made in the existing evaluation methodology as per the requirement. And a relationship is developed between LULC and Overall Index of Pollution (OIP) using multi linear regression. Findings

from this research work may help engineers, planners, policy makers and different stakeholders for sustainable development in the river basin. The novelty of the work is discussed in detail in Response 4 of this draft.

**Comment 2:** There is an imbalance in the level of detail applied throughout. Some areas provide too much detail (I don't think the equation for growth rate needs five lines of explanation and an equation) whereas some key aspects of the methodology (e.g. radiometric correction) aren't described at all and there is little critical interegation of the results (why are they the way they are – what factors contribute and what doesn't). There is a need to work on structuring the data to make it make better sense.

**Response 2:** Authors are sincerely thankful to the reviewer for pointing it out. The revised manuscript is thoroughly edited and modified as required. The redundant information like extra figures, tables and texts are removed wherever they are suggested. The paragraph discussing the growth rate is trimmed down and the equation is removed. As suggested the methodology section including the radiometric correction is elaborated as a whole. It is described in detail in the Responses 1 to 6 (specific comments) of Reviewer#1. The data is restructured incorporating the comments from all the three reviewers and results are described in detail wherever necessary mainly focussing on cause and effects.

**Comment 3:** The paper needs a thorough restructure, as it is repetitive, provides considerable amounts of extraneous material and doesn't clearly signpost what is relevant to read. The language would benefit from a thorough proof read by a native English speaker also.

**Response 3:** As per the reviewer suggestions, repetitive and extraneous material is removed and the write up is trimmed down just showing relevant information and to the point explanation of the data presented. Authors have corrected the missing words and grammar, improved the phraseology and checked the English in whole manuscript suitably. To properly restructure the manuscript and for better description of the results, some subsections are added as required. Hence, review suggestions regarding modification of structure of the paper are duly considered in the revised manuscript. Our endeavour will be that the revised paper is much better than the current version.

**Technical/Specific comments:**

**Comment 1:** There is too great a level (e.g. population figures to the person on page 18. Round to the nearest 1000?)

**Response:** As suggested by the reviewers, all the population figures used on page 18 are rounded off to the nearest 1000 and updated in the revised manuscript.

**Comment 2:** Check English.

**Response 2:** The whole manuscript have been suitably modified and the English is duly checked and corrected wherever it was required.

**Comment 3:** Identify a clear research question.

**Response 3:** The primary research question answered in this work is as follows: "What is the causative connection (nexus) between the changing patterns of population, Land Use/Land Cover (LULC) and water quality of water stressed Upper Ganga River basin"?

**Comment 4:** Explain why it is novel – this is very important and does not come across well in the current draft.

**Response 4:** Ganga River is extremely significant to its inhabitants as it supports various important services such as: (i) source of irrigation for farmers in agriculture and horticulture; (ii) provides water for domestic and industrial purposes in urban areas; (iii) source of hydro-power; (iv) serves as a drainage for waste and helps in pollution control; (v) acts as support system for terrestrial and aquatic ecosystems, (vi) provides religious and cultural services; (vii) helps in navigation; (viii) supports fisheries and other livelihood options, etc. (Amarasinghe et al. 2016; SoE report, 2012; Watershed Atlas of India, 2014). However, for the past few decades Upper Ganga River basin has experienced rapid growth in population, urbanization, industrialization, infrastructure development activities and agriculture. Due to these changes, maintaining the acceptable water quality for various uses is being challenged. Therefore, there is a need to study the causative connection (nexus) between the changing patterns of population, Land Use/Land Cover (LULC) and water quality at both river basin (small scale) and at districts level (large scale) for three different seasons. Such study is yet to be done for this large river basin. OIP developed specifically for Indian context is used in this study to assess the status of water quality across the study area. Due to unavailability of the continuous population, satellite and water quality data at desired interval, establishing the interrelationship between these factors is not trivial. Hence, in order to achieve the objectives a comprehensive set of analyses are performed in this study. Between LULC and OIP a 2-time slice analysis is done for the years 2001 and 2012 with seasonal component. A relationship is developed between LULC and OIP (Indian WQI) using correlation and multi linear regression analyses. Further, trend (Mann-Kendall method) analysis was performed on monthly water quality parameters of the monitoring stations from 2001 to 2012 to understand their temporal variations over the years. Also, it was interesting to see the effects of seasonal variations on status of water quality. Hence, finally the results were inferred from these comprehensive set of analyses to understand nexus between population-LULC-water quality of Upper Ganga River basin which is our main contribution.

**Comment 5:** Find a clear argument that flows throughout the paper and only select figures and data that make it easier for the reader to understand this argument. E.g. Remove superfluous data such as the city populations on page 18.

**Response 5:** As per the reviewer suggestion, whole manuscript write up has been improved wherever required. Our endeavor will be that the revised manuscript is much better than the current version. From page 18 (old manuscript) superfluous data have been removed and used rounding to the nearest values. Authors are grateful to the reviewer for giving suggestions to follow a clear argument throughout the manuscript. This point greatly helped to improve the revised manuscript. All across the manuscript the following argument was followed: "There is a causative connection between the population-LULC-water quality of Upper Ganga River basin and they are interrelated with each other". Only those tables and figures are used that

justify the given argument and helped to explain the results. City population data given on page 18 is removed as suggested.

**Comment 6:** Section 5.1 could be summarised in a paragraph of text. I over-simplify but much of it can be covered by the following sentence: "Growth rates for urban and rural areas were calculated from official statistics (Figure 3)". Is the individual city data relevant? How does it fit to the overall argument? Would spatial/mapped data be more useful or relevant when compared with RS data? Figure 3 simply repeats statistics shown in the text.

**Response 6:** Authors are sincerely thankful to reviewer for suggesting the summary of Section 5.1 as: "Growth rates for urban and rural areas were calculated from official statistics (Figure 3)". It is incorporated in Section 5.1 and further suitable updations and modifications were done in this section. The statistics of Figure 3 which is repeated in the text is removed and justifications are given on why these changes are occurring. Write up is improved suitably wherever it is required.

The water quality of a monitoring station is highly influenced by proximity of a city due to domestic and industrial discharges into the river from urban areas. The water quality status of these monitoring stations is dependent on the type of activities undergoing in and around the city. For e.g. water quality of Ankinghat monitoring station is affected mainly by discharge of effluents from tanning industries, municipal discharges and solid waste disposal into the river. Source of water quality pollutants are both point source and non-point source. Pollutants from both urban and rural areas affect the water quality. In urban areas water quality is mainly affected by municipal discharges and industrial effluents. However, in rural areas it is mainly affected by agricultural activities. Harmful chemical compounds are used in the agricultural lands in the form of herbicides, pesticides, weedicides, etc. During heavy rainfall, runoffs generated from these fields discharge directly into the nearby streams and carry these chemicals to the stream which causes water pollution. Therefore, just considering city data is not very rational approach. In a study, buffer zones of different thresholds were created surrounding a water quality monitoring station to determine the dominant LULC class that affects the water quality of that particular station (Kibena et al. 2014). In this study our argument is to determine the causative connection between changing patterns of population, Land Use/Land Cover (LULC) and water quality of the study area. The population data is available at district level not at buffer level. Reviewer#1 has suggested using district specific LULC statistics to establish relationship with OIP.

Reviewer#1 (Comment 8): *I would recommend the addition of district specific land use change maps to help support your discussion. At present, it is impossible to visually relate the pattern of land use change to the water quality and population statistics because the scale of the mapping in figure 4 is too small.*

Districts selected in this study consist of both urban and rural areas. District wise LULC change is extremely helpful in comprehending the water quality changes at the local scale and to identify source of pollutants at a particular monitoring station. Whereas LULC changes at the basin can only give a broad outlook on the status of water quality of the river basin which is also very useful for some applications. Hence, districts were chosen as a unit and district wise population and LULC were related to OIP of the monitoring station to comprehend the causative connection between them. Yes, the spatial/mapped data are more useful or relevant when compared with remote sensing data. But monitoring stations in the Upper Ganga River basin are scarce. Therefore, over a relatively large study area the interpolation maps

generated using OIP are not likely to provide very good comparison results with LULC changes. If the number of monitoring stations are sufficient or large then this method can be used. Hence, with the type of data available for this river basin, it will be better to relate OIP, with population and LULC of complete river basin and districts under study.

**Comment 7:** The remote sensing work seems well carried out. However more detail is needed on the interpretation of the confusion matrices etc. (what is confused with what?; why?; what does this mean for the interpretation of the results?)

**Response 7:** Authors are sincerely thankful to reviewers for appreciating our remote sensing work. With the suggestions from all the three reviewers we have further tried to improve it. In thematic mapping of remotely sensed data, the term accuracy is used typically to express the degree of correctness of a classified map (Foody 2002). The confusion matrix based accuracy assessment is a widely used approach that includes a simple cross-tabulation of the mapped class label against that observed on the ground (or reference data) for a sample of cases at specified locations. A simple random sampling of 649 pixels belonging to corresponding image objects were selected and verified against reference data (GCPs). As a rule of thumb, Congalton (1991) recommends a minimum of 75-100 sample points per category. In LULC classification, a good classification depends on the separability of the spectral signatures while preparing the training sets. It was found that it is difficult to distinguish between built up and wastelands. To avoid this confusion the built up is first masked out and then it is classified independently. Similarly, it is difficult to distinguish between forest and agricultural lands. But the uncertainties in their classification can be reduced by increasing the number of training samples. Snow and glaciers can be confused with wastelands or waterbodies sometimes. The spectral confusion may occur between any classes. Spectral confusion in the classes introduces uncertainties and errors in the classification. Therefore, selection of training sets should be done with care. After image classification, accuracy assessment results are presented in confusion matrix showing characteristic coefficients viz. User's accuracy, Producer's accuracy, Overall accuracy and Kappa coefficients. The User's and Producer's accuracy express the accuracy of each LULC types whereas the overall accuracy estimates the overall mean of user accuracy and producer accuracy. The Kappa coefficient denotes the agreement between two datasets corrected for the expected agreement (Gebremicael et al. 2017). Results from this study showed a good overall accuracy of 90.14% and Kappa coefficient of 0.88. All the LULC classes were classified with accuracy more than 83% and they were in almost similar range. Comparatively highest inaccuracy was observed in forest class (Table 7).

As suggested by the reviewer, the Section 5.3 on accuracy assessment is elaborated in the revised manuscript. Further, characteristic coefficients are described in details in Response 5 of Reviewer#3.

**Table 7.** Accuracy assessment of the 2012 LULC map produced from Landsat Enhanced Thematic Mapper Plus (ETM+) data, representing both the confusion matrix and the Kappa statistics

| Classified Data | Reference Data | | | | | | Row Total | User's Accuracy (%) | Overall Kappa Statistics |
| --- | --- | --- | --- | --- | --- | --- | --- | --- | --- |
| | Agricultural Land | Built Up | Forest | Snow & Glacier | Wastelands | Water Bodies | | | |
| **Agricultural Land** | **128** | 0 | 6 | 0 | 3 | 0 | 137 | 93.43 | |
| **Built Up** | 2 | **96** | 2 | 5 | 1 | 0 | 106 | 90.57 | |
| **Forest** | 11 | 0 | **88** | 3 | 0 | 3 | 105 | 83.81 | |
| **Snow & Glacier** | 0 | 4 | 1 | **103** | 2 | 1 | 111 | 92.79 | |
| **Wastelands** | 1 | 2 | 0 | 7 | **82** | 2 | 94 | 87.23 | 0.88 |
| **Water Bodies** | 0 | 0 | 1 | 1 | 6 | **88** | 96 | 91.67 | |
| *Column Total* | 142 | 102 | 98 | 119 | 94 | 94 | **649** | | |
| *Producer's Accuracy (%)* | 90.14 | 94.12 | 89.80 | 86.55 | 87.23 | 93.62 | | | |
| *Overall Classification Accuracy (%)* | 90.14 | | | | | | | | |

**Comment 8:** There is too much detail in some areas (e.g. full description of equation for population growth rate; detail of full Mann-Kendall method etc.) Only add detail like this if it is needed to help the reader understand the method, or if there is new method development else use references. Much of the Mann-Kendal work in 5.4.1 should be in methods not results.

**Response 8:** Authors are sincerely thankful to reviewers for their suggestions. The full description of equation for population growth rate and details of full Mann-Kendall method have been removed. The paragraphs having the details of these methods are modified. As suggested, only appropriate references are given for their methods. Section 5.1 and 5.4.1 (old manuscript) have been summarised in a one paragraph and write up has been improved wherever required. Section 5.4.1 have been removed from results section and included in methodology section in the modified manuscript.

**References**

Amarasinghe, U. A.; Muthuwatta, L.; Smakhtin, V.; Surinaidu, L.; Natarajan, R.; Chinnasamy, P.; Kakumanu, K. R.; Prathapar, S. A.; Jain, S. K.; Ghosh, N. C.; Singh, S.; Sharma, A.; Jain, S. K.; Kumar, S.; Goel, M. K. 2016. Reviving the Ganges water machine: potential and challenges to meet increasing water demand in the Ganges River Basin. Colombo, Sri Lanka: International Water Management Institute (IWMI). 42p. (IWMI Research Report 167). doi: 10.5337/2016.212.

Congalton, R. G. (1991). A review of assessing the accuracy of classifications of remotely sensed data. Remote sensing of environment, 37(1), 35-46.

Foody, G. M. (2002). Status of land cover classification accuracy assessment. Remote sensing of environment, 80(1), 185-201.

Gebremicael, T. G., Mohamed, Y. A., van der Zaag, P., & Hagos, E. Y. (2017). Quantifying longitudinal land use change from land degradation to rehabilitation in the headwaters of Tekeze-Atbara Basin, Ethiopia. Science of the Total Environment.

Kibena, J., Nhapi, I., & Gumindoga, W. (2014). Assessing the relationship between water quality parameters and changes in landuse patterns in the Upper Manyame River, Zimbabwe. Physics and Chemistry of the Earth, Parts A/B/C, 67, 153-163.

SoE report, 2012: http://www.ucost.in/document/publication/books/env-books.pdf. Accessed on 12 March, 2018.

Watershed Atlas of India, 2014, Ministry of Water Resources, Govt. of India. Accessed on 10 March, 2018.
http://www.indiawris.nrsc.gov.in/Publications/WatershedSubbasinAtlas/Watershed%20Atlas%20of%20India.pdf

---

## Author Comment (AC3) · 14 Mar 2018

**Authors' Response to the Reviewer Comments**

**Manuscript Ref.: hess-2017-384**

**Title: Population Growth – Land Use/Land Cover Transformations-Water Quality Nexus in Upper Ganga River Basin**

**Authors:** Anoop Kumar Shukla, Chandra Shekhar Prasad Ojha, Ana Mijic, Wouter Buytaert, Shray Pathak, Rahul Dev Garg and Satyavati Shukla

We sincerely thanks to the reviewers for their comments on the manuscript and offering their suggestions and critical input that has helped improve the manuscript. We provide here our replies to the reviewers' comments and highlight the changes made in the revised manuscript based on the comments. These have been incorporated in the manuscript as follows. The point wise replies of the comments of the Reviewer#3 are given below.

**General Comments:**

The scientific approach is valid but there is no reference to previous similar studies analysing land use-water quality index relationships (e.g. https://doi/abs/10.2989/16085914. land use-water quality index relationships (e.g. https://doi/abs/10.2989/16085914.2015.1077777, https://doi.org/10.1016/j.proenv.2012.01.140,https://www.ncbi.nlm.nih.gov/pubmed/27498508) nor any justification of the scientific relevance and novelty of the study. In relation to the writing, some parts of the text should be moved to different sections specially from results to methods. There is too much repetition of information along the paper. English is understandable but should be revised (e.g. word confusion, articles, etc.).

We sincerely thank the anonymous reviewer 3 and appreciate his/her efforts in providing very useful comments for the improvements of our contribution. As per reviewers instructions a few case studies are described in the Section 1 "Introduction" of the revised manuscript. The following paragraph has been added in addition to the extensive modifications in it. We are grateful to the reviewer for suggesting us some previous similar studies done related to our theme of work. All the research papers mentioned above and a few others are cited in the introduction section which has helped to improve our contribution.

"Demographic changes and anthropogenic activities have potential to affect the quantity and quality of available water resources on local, regional and global scale in a river basin. These drivers pose a threat to the quantity and quality of water resources directly by increased anthropogenic water demands and water pollution. Indirectly, the water resources are affected by LULC changes and associated changes in water use patterns (Yu et al. 2016). LULC changes may alter the chemical, physical and biological properties of a river system. Several studies are carried out across the world to understand this phenomenon. Hong et al. (2016) studied the effects of LULC changes on water quality of a typical inland lake of arid area in China. The study concluded that water pollution is positively correlated to agricultural land and urban areas whereas negatively correlated to water and grassland. Li et al. (2012) studied effects of LULC changes on water quality in the Liao River basin, China. In this river basin water quality of upstream was found better than downstream due to less influence from LULC changes in the region. Similarly, impact of LULC changes was studied at Likangala catchment, southern Malawi and downstream of the river was found more polluted with increase in the number of *E.Coli* and cation/anions even though the water quality remained in acceptable class (Pullanikkatil et al. 2015). The composition and distribution of benthic

macroinvertebrate assemblage were studied in the Upper Mthatha River, Eastern Cape, South Africa (Niba and Mafereka 2015). Results revealed that the distribution of the benthic macroinvertebrate assemblage is affected by season, substrate and habitat heterogeneity. LULC changes may induce changes into the river water which may affect their species distribution. Water quality changes of the Ganges river at various locations in Allahabad was studied for post-monsoon season by Sharma et al. (2014) using Water Quality Index (WQI) and statistical methods. Considerable water quality deterioration was observed at various locations due to the vicinity of the river to a highly urbanized city of Allahabad. A combination of water quality indices viz. CCME-WQI, Oregon Water Quality Index, (OWQI) and NSF-WQI were used to analyse the pollution of Sapanca Lake Basin (Turkey) and a good relationship was observed between the indices and parameters. Eutrophication was identified as a major threat to Sapanca Lake and stream system (Akkoyunlu and Akiner 2012)".

Justification of the scientific relevance and novelty of the study is addressed in detail in the Response 4 of Reviewer#2. As per reviewer's suggestion, the write up has been improved wherever required and some part of the text have been moved form results to methodology section. And the redundant information is removed. Our endeavour will be that the revised paper is much better than the current version.

**Specific comments:**

**Comment 1:** Keywords - Population or demographic change should be included.

**Response 1:** Word "*Demographic change*" has been incorporated in the keywords.

**Comment 2:** Introduction - Paragraphs could be used to better organise the ideas in the text. Lines 76-86: many water quality indices are cited, but no comment about their validity, similarities (clusters), differences, etc. is made. Why the OIP index is good compared to other? From the methods section, I see that it is only the average of individual indices of different pollutants. Should all pollutants have the same weight according to their impact on health, removal costs…? Does the OIP propose more pollutants apart from those considered in the present study? If not, what would be the approach if there are other relevant pollutants in the studied region? Objectives should be better organised and explained: Not clear that it is not a continuous time analysis, but a 2-time slice analysis (2001 and 2012) with seasonal component. Should the test of OIP as a valid index be principal in the study before it is used to extract conclusions?

**Response 2:** As per reviewer's suggestion, the introduction section is restructured and reorganized into paragraphs in the revised manuscript.

In lines 76-86, the following water quality indices were cited viz. Composite Water Quality Identification Index (CWQII), River Pollution Index (RPI), Forestry Water Quality Index (FWQI), National Sanitation Foundation Water Quality Index (NSFWQI), Canadian Water Quality Index (CWQI), Comprehensive water pollution index of China, Prati's implicit index of pollution, Horton's index, Nemerow and Sumitomo Pollution Index, Bhargava's index, Dinius second index, Smith's index, Aquatic toxicity index, Chesapeake Bay water quality indices, Modified Oregon WQI, Li's regional water resource quality assessment index, Stoner's index, Two-tier WQI, Canadian WQI by Canadian Council of Ministers of the Environment (CCME), Universal WQI, Overall index of pollution (OIP), Coastal WQI for

Taiwan, etc. (Abbasi and Abbasi 2012; Rai et al. 2011). These are the various water quality indices available worldwide that can be used for water quality assessment. Not much literature is currently available on comparisons between all the above mentioned water quality indices based on clusters, differences, validity, etc. However, in a study comparison was made between CCME and DELPHI water quality indices based on multivariate statistical techniques viz. coefficient of determination ($R^2$), root mean square error, and absolute average deviation. Results revealed that the DELPHI method had higher predictive capability than the CCME method (Sinha and Saha 2015). However, there is no worldwide accepted method for development of water quality indices. Therefore, there is no method by which 100% objectivity or accuracy can be achieved without any uncertainties. There is continuing interest across the world to develop accurate water quality indices that suit best for a local or regional area. Each water quality index has its own merits and demerits (Sutadian et al. 2016; Tyagi et al 2013).

Water quality is defined in terms of chemical, physical and biological (bacteriological) characteristics of the water. These characteristics may vary for different regions based on their topography, land use land cover (LU/LC) and climatic factors. The acceptable levels/ranges of concentrations of particular water quality parameters are defined as water quality criteria which is different for different regions/countries and water uses. Water quality management and planning in a river basin requires an understanding of the cumulative or overall pollution effect of all the water quality indicator parameters under consideration. This helps in assessing the overall water quality/pollution status of the river in a given space and time in a specific region. Overall Index of Pollution (OIP) developed by Sargaonkar and Deshpande (2003) is a general water quality classification scheme specifically for tropical Indian conditions where in the proposed classes (Excellent, Acceptable, Slightly Polluted, Polluted and Heavily Polluted water), the concentration levels/ranges of the significant water quality indicator parameters viz. Hardness $CaCO_3$, TDS, BOD, Cl, Coliform Total, Colour, DO%, pH, and Turbidity are defined based on the Indian water quality standards (Indian Standard Specification for Drinking Water, IS-10500, 1983 and Central Pollution Control Board, Government of India, classification of inland surface water, CPCB- ADSORBS/3/78-79). This classification scheme took into consideration various international water quality assessment schemes viz. European Community (EC) standards, World Health Organization (WHO) guidelines, standards by WQIHSR and Tehran Water Quality Criteria by McKee and Wolf. The concentration ranges used in the classes and the classification scheme helped to evaluate the surface water quality status with respect to particular individual parameter whereas the OIP helped to assess the overall water quality status specifically in the Indian context. It helped to identify the parameters which are affected due to pollution from urban and rural areas. OIP is immensely helpful in studying the spatial and temporal variations in the surface water quality status of both rural and urban subbasins due to the influence of demographic and LU/LC changes. The self-cleaning capacity of the river system investigated using OIP helped to comprehend the resilience capacity of the river system against the changes occurring in water quality due to anthropogenic activities. OIP has been used successfully to study the surface water quality status of the two most important and highly polluted rivers viz. Ganga and Yamuna of the tropical Indian region. It is also used for water quality assessment of comparatively smaller river like Chambal River and Sukhna lake of Chandigarh (Chardhry et al. 2013; Katyal et al. 2012; Shukla et al. 2017; Sargaonkar and Deshpande 2003; Yadav et al. 2014). Therefore, OIP is used in the present study as it is an effective tool to communicate the water quality information to concerned policy makers and planners.

Yes, in the OIP the aggregation method used is additive. No, the water quality pollutants should not have same weights with respect to impact on health, removal costs, etc. However, each water pollutant or water quality parameter is important and should not be ignored. It solely depends on the objectives of the work to be done. In addition to the 7 water quality parameters used, OIP considers total 13 water quality parameters viz. pH, turbidity, dissolved oxygen (DO), biochemical oxygen demand (BOD), hardness $CaCO_3$, total dissolved solids (TDS), total coliform and some of the toxicity indicator parameters viz. Arsenic (As) and Fluoride (F). If there are other relevant water quality parameters, the IPIs and OIP can developed for those parameters using methodology given by Sargaonkar and Deshpande (2003) which first involves, proposing a classification scheme for the particular water quality parameters based on some water use standards or pollution permissible limits. The concentration range of the parameters should be defined and a class score should be decided based on the standards. Then mathematical value function curves can be plotted to get the mathematic equations which will help to calculate IPIs. As OIP uses an additive aggregation method, the average of IPIs of all the parameters will estimate OIP.

Objectives of the work are now clearly defined in the revised manuscript. A paragraph is given below regarding the same:

Ganga River is extremely significant to its inhabitants as it supports various important services such as: (i) source of irrigation for farmers in agriculture and horticulture; (ii) provides water for domestic and industrial purposes in urban areas; (iii) source of hydro-power; (iv) serves as a drainage for waste and helps in pollution control; (v) acts as support system for terrestrial and aquatic ecosystems, (vi) provides religious and cultural services; (vii) helps in navigation; (viii) supports fisheries and other livelihood options, etc. (Amarasinghe et al. 2016; SoE report, 2012; Watershed Atlas of India, 2014). However, for the past few decades Upper Ganga River basin has experienced rapid growth in population, urbanization, industrialization, infrastructure development activities and agriculture. Due to these changes, maintaining the acceptable water quality for various uses is being challenged. Therefore, there is a need to study the causative connection (nexus) between the changing patterns of population, Land Use/Land Cover (LULC) and water quality at both river basin (small scale) and at districts level (large scale) for three different seasons. Such study is yet to be done for this large river basin. OIP developed specifically for Indian context is used in this study to assess the status of water quality across the study area. Due to unavailability of the continuous population, satellite and water quality data at desired interval, establishing the interrelationship between these factors is not trivial. Hence, in order to achieve the objectives a comprehensive set of analyses are performed in this study. Between LULC and OIP a 2-time slice analysis is done for the years 2001 and 2012 with seasonal component. A relationship is developed between LULC and OIP (Indian WQI) using correlation and multi linear regression analyses. Further, trend (Mann-Kendall method) analysis was performed on monthly water quality parameters of the monitoring stations from 2001 to 2012 to understand their temporal variations over the years. Also, it was interesting to see the effects of seasonal variations on status of water quality. Hence, finally the results were inferred from these comprehensive set of analyses to understand nexus between population-LULC-water quality of Upper Ganga River basin which is our main contribution. The test of OIP as a valid index can be done to assure the validity of the index. However, a number of studies have successfully used OIP to assess the surface water quality of various Indian rivers.

**Comment 3:** Case study - Justify why the 7 selected pollutants are important and not others – Not sure if the data sources should be detailed after the methods section and be, therefore, better related to each methodological stage.

**Response 3:** Authors have addressed the same question in detail in Response 4 of Reviewer#1. We agree with the reviewer that data sources should not be detailed after the method section. Hence after doing modifications in the methodology flow chart, the text was also updated suitably. And as suggested by the reviewer the data description is done at each methodological stage in methodology section of the manuscript.

**Comment 4:** Methods - There is repetition of information. Try to avoid it by re-organising the text (section 4.1, remove details from the introduction and explain them only in this section). Move table 1 to data section or to results, but it is not part of methods. The classification for OIP that relates the obtained values with an overall water quality status should be included in the methods section, like the IPI classification. The OIP classification is currently described in the results section (5.4.2). Explanation about the link between LULC and water quality at the diverse stations is missing. I assume that the sub-catchments draining to the locations of the water quality stations are defined and this is used to relate the impact of spatial LULC change with water quality. I think it is worth including that as part of the methods section and also in Figure 2.

**Response 4:** In methodology section, repetition text has been removed and re-organised according to the reviewer's suggestion. In introduction section, some text has been removed and explained in the section 4.1.

Table 1 has been removed from methodology section and added in results section.

Individual parameter indices (IPIs) and overall indices of pollution (OIPs) have been removed from the results section (5.4.2) and added in methodology section.

Authors agree with the reviewer that the link between LULC and water quality at the diverse stations was missing in the earlier manuscript. Earlier the manuscript had redundant information due to which the paper size was bigger. But after review, redundant information is identified and removed. Also, restructuring of the manuscript is done in which the description is added in "results and discussion" section with special focus on the link between LULC and water quality. A new subsection is added in the manuscript for describing the relationship developed between LULC and OIP using multi linear regression. Our endeavour will be that the revised paper is much better than the current version.

Reviewer#1 in Comment 7, has suggested to estimate the district specific LULC and to relate it to water quality.

Reviewer#1 (Comment 7): *I would recommend the addition of district specific land use change maps to help support your discussion. At present, it is impossible to visually relate the pattern of land use change to the water quality and population statistics because the scale of the mapping in figure 4 is too small.*

This study will help to determine the causative connection (nexus) between the changing patterns of population, Land Use/Land Cover (LULC) and water quality at both river basin (small scale) and at district level (large scale) for three different seasons. It will help to

understand water quality status of Upper Ganga River basin at both complete river basin and local scale. In addition to this, population information is available for the districts hence, this analysis was done. A relationship was developed between LULC and OIP. This new work done has been updated in the work flowchart accordingly. And further it is updated and explained in the methodology as well as results section.

**Comment 5:** Results - No need to explain how to calculate a % change (lines 27-31). If you decide to include it anyway, it should be placed under the methods section. Figure 3 would be more useful if showing the results for the upper and lower reaches separately, instead of aggregated for the whole basin. It would support the statement in lines 20-21 which is not proved based on results. Table 4 and Figure 5 present the same results. Only one of them should be included in the paper to avoid repetition of information. Lines 95-99 should be moved to the methods section, including a description and reference about the Kappa statistics. The meaning of user's and producer's accuracy is not clear. Lines 124-158 could be moved to methods section as they describe the Mann-Kendal test. Best scenario to select representative months based on what? In figures 6 and 7 it would be useful to depict the OIP thresholds as horizontal lines instead of as a legend. Some discussion about the conclusions and comparison with the current study should be included.

**Response 5:** Authors are sincerely thankful to reviewer 3 for suggestions on improving the write up. We agree that, the formula to calculate % change is generic and it is not required. Therefore it is removed and the text was edited suitably.

As per suggestions of reviewer, the Figure 3 is modified. Total population as well as PGR is estimated and presented for upper and lower reaches of the Upper Ganga River basin. We are grateful to the reviewer for this comment, it helped us to support the statement in lines 20-21 which was not proved based on results. The necessary modifications are done in the paragraphs and it improved the revised manuscript.

**(a)**

[Figure]

**(b)**

[Figure]

**Figure 3:** Growth in the rural and urban population of upper and lower reaches of Upper Ganga River basin between 2001-2011 (a) Total population, and (b) Population Growth Rate (PGR)

As suggested by Reviewer#1 in Comment 9, Table 4 was removed and a cross tabulation table of the 2001 and 2012 LULC classes is presented. It will help reader to see what has changed to what and then the gross and net changes are shown in Figure 5. The table is presented in Response 9 of Reviewer#1.

As suggested, lines (95-99) have been moved to the methodology section, including a description and reference about the Kappa statistics.

Lines 124-158 have been moved to methods section as they describe the Mann-Kendal test in the revised manuscript.

After accuracy assessment of the satellite images an error matrix (confusion matrix) is generated which gives the ratio of number of correctly classified samples to the total number of samples in the reference data. Overall accuracy depicts the accuracy of the whole classification. To determine the accuracy of individual classes two coefficients are used: (a) producer's accuracy, and (b) user's accuracy.

In producer's accuracy, the interest of the image producer is in how well the samples from the reference data can be mapped using remotely sensed data. That is why it is called producer's accuracy. It measures errors of omission, which is a measure of how well a real-world LULC types can be classified. On Contrary, the user's accuracy indicates the probability that a sample from the classified image would actually represent that particular class on the reference data. User's accuracy measures errors of commission, which represents the likelihood of a classified pixel matching the land cover type of its corresponding real-world location. Producer's accuracy is estimated by dividing the number of correctly

classified samples of a class by the column total. Whereas the user's accuracy can be estimated by dividing the number of correctly classified samples of a class by the row total (Campbell 2007; Congalton 1991; Jensen 2005). A description on accuracy assessment and Kappa coefficient is given in Comment 7 of Reviewer 2. Confusion matrix is also presented to support the description.

It is interesting to see in which season the water quality of the river is affected more by LULC changes with respect to climatic variables. Best scenario/representative month means the season in which the water quality of the river is mainly affected by LULC changes not by climatic conditions such as rainfall during monsoon. It is observed based on the OIP values at a monitoring station during different seasons.

As suggested by the reviewer, in Figures 6 and 7 the OIP thresholds are now given as horizontal lines not as a legend. All the sub-figures of Figure 6 and 7 are modified suitably.

As the modifications in the structure of the manuscript were suggested by all the reviewers, hence, the manuscript is thoroughly edited and modified in all the sections of the manuscript wherever it was required. Descriptions are duly added in the results and discussion; and conclusion section of the manuscript.

**(a)**

[Figure]

**(b)**

[Figure]

**(c)**

[Figure]

**Figure 6.** Spatial variations in the overall indices of pollution of upper Ganga River basin for **(a)** Pre-monsoon period **(b)** Monsoon period, **(c)** Post-monsoon period

**(a)**

[Figure]

**(b)**

[Figure]

**(c)**

[Figure]

**Figure 7.** Temporal variations in the overall indices of pollution of upper Ganga River basin for **(a)** Pre-monsoon period **(b)** Monsoon period, **(c)** Post-monsoon period

**References**

Abbasi, T., and Abbasi, S. A. (2012). "Water quality indices." Elsevier.

Akkoyunlu, A., & Akiner, M. E. (2012). Pollution evaluation in streams using water quality indices: A case study from Turkey's Sapanca Lake Basin. Ecological Indicators, 18, 501-511.

Amarasinghe, U. A.; Muthuwatta, L.; Smakhtin, V.; Surinaidu, L.; Natarajan, R.; Chinnasamy, P.; Kakumanu, K. R.; Prathapar, S. A.; Jain, S. K.; Ghosh, N. C.; Singh, S.; Sharma, A.; Jain, S. K.; Kumar, S.; Goel, M. K. 2016. Reviving the Ganges water machine: potential and challenges to meet increasing water demand in the Ganges River Basin. Colombo, Sri Lanka: International Water Management Institute (IWMI). 42p. (IWMI Research Report 167). doi: 10.5337/2016.212.

Campbell, J.B. (2007) Introduction to Remote Sensing. 4th Edition, The Guilford Press, New York.

Chardhry, P., Sharma, M. P., Bhargava, R., Kumar, S., & Dadhwal, P. J. S. (2013). Water quality assessment of Sukhna Lake of Chandigarh city of India. Hydro Nepal: Journal of Water, Energy and Environment, 12, 26-31.

Congalton, R. G. (1991). A review of assessing the accuracy of classifications of remotely sensed data. Remote sensing of environment, 37(1), 35-46.

Hong, C., Xiaode, Z., Mengjing, G., & Wei, W. (2016). Land use change and its effects on water quality in typical inland lake of arid area in China. Journal of environmental biology, 37(4), 603.

Jensen, J.R. (2005) Introductory Digital Image Processing: A Remote Sensing Perspective. 3rd Edition, Pearson Prentice Hall, Upper Saddle River, NJ.

Katyal, D., Qader, A., Ismail, A. H., & Sarma, K. (2012). Water quality assessment of Yamuna River in Delhi region using index mapping. Interdisciplinary Environmental Review, 13(2-3), 170-186.

Li, Y. L., Liu, K., Li, L., & Xu, Z. X. (2012). Relationship of land use/cover on water quality in the Liao River basin, China. Procedia Environmental Sciences, 13, 1484-1493.

Niba, A. S., & Mafereka, S. P. (2015). Benthic macroinvertebrate assemblage composition and distribution pattern in the upper Mthatha River, Eastern Cape, South Africa. African Journal of Aquatic Science, 40(2), 133-142.

Pullanikkatil, D., Palamuleni, L. G., & Ruhiiga, T. M. (2015). Impact of land use on water quality in the Likangala catchment, southern Malawi. African journal of aquatic science, 40(3), 277-286.

Rai, R. K., Upadhyay, A., Ojha, C. S. P., and Singh, V. P. (2011). "The Yamuna river basin: water resources and environment." Springer Science & Business Media, 66.

Sargaonkar, A., & Deshpande, V. (2003). Development of an overall index of pollution for surface water based on a general classification scheme in Indian context. Environmental monitoring and assessment, 89(1), 43-67.

Sharma, P., Meher, P. K., Kumar, A., Gautam, Y. P., & Mishra, K. P. (2014). Changes in water quality index of Ganges river at different locations in Allahabad. Sustainability of Water Quality and Ecology, 3, 67-76.

Shukla, A. K., Shukla, S., & Ojha, R. (2017). Geospatial Technologies for Rainfall and Atmospheric Water Vapor Measurement over Arid Regions of India. In Sustainable Water Resources Management (pp. 263-292).

Sinha, K., & Das, P. (2015). Assessment of water quality index using cluster analysis and artificial neural network modeling: a case study of the Hooghly River basin, West Bengal, India. Desalination and Water Treatment, 54(1), 28-36.

SoE report, 2012: http://www.ucost.in/document/publication/books/env-books.pdf. Accessed on 12 March, 2018.

Sutadian, A. D., Muttil, N., Yilmaz, A. G., & Perera, B. J. C. (2016). Development of river water quality indices—a review. Environmental monitoring and assessment, 188(1), 58.

Tyagi, S., Sharma, B., Singh, P., & Dobhal, R. (2013). Water quality assessment in terms of water quality index. American Journal of Water Resources, 1(3), 34-38.

Watershed Atlas of India, 2014, Ministry of Water Resources, Govt. of India. Accessed on 10 March,2018.http://www.indiawris.nrsc.gov.in/Publications/WatershedSubbasinAtlas/Watershed%20Atlas%20of%20India.pdf

Yadav, N. S., Kumar, A., & Sharma, M. P. (2014). Ecological health assessment of Chambal River using water quality parameters. Journal of Integrated Science and Technology, 2(2), 52-56.

Yu, S., Xu, Z., Wu, W., & Zuo, D. (2016). Effect of land use types on stream water quality under seasonal variation and topographic characteristics in the Wei River basin, China. Ecological indicators, 60, 202-212.

---

## Author Response (AR2)

**Authors' Response to the Editors Comments**

**Manuscript Ref.: hess-2017-384**

**Title: Population Growth – Land Use Land Cover Transformations-Water Quality Nexus in Upper Ganga River Basin**

**Authors:** Anoop Kumar Shukla, Chandra Shekhar Prasad Ojha, Ana Mijic, Wouter Buytaert, Shray Pathak, Rahul Dev Garg and Satyavati Shukla

We sincerely thank the editor for offering critical comments and valuable suggestions that has helped to improve the manuscript. We hereby provide our responses to the editor comments and highlight the changes made in the revised manuscript based on the comments provided. These have been incorporated in the revised manuscript as follows. The point wise replies of the comments of the editor are given below:

**General Comments:**

The authors have made a satisfactory revision of the manuscript, addressing the main concerns of the reviewers regarding placing the research and its aim within the wider literature; correcting the imbalance in level of detail through the manuscript; re-structuring to improve the narrative flow and enhancing the methodological detail (particularly with regard to the processing of the remote sensing data).

**Response:** Authors are sincerely thankful to the editor for appreciating our efforts and remote sensing work. With the given suggestions, we have further tried to improve the manuscript. We are grateful to the editor for his/her efforts in providing extremely useful comments that has helped in improving our research work further. The comments provided by editor are duly complied in the revised manuscript, which we believe have significantly improved the manuscript.

**Comments to the author:**

**Comment 1:** However, the methodological description of the derivation of the OIP and IPIs (and the use of the information within Tables 1 and 2) is still insufficiently clear and needs to be improved.

**Non-public comments to the author:**

**Comment 2:** The methodological description of the derivation of the OIP and IPIs (and the use of the information within Tables 1 and 2) is still insufficiently clear and needs to be improved. The lack of clarity partly (but not completely) arises because the concentration limits in Table 1 and Table 2 are different.

**Comment 3:** However, the OIP is derived from the average of the IPIs, so the purpose of the Class Index / Score of the OIP in Table 1 is not clear. The definition of x and y in Table 2 are not given. Is x the calculated value of the IPI, and y the Score from Table 1 for a given concentration?

**Response 1-3:** Authors are sincerely thankful to the editor for pointing out this very important question. Section "4.4.3 Estimation of OIP" in the manuscript has been elaborated significantly while answering the questions/comments 1 to 3 from the editor. The

methodology used to decide the concentration ranges of water quality parameters, class index score (Table 1), mathematical value function curves, their equations as well as terms (Table 2), estimation of IPIs and OIP are described in details for better understanding of the OIP. The updated, detailed methodology and text is highlighted in the manuscript as follows:

[revised manuscript text omitted]

*"The lack of clarity partly (but not completely) arises because the concentration limits in Table 1 and Table 2 are different".*

Table 1 represents the classes considered in the OIP classification scheme, respective class index score in geometric progression and the concentration limits/ranges of water quality parameters defined based on various water quality classification schemes by national and international agencies for different water uses. In Table 2 mathematical equations are given for value function curves. For each of the parameter concentration levels, the mathematical

expressions were fitted to obtain the numerical value called an index ($P_i$) or (IPI) which indicated the level of pollution for that particular parameter. For any particular given concentration (including the concentration limit/range of parameters in various classes in Table 1), the corresponding index can be read directly from these curves or can be estimated using mathematical equations of value function curves illustrated in Table 2.

[Figure]

Figure 1. Value function curves for important water quality parameters in the given scheme. (Source: Sargaonkar and Deshpande 2003)

**Comment 4:** A few additional minor text edits are also required. Also please ensure that all figures are of appropriate resolution /dpi e.g. Figure 5.

**Response 4:** Authors are sincerely thankful to the editor for pointing it out. Editor suggestions regarding the additional minor text edits, proofreading of the manuscript has been conducted and typos are corrected in the revised manuscript. Authors have corrected the missing words and grammar, improved the phraseology and checked the English in whole manuscript. Authors have improved the quality of all the Figures 1 to 6 in the manuscript as suggested by the editor. The modified figures are of appropriate resolution i.e. 1000 dpi. The improved Figure 1 to Figure 6 have been updated in the revised manuscript. Our endeavour will be that the revised paper is much better than the current version.

*Non-public comments to the author:*

**Comment 5:** L479 states that population increased in all 77 districts but L483 and Table 3 states that PGR only increased in 74 districts. Correct this inconsistency.

**Response 5:** Authors are sincerely thankful to the reviewer for pointing it out. We checked the results obtained from demographic analysis and as suggested by Editor, in section "5.1 Population dynamics" the sentence L479 is modified and the paragraph is updated accordingly in the manuscript. L478-491 is modified as follows:

"Analysis of the population dataset of the years 2001 and 2011 acquired from Census of India, GoI reveals that in the UGRB, out of the 77 districts that fall in four different states, viz. Uttar Pradesh, Uttarakhand, Bihar and Himanchal Pradesh, total population and PGR has increased in 74 districts. With majority of the districts showing population increase, the total population of UGRB has increased consequently (Table 3). The population growth rate (PGR) of 20.45% is observed in the total population of UGRB from 2001 to 2011. Table 3 illustrates that the PGR is ≥20% in the districts having bigger urban agglomerations or cities e.g. Agra, Allahabad, Bahraich, Ghaziabad, Lucknow, Kanpur (Dehat+Nagar), Varanasi, Patna, etc. However, Almora, Pauri Garhwal and Shravasti are showing decreasing PGR. It is to be observed that these are either hilly or very small towns with poor employment opportunities. People migrate from these locations to nearby cities, therefore, decreasing the PGR. It was noticed from Census of India reports that the population density of Dehradun (Rishikesh), Kanpur, Allahabad and Varanasi districts are much higher against the average population density of Ganga River basin, i.e. 520 per square km. Varanasi is one of the most populated districts in the country".

**Comment 6:** L620-627 includes significant repetition - please condense.

**Response 6:** Authors are sincerely grateful to the reviewer for pointing it out.

As suggested by Editor, the authors have modified section "5.4 Trend analysis on monthly water quality data" and L618-630 are as updated follows: "From the results of trend analysis (Mann Kendall rank test) it is observed that each water quality parameter varies with time

and location, hence the changes in the water quality parameters are observed in all the months (Table 7). No regular trends are observed in the water quality data, therefore, they are very site-specific. Results from statistical analyses reflect that comparatively high SD and significant changes are observed in water quality of the monsoon month (July), which is followed by pre-monsoon and post-monsoon months in decreasing order. Effect of different seasons on water quality is reported from various studies (Islam et al. 2017; Sharma and Kansal 2011; Singh and Chandna 2011). In this study, three significant seasons are identified and hence the water quality data is organized into three groups: pre-monsoon season (February-May), monsoon season (June-September) and post-monsoon season (October-January). From each group, one representative month i.e. May, July, November month is chosen, which represents that particular season the best".